# GENERAL STABILITY ANALYSIS FOR ZEROTH-ORDER OPTIMIZATION ALGORITHMS

**Xinyue Liu**[1,2], **Hualin Zhang** [4], **Bin Gu**[3,4*], **Hong Chen**[1,2*]

[1]College of Informatics, Huazhong Agricultural University, China
[2]Engineering Research Center of Intelligent Technology for Agriculture, China
[3]School of Artificial Intelligence, Jilin University, China
[4]Mohamed bin Zayed University of Artificial Intelligence, UAE
`lxy36@webmail.hzau.edu.cn,zhanghualin98@gmail.com,`
`jsgubin@gmail.com, chenh@mail.hzau.edu.cn`

## ABSTRACT

Zeroth-order optimization algorithms are widely used for black-box optimization problems, such as those in machine learning and prompt engineering, where the gradients are approximated using function evaluations. Recently, a generalization result was provided for zeroth-order stochastic gradient descent (SGD) algorithms through stability analysis. However, this result was limited to the vanilla 2-point zeroth-order estimate of Gaussian distribution used in SGD algorithms. To address these limitations, we propose a general proof framework for stability analysis that applies to convex, strongly convex, and non-convex conditions, and yields results for popular zeroth-order optimization algorithms, including SGD, GD, and SVRG, as well as various zeroth-order estimates, such as 1-point and 2-point with different distributions and coordinate estimates. Our general analysis shows that coordinate estimation can lead to tighter generalization bounds for SGD, GD, and SVRG versions of zeroth-order optimization algorithms, due to the smaller expansion brought by coordinate estimates to stability analysis.

## 1 INTRODUCTION

Zeroth-order (ZO) optimization algorithms have gained widespread use in solving black-box optimization problems, particularly in machine learning fields (Chen et al., 2017; Kurakin et al., 2016; Madry et al., 2017; Sun et al., 2022). In such problems, obtaining the explicit gradient is often either unattainable or too expensive, and only the function value of the output can be obtained. For instance, pre-trained language models (PTMs) like GPT-3 (Brown et al., 2020) allow only user-designed task-specific prompts to query them, and their gradients are usually unavailable. Similarly, in scenarios where the internal structure and training data of a deep neural network (DNN) are unknown, as in (Papernot et al., 2017), and only the input and output are accessible, the black-box attack can be launched to manipulate the model output and cause incorrect predictions. Here, ZO algorithms can achieve excellent performance, comparable or even better than white-box models where gradients are known (Sun et al., 2022; Papernot et al., 2017).

In ZO optimization algorithms, the approximation of gradients is achieved by using function values. The accuracy of these zeroth-order (ZO) gradient estimates is critical for determining the convergence rate and optimal learning rate settings. Therefore, conducting research to improve the accuracy of ZO gradient estimates is of utmost importance. There are three main categories of ZO gradient estimates: 1-point (Flaxman et al., 2004), 2-point (Nesterov & Spokoiny, 2017; Duchi et al., 2015) with different distributions, and coordinate estimates. As the number of function evaluations increases, the accuracy of the gradient estimates improves (Duchi et al., 2015). When the number of function evaluations reaches the problem dimension $d$, it is preferable to use the deterministic coordinate-wise gradient

---

*Corresponding authors.

estimate instead of the 1-point and 2-point gradient estimates with random directions. This estimate has a significantly lower approximation error (Kiefer & Wolfowitz, 1952; Berahas et al., 2022; Lian et al., 2016), of order $\mathcal{O}\left(d\mu^2\right)$ (where $\mu$ is the smoothing parameter).

Multiple ZO algorithms have been proposed to address the core problem of minimizing the loss function in optimization. Initially, the ZO gradient descent (ZO-GD) algorithm (Nesterov & Spokoiny, 2017) was the focus of the ZO optimization algorithm field. Yurii et al. (Nesterov & Spokoiny, 2017) proposed the ZO-GD algorithm that employs a 2-point Gaussian random gradient estimator and established convergence guarantees for it. Subsequently, Saeed et al. (Ghadimi & Lan, 2013) introduced the ZO-SGD algorithm that uses the same estimator. However, due to the high variance of the ZO stochastic gradients, the convergence rate of ZO-SGD is limited to $\mathcal{O}\left(\sqrt{1/T}\right)$ (Ghadimi & Lan, 2013) for non-convex problems, where $T$ is the total number of iterations. To accelerate its convergence, some variance reduction strategies commonly used in first-order optimization algorithms, such as SVRG and SAGA, were introduced into the ZO setting. These strategies have been shown to be effective in the first-order setting and are also effective in the ZO setting. The ZO-SVRG (Liu et al., 2018; Ji et al., 2019) and ZO-SAGA (Huang et al., 2019) algorithms can improve the convergence rate to $\mathcal{O}\left(1/T\right)$.

As mentioned above, research on ZO algorithms in the optimization field is extensive and comprehensive. However, there is only one generalization study that focuses specifically on ZO-SGD (Nikolakakis et al., 2022). In their study, Konstantinos E. et al. (Nikolakakis et al., 2022) provide generalization results of ZO-SGD under a two-point Gaussian smoothing estimate and demonstrate that the algorithm has comparable generalization performance to SGD in non-convex environments when using diminutive step sizes. While Konstantinos E. et al. (Nikolakakis et al., 2022) presents surprising results, it does not encompass the diverse range of estimates and optimization algorithms in the ZO optimization field. In contrast to the limited research on ZO generalization, first-order analysis has yielded a more diverse body of work, specifically regarding algorithms such as SGD (Hardt et al., 2016), GD (Hoffer et al., 2017; Charles & Papailiopoulos, 2018), and stage-wise learning strategy (Yuan et al., 2019). These discussions have inspired us to further explore the generalization performance of ZO algorithms.

To address the limitations of existing research, this paper focuses on studying the generalization behavior of a more comprehensive ZO optimization algorithm and provides a more inclusive proof framework.The contributions of this work are as follows:

- We propose a more general proof framework for ZO optimization algorithms. Leveraging our proof framework, we provide for the first time generalization bounds for different ZO algorithms, including ZO-SGD, ZO-GD, and ZO-SVRG with different ZO gradient estimates under convexity and non-convexity conditions. To the best of our knowledge, this is the first work on generalization analysis for ZO-SVRG.

- ZO optimization algorithms differ from first-order optimization algorithms in that they approximate gradient values using function values. As a result, approximation errors introduce new expansions for generalization analysis. Our results show that, ZO algorithms that use coordinate estimates lead to tighter generalization bounds for ZO-SGD, ZO-GD and ZO-SVRG, and importantly same to the ones of first-order algorithms.

## 2 PRELIMINARIES

### 2.1 NOTATIONS

Throughout this paper, we let $\mathrm{N}\left(0, \mathbf{I}\right)$ denote the standard normal distribution on $\mathbb{R}^d$. We denote the $\ell_2$-ball in $\mathbb{R}^d$ with radius $r$ centered at $v$ by $\mathbb{B}^d(v, r)$, and $\mathbb{S}^{d-1}(v, r)$ denotes the $\ell_2$-sphere in $\mathbb{R}^d$ with radius $r$ centered at $v$. We also use the abbreviations $\mathbb{B}^d = \mathbb{B}^d(0, 1)$, $\mathbb{S}^{d-1} = \mathbb{S}^{d-1}(0, 1)$, and $\mathbf{1}$ for the all-ones vector.

### 2.2 PROBLEM SETUP

Let $D := \{z_i\}_{i=1}^n$ be the training dataset of size $n$, where each sample is independently identically distributed over an unknown distribution $\mathcal{D}$. For a prescribed model $w$, its performance on a single

Table 1: A list of the generalization error bounds developed here in for zeroth-order optimization algorithms. (NC, C and SC are the abbreviations of nonconvex, convex, and strongly convex, respectively. $T$ is the whole iteration number, $n$ is the sample size, $C$ is a constant, beta is the smoothness constant, and $S$ denotes the stage number of ZO-SVRG. The corresponding details for SVRG are provided in the appendix. )

| Algorithms | Reference | Gradient Estimator | Problem | Generalization Bound |
|---|---|---|---|---|
| SGD | Hardt et al. (2016) | None | NC | $\mathcal{O}\left(T^{\frac{\beta C}{\beta C+1}}/n\right)$ |
| | | | C | $\mathcal{O}\left(\log T/n\right)$ |
| | | | SC | $\mathcal{O}\left(1/n\right)$ |
| ZO-SGD | Nikolakakis et al. (2022) | 2-point | NC, C, SC | $\mathcal{O}(T/n)$ |
| | Ours | 2-point | NC, C, SC | $\mathcal{O}\left(T/n\right)$ |
| | Ours | 1-point | NC, C, SC | $\mathcal{O}\left(T/n^{1-\frac{c}{d^2 LC\sqrt{n}}}\right)$ |
| | Ours | Coordinate-wise | NC | $\mathcal{O}\left(T^{\frac{\beta C}{\beta C+1}}/n\right)$ |
| | | | C | $\mathcal{O}\left(\log T/n\right)$ |
| | | | SC | $\mathcal{O}\left(1/n\right)$ |
| GD | Nikolakakis et al. (2022) | None | NC | $\mathcal{O}\left(T^{\beta C}/n\right)$ |
| | | | C | $\mathcal{O}\left(\log T/n\right)$ |
| | | | SC | $\mathcal{O}\left(1/n\right)$ |
| ZO-GD | Nikolakakis et al. (2022) | 2-point | NC | $\mathcal{O}(T^{\beta C}/n)$ |
| | Ours | 1-point | NC, C, SC | $\mathcal{O}\left(T^{C}/n\right)$ |
| | Ours | Coordinate-wise | NC | $\mathcal{O}\left(T^{\beta C}/n\right)$ |
| | | | C | $\mathcal{O}\left(\log T/n\right)$ |
| | | | SC | $\mathcal{O}\left(1/n\right)$ |
| SVRG | Ours | None | NC | $\mathcal{O}\left(S^{3\beta C}/n\right)$ |
| | | | C | $\mathcal{O}\left(S^{2\beta C}/n\right)$ |
| | | | SC | $\mathcal{O}\left(S/n\right)$ |
| ZO-SVRG | Ours | 1-point | NC, C, SC | $\mathcal{O}\left(S^{3C}/n\right)$ |
| | | 2-point | NC, C, SC | $\mathcal{O}\left(S^{3\beta C}/n\right)$ |
| | | Coordinate-wise | NC | $\mathcal{O}\left(S^{3\beta C}/n\right)$ |
| | | | C | $\mathcal{O}\left(S^{2\beta C}/n\right)$ |
| | | | SC | $\mathcal{O}\left(S/n\right)$ |

example $z$ is measured by the loss function $f(w, z)$. In this paper, we are particularly interested in the generalization performance of a model $w$ on $\mathcal{D}$ measured by the following expected risk:

$$R(w) := \mathbb{E}_{z\sim\mathcal{D}}[f(w, z)]. \tag{1}$$

Since $\mathcal{D}$ is unknown, optimizing $R(w)$ directly is not possible. In practice, we measure the model $w$ by the empirical risk based on the training dataset $D$:

$$R_D(w) := \frac{1}{n}\sum_{i=1}^{n} f(w, z_i). \tag{2}$$

For a (randomized) algorithm $\mathcal{A}$ trained on dataset $D$ with output $\mathcal{A}(D)$, the generalization error is defined as the gap between the population risk and empirical risk:

$$\epsilon_{\text{gen}} := \mathbb{E}_{D,\mathcal{A}}[R(\mathcal{A}(D)) - R_D(\mathcal{A}(D))]. \tag{3}$$

A useful tool for analyzing the generalization error is the stability, which measures the sensitivity of the algorithm's output w.r.t. the perturbation of a training dataset. The formal definition of uniform stability is as follows:

**Definition 1** (Uniform Stability). *A (randomized) algorithm $\mathcal{A}$ has uniform stability $\epsilon$ if for all $D, D' \in \mathcal{Z}$ that differ in one sample, we have $\sup_z \mathbb{E}_{\mathcal{A}}[f(\mathcal{A}(\mathcal{D}), z) - f(\mathcal{A}(\mathcal{D}'), z)] \leq \epsilon$.*

An important relationship between the uniform stability and generalization error bound is implied by the following lemma:

**Lemma 1** (Hardt et al. (2016), Theorem 2.2). *Let $\mathcal{A}$ be $\epsilon$ uniform stable, then $|\epsilon_{gen}| \leq \epsilon$.*

The goal of this paper is to derive uniform stability bounds for a class of zeroth-order (randomized) algorithms with update rule of the form:

$$w_{t+1} = w_t - \alpha_t \mathcal{G}_f(w_t), \tag{4}$$

where $\mathcal{G}_f(w_t)$ depends only on the evaluations of the function values. Denote by $\hat{\nabla} f(w, z)$ the gradient estimation of the true gradient $\nabla f(w_t, z)$, which will be detailed described in the next subsection. Then, the update rule Eq.equation 4 covers a wide range of gradient estimation based zeroth-order (gradient-free) algorithms. For example, zeroth-order gradient descent (ZO-GD) with $\mathcal{G}_f(w_t, z) = \hat{\nabla} R_D(w_t)$; zeroth-order stochastic gradient descent (ZO-SGD) with $\mathcal{G}_f(w_t, z) = \hat{\nabla} f(w_t, z_{i_t})$, where $z_{i_t} \in D$ is a randomly sampled example at iteration $t$; zeroth-order stochastic variance reduced gradient (ZO-SVRG) with $\mathcal{G}_f(w_t) = \hat{\nabla} f(w_t, z_{i_t}) - \hat{\nabla} f(\tilde{w}, z_{i_t}) + \hat{\nabla} R_D(\tilde{w})$, where $\tilde{w}$ is the final output of ZO-SVRG in the last stage. Now we formally describe the gradient estimations that are commonly used in the literature of zeroth-order optimization.

## 2.3 ZO GRADIENT ESTIMATION

**Random Gradient Estimation.** Intuition for zeroth-order gradient estimator follows by the fact about definition of directional derivative that:

$$\mathbb{E}[f'(w, u)u] = \mathbb{E}[< \nabla f(w), u > u] = \nabla f(w) \quad \text{when} \quad \mathbb{E}[uu^\top] = \mathbf{I},$$

where $f'(w, u) := \lim_{\mu \downarrow 0} \frac{f(w + \mu u) - f(w)}{\mu}$ denotes the directional directive of function $f$ with respect to a direction vector $u$ at point $w$.

**Lemma 2** (Baydin et al. (2022), Lemma 1 & 2). *Let $u = [u^1, \ldots, u^d]^\top$, and $u^i, i = 1, \ldots, d$ are i.i.d random variables satisfying $\mathbb{E}[u^i] = 0$, $Var[u^i] = 1$, then we have $\mathbb{E}[uu^\top] = \mathbf{I}$.*

Note that, several random distributions satisfy the above condition. For example, the standard multivariate normal distribution: $N(0, \mathbf{I})$, uniform distribution over a ball or a sphere: $\mathrm{Unif}\left(\sqrt{d+2}\mathbb{B}^d\right)$ and $\mathrm{Unif}\left(\sqrt{d}\mathbb{S}^{d-1}\right)$. Based on the fact that $\mathbb{E}\frac{f(w+\mu u)-f(w)}{\mu}u = \mathbb{E}\frac{f(w+\mu u)}{\mu}u$, the averaged 1-point gradient estimate of $f$ has the following generic form:

$$\hat{\nabla} f(w) := \frac{1}{\mu K} \sum_{k=1}^{K} f(w + \mu u_k) u_k, \tag{5}$$

where $u_k, k = 1, \ldots, K$ are i.i.d random vectors satisfying Lemma 2. It is easy to verify that the 1-point gradient estimate suffers from large variance. As a comparison, the averaged 2-point gradient estimate can significantly reduce the variance:

$$\hat{\nabla} f(w) := \begin{cases} \frac{1}{K} \sum_{k=1}^{K} \frac{f(w+\mu u_k)-f(w)}{\mu} u_k, & \text{forward difference,} \\ \frac{1}{K} \sum_{k=1}^{K} \frac{f(w+\mu u_k)-f(w-\mu u_k)}{2\mu} u_k, & \text{central difference.} \end{cases} \tag{6}$$

**Deterministic Coordinate-Wise Gradient Estimation:** When the number of function evaluations reaches the problem dimension $d$, then instead of using randomized directions $\{u_i\}_{i=1}^{d}$, one can employ the deterministic coordinate-wise gradient estimate:

$$\hat{\nabla} f(w) := \begin{cases} \sum_{i=1}^{d} \frac{f(w+\mu \mathbf{e}_i)-f(w)}{\mu} \mathbf{e}_i & \text{forward difference,} \\ \sum_{i=1}^{d} \frac{f(w+\mu \mathbf{e}_i)-f(w-\mu \mathbf{e}_i)}{2\mu} \mathbf{e}_i, & \text{central difference.} \end{cases} \tag{7}$$

where $\mathbf{e}_i \in \mathbb{R}^d$ denotes the $i$ th elementary basis vector, with 1 at the $i$-th coordinate and 0 elsewhere.

# 3 A GENERIC GENERALIZATION ANALYSIS FRAMEWORK OF ZO OPTIMIZATION

In this section, we will establish a generic generalization analysis framework for zeroth-order optimization with update rule of the form Eq.equation 4. By derivation, we conclude that as the approximate gradient approximates the true gradient, the generalization boundary will also tighten with this approximation. The smaller the error of between the approximate gradient and the true gradient is, the better the generalization of the model is.

Let $G_t(\cdot)$ and $G'_t(\cdot)$ be update rules of first-order methods under samples $D$ and $D'$ respectively, which take the form of

$$G_t(w_t) = w_t - \alpha_t \mathcal{G}(w_t, z_{i_t}), \quad G'_t(w_t) = w_t - \alpha_t \mathcal{G}(w_t, z'_{i_t}), \tag{8}$$

where $\mathcal{G}(w_t, z_{i_t})$ and $\mathcal{G}(w_t, z'_{i_t})$ represent the gradient estimation of $\nabla R_D(w_t)$ and $\nabla R_{D'}(w_t)$ via first-order oracle. Similarly, let $\tilde{G}_t(\cdot)$ and $\tilde{G}'_t(\cdot)$ be the update rules of zeroth-order methods as in Eq.equation 4, i.e.,

$$\tilde{G}_t(w_t) = w_t - \alpha_t \mathcal{G}_f(w_t, z_{i_t}), \quad \tilde{G}'_t(w_t) = w_t - \alpha_t \mathcal{G}_f(w_t, z'_{i_t}), \tag{9}$$

where $\mathcal{G}_f(w_t, z_{i_t})$ and $\mathcal{G}_f(w_t, z'_{i_t})$ denote the gradient estimation via zeroth-order oracle. Hardt et al. (2016) provided a proof framework for SGD, but this framework does not apply to more complex algorithms. Therefore, we present two new sequences of update rules.

**Definition 2** (Hardt et al. (2016), Definition 2.3). *An update rule is $\sigma$-bounded if* $\sup_{w \in \Omega} \|G(w) - w\| \leq \sigma$, *and is $\eta$-expansive if* $\sup_{w,v \in \Omega} \|G(w) - G(v)\| \leq \eta \|w - v\|$.

With these two properties, we can establish the following lemma to demonstrate how the update sequence of the model diverges when the training set is perturbed.

**Lemma 3** (Growth Recursion). *Let $\{\tilde{G}_t\}_{t=1}^T$ and $\{\tilde{G}'_t\}_{t=1}^T$ be two update sequences. Let $w_0 = w'_0$ be the starting point, $w_{t+1} = \tilde{G}_t(w_t)$ and $w'_{t+1} = \tilde{G}'_t(w_t)$. Then for any $w_t, w'_t \in \mathbb{R}^d$ and $t \geq 0$, we have*

$$\mathbb{E}\left[\left\|\tilde{G}_t(w_t) - \tilde{G}'_t(w'_t)\right\|\right] \leq \begin{cases} (\eta + \eta'_t)\|w_t - w'_t\| + \alpha_t \sigma_{t,1}, & \text{if } \tilde{G}_t(\cdot) = \tilde{G}'_t(\cdot), \\ \|w_t - w'_t\| + 2\alpha_t \sigma_{t,2} + \alpha_t \sigma_{t,3}, & \text{if } \tilde{G}_t(\cdot) \neq \tilde{G}'_t(\cdot). \end{cases}$$

The generalization analysis of zeroth-order optimization is similar to the proof framework proposed in Hardt et al. (2016). Here, $\eta$ is the expansive factor generated by the first-order algorithms based on different convexity properties. $\eta'_t$ is induced by the approximation error of the zeroth-order gradient estimation. And different estimation would induce different $\eta'_t$. Particularly, in the case of coordinate-wise estimation, there won't introduce a new $\eta'_t$. Thus the shrinkage effect brought about by $\eta$ becomes evident under convex and strongly convex conditions. $\sigma_{t,1}$ is induced by the approximation error of the zeroth-order gradient estimation, which is an upper bound of $\mathbb{E}\|\mathcal{G}(w_t, z_{i_t}) - \mathcal{G}_f(w_t, z_{i_t}) - (\mathcal{G}(w'_t, z_{i_t}) - \mathcal{G}_f(w'_t, z_{i_t}))\|$. $\sigma_{t,2}$ is induced by the fact that the first-order update rule $G_t(\cdot)$ is $\sigma_{t,2}$-bounded at iteration $t$. Finally, $\sigma_{t,3}$ is induced by another approximation error of the zeroth-order gradient estimation, which is an upper bound of $\mathbb{E}\|\mathcal{G}(w_t, z_{i_t}) - \mathcal{G}_f(w_t, z_{i_t}) - (\mathcal{G}(w'_t, z'_{i_t}) - \mathcal{G}_f(w'_t, z'_{i_t}))\|$.

**Lemma 4.** *Consider zeroth-order algorithms with final output $w_T$ and $w'_T$ with respect to dataset $D, D'$, respectively. Denote $\delta_T := \|w_T - w'_T\|$ and $\forall t_0 \in \{0, 1, \ldots, n\}$, define the event $\mathcal{E}_{\delta_{t_0}} := \{\delta_{t_0} = 0\}$, we have*

$$\mathbb{E}[\delta_T | \mathcal{E}_{\delta_{t_0}}] \leq \sum_{t=t_0+1}^T \left(\eta + n^{-1}(1-\eta)\right)^{T-t} \alpha_t \left[(1 - n^{-1})\sigma_{t,1} + n^{-1}(2\sigma_{t,2} + \sigma_{t,3})\right]$$

Finally, we can obtain the generalization error through the following lemma:

**Lemma 5** (Hardt et al. (2016), Lemma 3.11). *Assume that the loss function $f(\cdot, z)$ is L-Lipschitz for all $z$. Let $D$ and $D'$ be two samples of size $n$ that differ in only one sample. Denote by $w_T$ and $w'_T$ the final output of zeroth-order algorithms with respect to dataset $D$ and $D'$ respectively. Then for every $z$ and $t_0 \in \{0, 1, \ldots, n\}$, we have $\mathbb{E}|f(w_T, z) - f(w'_T, z)| \leq L\mathbb{E}[\delta_T | \mathcal{E}_{\delta_{t_0}}] + t_0 \sup_{w,z} f(w, z)/n$.*

## 4 APPLICATIONS

In this section, we apply the generic generalization analysis to different zeroth-order algorithms such as zeroth-order stochastic gradient descent, and zeroth-order stochastic variance reduced gradient.

### 4.1 ZO-SGD GENERALIZATION ANALYSIS

**ZO-SGD with 1-Point Gradient Estimation.** To begin with, we first define some basic notations used in zeroth-order stochastic gradient descent with averaged 1-point gradient estimation of the form Eq.equation 5. At the $t$-th iteration, we uniformly randomly select a sample index $i_t$ from $[n]$, and then update parameter $w_t$ by

$$w_{t+1} = w_t - \alpha_t \hat{\nabla}_1^K f(w_t, z_{i_t}),  \tag{10}$$

where $\{\alpha_t\}$ is a sequence of positive step-size and $\hat{\nabla}_1^K f(w_t, z_{i_t})$ denotes the 1-point gradient estimation of form equation 5. Under the Lipschitz continuous and smooth condition, ZO-SGD with 1-point gradient estimation has the following generalization bound.

**Theorem 1** (Nonconvex). *Assume that the loss $f(\cdot, z) \in [0, 1]$ is L-Lipschitz and $\beta$-smooth for all $z \in \mathcal{Z}$. Consider the update rule Eq.equation 10 with $T$ the total number of iterates, $\alpha_t \le C/t$ for some (fixed) $C > 0$ and for all $t \le T$. Then the generalization error of ZO-SGD with 1-point gradient estimation is bounded by*

$$\epsilon_{gen} \le \left(1 + (\beta C + CL/c)^{-1}\right) \frac{T e^{1+1/e}}{n^{1 - \frac{c}{d^2 L C \sqrt{n}}}}.$$

**ZO-SGD with 2-Point Gradient Estimation.** Then we consider the generalization error bound of ZO-SGD with 2-point gradient estimation. As formulated in Eq.equation 6, there are two general forms of 2-point gradient estimation: central difference and forward difference. At the $t$-th iteration, we uniformly randomly select a sample index $i_t$ from $[n]$, and then update parameter $w_t$ by:

$$w_{t+1} = w_t - \alpha_t \hat{\nabla}_2^K f(w_t, z_{i_t}),  \tag{11}$$

where $\{\alpha_t\}$ is a sequence of positive step-size and $\hat{\nabla}_2^K f(w_t, z_{i_t})$ denotes the 2-point gradient estimator of form equation 6. Under the same conditions as in ZO-SGD with 1-point gradient estimation, we found that both the forward difference version and central difference version update rule would derive the same expansive factor and tighter generalization error bound of the same order.

**Theorem 2** (Nonconvex). *Assume that the loss $f(\cdot, z) \in [0, 1]$ is L-Lipschitz and $\beta$-smooth for all $z \in \mathcal{Z}$. Consider the update rule Eq.equation 11 with $T$ the total number of iterates, $\alpha_t \le C/t$ for some (fixed) $C > 0$ and for all $t \le T$. Then the generalization error of ZO-SGD with 2-point gradient estimation is bounded by $\epsilon_{gen} \le \left(1 + (\beta C)^{-1}\right)^2 \left(1 + (2 + c)CL^2\right) 3Te/2n$.*

**ZO-SGD with Coordinate-Wise Gradient Estimation.** Then we consider the generalization errror bound of ZO-SGD with coordinate-wise gradient estimation. At the $t$-th iteration, zeroth-order stochastic gradient with coordinate-wise gradient estimation uniformly randomly select a sample index $i_t$ from $[n]$, and then update parameter $w_t$ by:

$$w_{t+1} = w_t - \alpha_t \hat{\nabla}^d f(w_t, z_{i_t})  \tag{12}$$

where $\{\alpha_t\}$ is a sequence of positive step-size and $\hat{\nabla}^d f(w_t, z_{i_t})$ denotes the coordinate-wise gradient estimation of form equation 7. We separate the generalization error bound of ZO-SGD with coordinate-wise gradient estimation into three different cases and get the following theorem.

**Theorem 3.** *Assume that the loss function $f(\cdot; z)$ is $\beta$-smooth and L-Lipschitz for every $z$. Let $T$ be the total number of iterates of ZO-SGD with coordinate-wise gradient estimation and for any $t \le T$,*

- *if each $f(\cdot, z)$ is convex, $\alpha_t \le \frac{2}{\beta}$ and choose $\mu \le \frac{c}{dn}$, then we have $\epsilon_{gen} \le \left(\frac{2L^2 + \beta cL}{n}\right) \sum_{t=1}^{T} \alpha_t$.*

- *if each $f(\cdot, z)$ is $\gamma$-strongly convex, $\alpha_t \le 1/\beta$ and choose $\mu \le \frac{c}{dn}$, then we have $\epsilon_{gen} \le \frac{2L^2 + \beta cL}{\gamma n}$.*

- *if the loss $f(\cdot, z) \in [0, 1]$ for all $z$, $\alpha_t \leq C/t$ for some (fixed) $C > 0$, then we have*

$$\epsilon_{gen} \leq \frac{1 + \frac{1}{\beta C}}{n} \left(2CL^2 + \beta CcL\right)^{\frac{1}{\beta C+1}} (eT)^{\frac{\beta C}{\beta C+1}}.$$

**Remark 1.** *Both 1-point and 2-point gradient estimations lead to a new expansive factor, which invalidates the contraction property induced by the expansive factor $\eta$ of first-order update rules under convex and strongly convex conditions. As a result, the generalization bound under convex and strongly convex conditions is of the same order as that under non-convex conditions. As a comparison, coordinate-wise estimate does not generate new expansive factors, and the $\sigma_{t,1}$ it brings is also independent of the dimension $d$.*

## 4.2 ZO-GD GENERALIZATION ANALYSIS

**ZO-GD with 1-point Gradient Estimation.** At the $t$-th iteration, zeroth-order gradient descent with 1-point gradient estimation updates parameter $w_t$ by:

$$w_{t+1} = w_t - \alpha_t \hat{\nabla}_K^1 R_D(w_t), \tag{13}$$

where $\hat{\nabla}_K^1 R_D(w_t)$ denotes the 1-point gradient estimator of form equation 5.

**Theorem 4** (Nonconvex). *Assume that the loss $f(\cdot, z) \in [0, 1]$ is L-Lipschitz and $\beta$-smooth for all $z \in \mathcal{Z}$. Consider the update rule Eq.equation 13 with $T$ the total number of iterates, $\alpha_t \leq C/t\big(\beta(\Gamma_d^k + 1) + LA_d/\mu\big)$ for some (fixed) $C > 0$ and for all $t \leq T$. Then the generalization error of ZO-GD with 1-point gradient estimation is bounded by $\epsilon_{gen} \leq \min\{C + 1, C\log(eT)\}(2L^2 + \beta + \beta L)(eT)^C/(n\beta)$.*

**ZO-GD with 2-point Gradient Estimation.** At the $t$-th iteration, zeroth-order gradient descent with 2-point gradient estimation updates parameter $w_t$ by:

$$w_{t+1} = w_t - \alpha_t \hat{\nabla}_K^2 R_D(w_t), \tag{14}$$

where $\hat{\nabla}_K^2 R_D(w_t)$ denotes the 2-point gradient estimator of form equation 6.

**Theorem 5** (Nonconvex). *Assume that the loss $f(\cdot, z) \in [0, 1]$ is L-Lipschitz and $\beta$-smooth for all $z \in \mathcal{Z}$. Consider the update rule Eq.equation 14 with $T$ the total number of iterates, $\alpha_t \leq C/\left(\Gamma_d^k + 1\right)t$ for some (fixed) $C > 0$ and for all $t \leq T$. Then the generalization error of ZO-GD with 2-point gradient estimation is bounded by*

$$\epsilon_{gen} \leq \frac{(2 + c)L^2}{n} (eT)^{C\beta} \min\{\frac{C\beta + 1}{\beta}, C\log(eT)\}.$$

**ZO-GD with Coordinate Gradient Estimation.** At the $t$-th iteration, zeroth-order gradient descent with coordinate-wise gradient estimation updates parameter $w_t$ by:

$$w_{t+1} = w_t - \alpha_t \hat{\nabla}^d R_D(w_t), \tag{15}$$

where $\hat{\nabla}^d R_D(w_t)$ denotes the coordinate-wise gradient estimator of form equation 7.

**Theorem 6.** *Assume that the loss function $f(\cdot; z)$ is $\beta$-smooth and L-Lipschitz for every $z$. Let $T$ be the total number of iterates of ZO-GD with coordinate-wise gradient estimation and for any $t \leq T$,*

- *if each $f(\cdot, z)$ is convex, $\alpha_t \leq \frac{2}{\beta}$ and choose $\mu \leq \frac{c}{dn}$, then we have $\epsilon_{gen} \leq \left(\frac{2L^2 + \beta cL}{n}\right) \sum_{t=1}^T \alpha_t$.*

- *if each $f(\cdot, z)$ is $\gamma$-strongly convex, $\alpha_t \leq \frac{1}{\beta}$ and choose $\mu \leq \frac{c}{dn}$, then we have $\epsilon_{\text{gen}} \leq \frac{2L^2 + \beta cL}{\gamma n}$.*

- *if the loss $f(\cdot, z) \in [0, 1]$ for all $z$, $\alpha_t \leq C/t$ for some (fixed) $C > 0$, then we have*

$$\epsilon_{gen} \leq \frac{(2 + c)L^2}{n} (eT)^{C\beta} \min\{\frac{C\beta + 1}{\beta}, C\log(eT)\}.$$

### 4.3 ZO-SVRG GENERALIZATION ANALYSIS

In this subsection, we will analyze the generalization bound of zeroth-order stochastic variance reduced gradient method. To the best of our knowledge, there are multiple versions of ZO-SVRG. In the paper, we consider the single-sample version of the ZO-SVRG. Specifically, in the $s$-th stage, $t$-th iteration, ZO-SVRG uniformly randomly selects a sample index $i_t^s$, and update parameter $w_t^s$ by

$$v_t^s = \hat{\nabla} f(w_t^s, z_{i_t^s}) - \hat{\nabla} f(\tilde{w}^s, z_{i_t^s}) + \hat{\nabla} R_D(\tilde{w}^s), \quad w_{t+1}^s = w_t^s - \alpha_t v_t^s, \tag{16}$$

where $\tilde{w}^s = w_m^{s-1}$ is the final iterate of stage $s - 1$, where $m$ is number of steps of each stage, and $\{\alpha_t\}$ is a sequence of positive step-size.

**ZO-SVRG with 1-point Gradient Estimation.** We first consider the generalization error bound of ZO-SVRG with 1-point gradient estimation. Let $\hat{\nabla} f(w_t^s, z_{i_t^s})$, $\hat{\nabla} f(\tilde{w}^s, z_{i_t^s})$ and $\hat{\nabla} R_D(\tilde{w}^s)$ be approximated Eq.equation 5, and update parameter $w_t^s$ through Eq. equation 16. Then, we have the following theorem.

**Theorem 7.** *Assume that the loss function $f(\cdot; z)$ is $\beta$-smooth, and L-Lipschitz for every z. Suppose that we run ZO-SVRG with step sizes $\alpha_t \leq C/\left(sm(\beta(\Gamma_d^k + 1) + \frac{LA_d}{\mu})\right)$ for T steps. After S stages, ZO-SVRG satisfies with $\epsilon_{gen} \leq (2L^2 + \beta + \beta L)S^{3C} \exp C/(C\beta n)$.*

**ZO-SVRG with 2-point Gradient Estimation.** Then we consider the generalization error bound of ZO-SVRG with 2-point gradient estimation. Let $\hat{\nabla} f(w_t^s, z_{i_t^s})$, $\hat{\nabla} f(\tilde{w}^s, z_{i_t^s})$ and $\hat{\nabla} R_D(\tilde{w}^s)$ be approximated Eq.equation 6, and update parameter $w_t^s$ through Eq. equation 16. Then, we have the following theorem.

**Theorem 8.** *Assume that the loss function $f(\cdot; z)$ is $\beta$-smooth, and L-Lipschitz for every z. Suppose that we run ZO-SVRG with step sizes $\alpha_t \leq C/sm\left(\Gamma_d^k + 1\right)$ for T steps. After S stages, ZO-SVRG satisfies $\epsilon_{gen} \leq (2L^2 + cL)S^{3\beta C} \exp C\beta/(\beta n)$.*

**ZO-SVRG with Coordinate-Wise Gradient Estimation.** Then we consider the generalization error bound of ZO-SVRG with coordinate-wise gradient estimation. Let $\hat{\nabla} f(w_t^s, z_{i_t^s})$, $\hat{\nabla} f(\tilde{w}^s, z_{i_t^s})$ and $\hat{\nabla} R_D(\tilde{w}^s)$ be approximated Eq.equation 7, and update parameter $w_t^s$ through Eq. equation 16. Then, we have the following theorem.

**Theorem 9.** *Assume that the loss function $f(\cdot; z)$ is $\beta$-smooth, and L-Lipschitz for every z. Let T be he total number of iterates of ZO-SVRG with coordinate-wise gradient estimation and for any $t \leq T$,*

- *Suppose we choose $\alpha_t \leq \frac{C}{sm}$ at state S. After S states, ZO-SVRG satisfies uniform stability with $\epsilon_{gen} \leq (2L^2 + cL)S^{3\beta C} \exp C\beta/(C\beta n)$.*

- *if $f(\cdot, z)$ is convex for all z, $\alpha_t \leq \frac{C}{sm}$ at stage S. After S stages, ZO-SVRG satisfies uniform stability with $\epsilon_{gen} \leq \min\{C + \beta^{-1}, c\log(eS)\}(6L^2 + 3cL)(eS)^{2\beta C}/n$.*

- *if $f(\cdot, z)$ is $\gamma$-strongly convex for all z, $\alpha_t \leq \frac{1}{\beta}$. After S stages, ZO-SVRG satisfies uniform stability with $\epsilon_{gen} \leq (6SL^2 + 3cS)/(n\gamma)$.*

**Remark 2.** *Utilizing full-batch updates with ZO-GD and ZO-SVRG, it's not possible to adopt the same step size as in ZO-SGD for first-order updates. Furthermore, under the condition of selecting a smaller step size, the generalization bound of ZO-GD, $O(T^{\beta C}/n)$, is looser compared to that of ZO-SGD, $O(\frac{T}{n})$.*

## 5 NUMERICAL EXPERIMENTS

In this section, we assess the generalization errors associated with optimizing nonconvex loss functions using ZO-GD, ZO-SGD, and ZO-SVRG. To primary goal is to verify the generalization errors of different zeroth-order optimization algorithms and different gradient estimators. To achieve this, we conduct experiments on two nonconvex models: nonconvex logistic regression and a two layer neural network.

## 5.1 Experimental Setups

**Nonconvex logistic regression:** Initially, we examine a logistic regression problem featuring a nonconvex regularization term. The loss function is formulated as:

$$f(x) = \frac{1}{n} \sum_{i=1}^{n} - \left( y_i \log \sigma(-a_i^T x) + (1 - y_i) \log \sigma(a_i^T x) \right) + \lambda \sum_{i=1}^{d} \frac{x_i^2}{1 + x_i^2},$$

where $a_i \in \mathbb{R}^d$ represents the sample, $y_i \in \{0, 1\}$ signifies the label, $\sigma(z) = \frac{1}{1+e^{-z}}$, and $\lambda$ is assigned a value of $0.5$. **Neural network:** Subsequently, we focus on a binary classification task using a two-layer neural network with relu activation function and binary cross entropy loss. **Dataset:** For both two nonconvex models, we utilize the LIBSVM's Australian dataset. We separate the dataset into two parts: 80% for training and 20% for test.

## 5.2 Comparison of Generalization Errors on Different ZO algorithms

In this part, we assess the generalization errors of various zeroth-order optimization algorithms using identical gradient estimators. Generalization errors are measured by the absolute difference between training and testing losses. Figure 1 depicts our results, with each subfigure comparing the performance of ZO-GD, ZO-SGD, and ZO-SVRG under a consistent gradient estimation method. The figures collectively indicate that generalization errors for all algorithms tend to rise as the number of iterations increases. Specifically, ZO-GD exhibits the most rapid increase in error, with ZO-SVRG following. As the algorithms approach convergence, ZO-SGD consistently demonstrates the smallest generalization error. These empirical findings corroborate our theoretical results presented in Table 1.

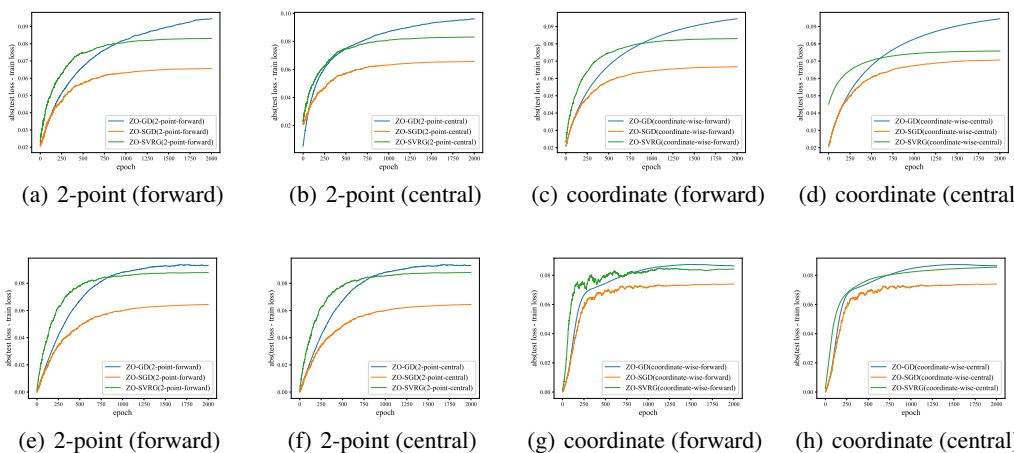

|  |  |  |  |
|---|---|---|---|
| (a) 2-point (forward) | (b) 2-point (central) | (c) coordinate (forward) | (d) coordinate (central) |
| (e) 2-point (forward) | (f) 2-point (central) | (g) coordinate (forward) | (h) coordinate (central) |

Figure 1: Comparison of generalization errors of different algorithms. (a)-(d) are results for nonconvex logistic regression and (e)-(h) are results for neural network

## 5.3 Comparison of Generalization Errors on Different ZO Gradient Estimators

Due to space limitation, we defer the detailed experimental results and discussions of this part to Appendix B.

## 6 Conclusion

In this paper, we introduce a generalized proof framework for zeroth-order optimization algorithms. Using this framework, we establish generalization bounds for ZO-SGD, ZO-GD, and ZO-SVRG with various estimators and convexity conditions. Our findings reveal that employing coordinate estimates in ZO algorithms outperforms other estimators in terms of generalization performance. Additionally, we demonstrate that ZO algorithms can achieve comparable generalization performance to first-order algorithms by leveraging coordinate estimates without reducing the learning rate.

## ACKNOWLEDGMENTS

This work was supported in part by the National Natural Science Foundation of China (Nos. 62376104 and 12071166), the Fundamental Research Funds for the Central Universities of China (No. 2662023LXPY005), and HZAU-AGIS Cooperation Fund (No. SZYJY2023010).

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

# A    NOTATIONS

The main notations of this paper are summarized in Table 2.

Table 2: Summary of main notations involved in this paper.

| Notations | Descriptions |
|---|---|
| ZO | Zeroth-order |
| SGD | Stochastic gradient descent |
| GD | Gradient descent |
| SVRG | Stochastic Variance Reduced Gradient |
| $z_{i_t}$ | the random sample |
| $D = \{z_1, ..., z_n\}$ | the dataset |
| $n$ | the numbers of samples |
| $w, \mathcal{W}$ | the parameter of training model and model parameter space, respectively |
| $d$ | the dimensions of $\mathcal{W}$, respectively |
| $f(w)$ | the loss function defined as $f(w; z_{i_t})$ |
| $\nabla f$ | the gradient of $f(w; z_{i_t})$ to the first argument $w$ |
| $\hat{\nabla} f$ | the gradient estimation of the true gradient $\nabla f$ |
| $G_t(\cdot), G'_t(\cdot)$ | the update rules of first-order methods under samples $S$ and $S'$, respectively |
| $\tilde{G}_t(\cdot), \tilde{G}'_t(\cdot)$ | the update rules of zeroth-order methods under samples $\mathbb{S}$ and $\mathbb{S}'$, respectively |
| $\mu$ | the smoothing parameter |
| $s$ | the stage for SVRG |
| $S$ | the whole number of stages |
| $m$ | the number of steps of each stage |
| $U$ | the random direction vector drawn from a certain distribution $\mathbb{D}$ |
| $K$ | the number of direction vector |
| $R, R_D$ | the population risk and empirical risk based on training dataset $D$, respectively |
| $T$ | the whole number of iterative steps |
| $w_t$ | the model parameter derived by SGD after $t$-th update |
| $\alpha_t$ | the step size at the $t$-th update |
| $\alpha$ | the step size |
| $A, A(D)$ | the given algorithm and its output model parameter based on training dataset $D$, respectively |
| $\gamma, L, \beta$ | the parameters of strong convexity, Lipschitz continuity and smoothness, respectively |
| $e$ | the base of the natural logarithm |

# B    COMPARISON OF GENERALIZATION ERRORS ON DIFFERENT ZO GRADIENT ESTIMATORS

In this section, we present supplementary experimental settings and evaluate the impact of different gradient estimators on generalization errors and report the absolute difference between the training and testing losses in Figure 2.

For all experiments, we set the maximum number of iterations to be 2000. The batch size of the stochastic gradient is set to be 50. The initial learning rate is set to 0.01. This rate is systematically decreased every $T$ iterations by a factor of $\gamma$. Both $T$ and $\gamma$ are optimally determined through a grid search process where $T$ and $\gamma$ are chosen from $\{30, 60, 100, 150, 200, 250\}$ and $\{0.6, 0.7, 0.8, 0.9\}$, respectively. The adjustment of the learning rate at regular intervals helps in fine-tuning the learning process, potentially leading to more effective and efficient convergence of the model. For 2-point gradient estimator, we also conduct a grid search for the parameter $K$ chosen from the set $\{2, 3, 4, 6, 8, 9, 12\}$.

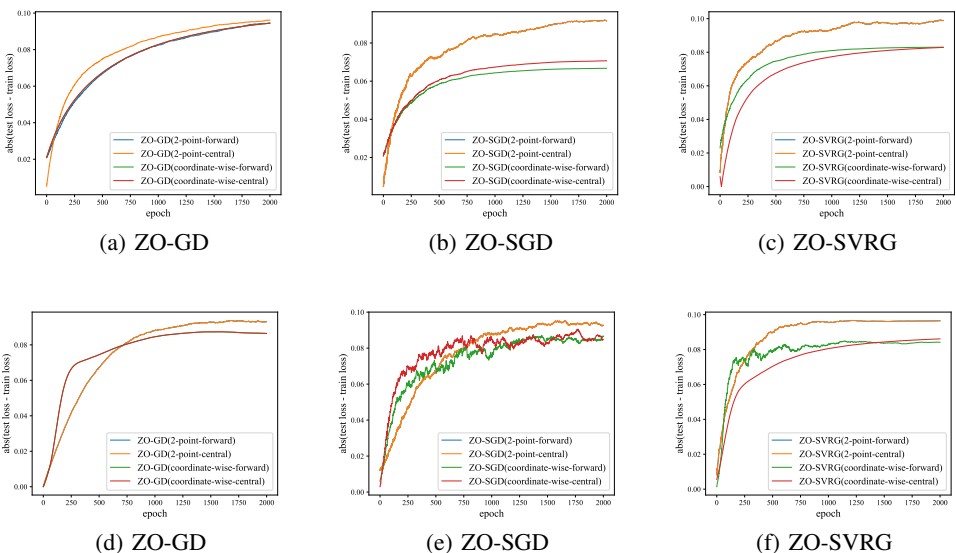

Figure 2: Comparison of generalization errors of different gradient estimators. (a)-(c) are results for nonconvex logistic regression and (d)-(f) are results for neural network

As illustrated in the figure, the coordinate-wise gradient estimator has smaller generalization errors than the 2-point gradient estimator. Additionally, both the forward and central difference versions of these estimators demonstrate comparable generalization errors.

## C    PREPARATION FOR PROOF

### C.1    SOME BASIC DEFINITIONS AND LEMMAS

**Definition 3** ($L$-Lipschitz). *We say that $f$ is $L$-Lipschitz if for all points $u$ in the domain of $f$ we have $\|\nabla f(x)\| \leq L$. This implies that*

$$|f(u) - f(v)| \leq L\|u - v\|.$$

**Definition 4** ($\beta$-Smooth). *A function $f : \Omega \to \mathbb{R}$ is $\beta$-smooth if for all for all $u, v \in \Omega$ we have*

$$\|\nabla f(u) - \nabla f(v)\| \leq \beta\|u - v\|.$$

**Definition 5** (Convex). *A function $f : \Omega \to \mathbb{R}$ is convex if for all $u, v \in \Omega$ we have*

$$f(u) \geq f(v) + \langle \nabla f(v), u - v \rangle.$$

**Definition 6** ($\gamma$-Strongly Convex). *A function $f : \Omega \to \mathbb{R}$ is $\gamma$-strongly convex if for all $u, v \in \Omega$ we have*

$$f(u) \geq f(v) + \langle \nabla f(v), u - v \rangle + \frac{\gamma}{2}\|u - v\|^2.$$

**Lemma 6** (Hardt et al. (2016), Lemma 3.7). *Assume that $f$ is $\beta$-smooth. Then, the following properties hold.*

*1. $G$ is $(1 + \alpha\beta)$-expansive.*

*2. Assume in addition that $f$ is convex. Then, for any $\alpha \leq 2/\beta$, the gradient update $G$ is 1-expansive.*

*3. Assume in addition that $f$ is $\gamma$-strongly convex. Then, for $\alpha \leq \frac{2}{\beta+\gamma}$, $G$ is $\left(1 - \frac{\alpha\beta\gamma}{\beta+\gamma}\right)$ expansive.*

# D   THE APPROXIMATION ERROR UNDER DIFFERENT ZO GRADIENT ESTIMATION

From Lemma 1, if we establish the bound for $\mathbb{E}_{\mathcal{A}}[f(\mathcal{A}(D), z) - f(\mathcal{A}(D'), z)]$, then we determine the generalized error bound for zeroth-order optimization algorithms. And we apply the lipchitz condition on $f(\cdot, z)$ to get

$$\mathbb{E}\left|f\left(w_T; z\right) - f\left(w'_T; z\right)\right| \leq L\mathbb{E}\|w_T - w'_T\|,$$

So, bounding the limit of $\mathbb{E}[\delta_T] = \mathbb{E}\|w_T - w'_T\|$ is the main task of this work. For zeroth-order optimization, $w_{t+1} = \tilde{G}_t(w_t) = w_t - \alpha_t \mathcal{G}_f(w_{t+1})$ then we have

$$
\begin{aligned}
&\mathbb{E}\|\tilde{G}_t\left(w_t\right) - \tilde{G}'_t\left(w'_t\right)\| = \mathbb{E}\|\tilde{G}_t(w_t) - \tilde{G}_t(w'_t)\| \\
=&\mathbb{E}\|G_t(w_t) - G_t(w'_t) + \alpha_t\left[\mathcal{G}(w_t, z_{i_t}) - \mathcal{G}_f(w_t, z_{i_t}) - (\mathcal{G}(w'_t, z_{i_t}) - \mathcal{G}_f(w'_t, z_{i_t}))\right]\| \\
\leq&\mathbb{E}\left\|G_t\left(w_t\right) - G_t(w'_t)\right\| + \alpha_t\mathbb{E}\left\|\mathcal{G}(w_t, z_{i_t}) - \mathcal{G}_f(w_t, z_{i_t}) - (\mathcal{G}(w'_t, z_{i_t}) - \mathcal{G}_f(w'_t, z_{i_t}))\right\|.
\end{aligned}
$$

$\mathbb{E}\left\|\mathcal{G}(w_t, z_{i_t}) - \mathcal{G}_f(w_t, z_{i_t}) - (\mathcal{G}(w'_t, z_{i_t}) - \mathcal{G}_f(w'_t, z_{i_t}))\right\|$ constitutes the primary part influenced by the estimator.

## D.1   1-POINT GRADIENT ESTIMATION

The formula for 1-point gradient estimation is as follows:

$$\hat{\nabla}f(w, z) = \frac{1}{K}\sum_{k=1}^{K}\frac{f\left(w + \mu u_k, z\right)}{\mu}u_k.$$

By applying the Taylor expansion there exist vectors $W_k^*$, $W_k^\star$ with $j^{\text{th}}$ coordinates in the intervals $\left(w^{(j)}, w^{(j)} + \mu U_k^{(j)}\right) \cup \left(w^{(j)} + \mu U_k^{(j)}, w^{(j)}\right)$, $\left(w'^{(j)}, w'^{(j)} + \mu U_k^{(j)}\right) \cup \left(w'^{(j)} + \mu U_k^{(j)}, w'^{(j)}\right)$, such that for any $w, w' \in \mathbb{R}^d$ we have

$$
\begin{aligned}
&\mathbb{E}\left\|\left(\nabla f(w, z) - \hat{\nabla}f(w, z)\right) - \left(\nabla f(w', z) - \hat{\nabla}f(w', z)\right)\right\| \\
\leq&\mathbb{E}\left\|\frac{1}{K}\sum_{k=1}^{K}\frac{<f\left(w, z\right) - f\left(w', z\right), u_k>}{\mu}\right\| \\
+&\mathbb{E}\left\|\left(\frac{1}{K}\sum_{k=1}^{K}\langle\nabla f\left(w, z\right) - \nabla f\left(w', z\right), u_k\rangle u_k - (\nabla f\left(w, z\right) - \nabla f\left(w', z\right))\right)\right\| \\
+&\mathbb{E}\left\|\frac{1}{K}\sum_{k=1}^{K}\left(\frac{\mu}{2}u_k^{\mathrm{T}}\nabla_w^2 f\left(W_k^*, z\right)u_k\right)u_k\right\| + \mathbb{E}\left\|\frac{1}{K}\sum_{k=1}^{K}\left(\frac{\mu}{2}u_k^{\mathrm{T}}\nabla_w^2 f\left(W_k^\star, z\right)u_k\right)u_k\right\| \\
\leq&\frac{1}{\mu K}\sum_{k=1}^{K}\mathbb{E}\left\|u_k\right\|\mathbb{E}\left\|f\left(w, z\right) - f\left(w', z\right)\right\| \\
+&\frac{1}{K}\sum_{k=1}^{K}\mathbb{E}\left[\|u_k\|^2\right]\mathbb{E}\left[\|\nabla f\left(w, z\right) - \nabla f\left(w', z\right)\|\right] + \frac{1}{K}\sum_{k=1}^{K}\mu\beta\mathbb{E}\left[\|u_k\|^3\right] \\
\leq&\begin{cases}\left(\frac{L}{\mu}\mathbb{E}\|u_1\| + \beta\mathbb{E}\left[\|u_1\|^2\right]\right)\|w - w'\| + \mu\beta\mathbb{E}\left[\|u_k\|^3\right], & \text{if } \tilde{G}_t(\cdot) = \tilde{G}'_t(\cdot), \\ \frac{L}{\mu}\mathbb{E}\|u_1\|\|w - w'\| + 2L\mathbb{E}\left[\|u_1\|^2\right] + \mu\beta\mathbb{E}\left[\|u_k\|^3\right], & \text{if } \tilde{G}_t(\cdot) \neq \tilde{G}'_t(\cdot).\end{cases}
\end{aligned}
$$

## D.2 2-POINT GRADIENT ESTIMATION

The formula for central difference is as follows:

$$\hat{\nabla} f(w) = \frac{1}{K} \sum_{k=1}^{K} \frac{f(w + \mu u_k, z) - f(w - \mu u_k, z)}{2\mu} u_k$$

By applying the Taylor expansion there exist vectors $W_k^*$ , $W_k^\star$, $W_k^\circledast$ and $W_k^\divideontimes$ with $j^{\text{th}}$ coordinates in the intervals $\left( w^{(j)}, w^{(j)} + \mu u_k^{(j)} \right) \cup \left( w^{(j)} + \mu u_k^{(j)}, w^{(j)} \right)$ , $\left( w^{(j)} - \mu u_k^{(j)}, w^{(j)}, \right) \cup \left( w^{(j)}, w^{(j)} - \mu u_k^{(j)} \right)$, $\left( w'^{(j)}, w'^{(j)} + \mu u_k^{(j)} \right) \cup \left( w'^{(j)} + \mu u_k^{(j)}, w'^{(j)} \right)$, $\left( w'^{(j)} - \mu u_k^{(j)}, w'^{(j)} \right) \cup \left( w'^{(j)}, w'^{(j)} - \mu u_k^{(j)} \right)$, such that for any $w, w' \in \mathbb{R}^d$ we have

$$
\begin{aligned}
&\mathbb{E} \left\| (\nabla f(w, z) - \hat{\nabla} f(w, z)) - (\nabla f(w', z) - \hat{\nabla} f(w', z)) \right\| \\
&\leq \left\| \left( \frac{1}{K} \sum_{k=1}^{K} \langle \nabla f(w, z) - \nabla f(w', z), u_k^t \rangle u_k^t - (\nabla f(w, z) - \nabla f(w', z)) \right) \right\| \\
&\quad + \alpha_t \left\| \frac{1}{K} \sum_{k=1}^{K} \left( \frac{\mu}{4} u_k^{\mathrm{T}} \nabla_w^2 f(W_k^*, z) u_k \right) u_k \right\| + \left\| \frac{1}{K} \sum_{k=1}^{K} \left( \frac{\mu}{4} u_k^{\mathrm{T}} \nabla_w^2 f(W_k^\star, z) u_k \right) u_k \right\| \\
&\quad + \left\| \frac{1}{K} \sum_{k=1}^{K} \left( \frac{\mu}{4} u_k^{\mathrm{T}} \nabla_w^2 f(W_k^\circledast, z) u_k \right) u_k \right\| + \left\| \frac{1}{K} \sum_{k=1}^{K} \left( \frac{\mu}{4} u_k^{\mathrm{T}} \nabla_w^2 f(W_k^\divideontimes, z) u_k \right) u_k \right\| \\
&\leq \frac{1}{K} \sum_{k=1}^{K} \mathbb{E} \left[ \|u_k\|^2 \right] \mathbb{E} \left[ \|\nabla f(w_t, z) - \nabla f(w'_t, z)\| \right] + \frac{1}{K} \sum_{k=1}^{K} \mu \beta \mathbb{E} \left[ \|u_k^t\|^3 \right] \\
&\leq \begin{cases} \beta \mathbb{E} \left[ \|u_1\|^2 \right] \|w - w'\| + \mu \beta \mathbb{E} \left[ \|u_1\|^3 \right], & \text{if } \tilde{G}_t(\cdot) = \tilde{G}'_t(\cdot), \\ 2L \mathbb{E} \left[ \|u_1\|^2 \right] + \mu \beta \mathbb{E} \left[ \|u_1\|^3 \right], & \text{if } \tilde{G}_t(\cdot) \neq \tilde{G}'_t(\cdot). \end{cases}
\end{aligned}
$$

## D.3 COORDINATE-WISE GRADIENT ESTIMATION

The formula for forward difference is as follows:

$$\hat{\nabla} f(w) = \sum_{i=1}^{d} \frac{f(w + \mu \mathbf{e}_i, z) - f(w, z)}{\mu} \mathbf{e}_i$$

By applying the Taylor expansion there exist vectors $W_i^*$ and $W_i^\dagger$ with $i^{\text{th}}$ coordinates in the intervals $(w, w + \mu e_i)$ and $(w', w' + \mu e_i)$, respectively, such that for any $w, w' \in \mathbb{R}^d$ we have

$$
\begin{aligned}
&\mathbb{E} \left\| (\nabla f(w, z) - \hat{\nabla} f(w, z)) - (\nabla f(w', z) - \hat{\nabla} f(w', z)) \right\| \\
&\leq \| \sum_{i=1}^{d} \langle \nabla f(w, z) - \nabla f(w', z), e_i \rangle e_i - (\nabla f(w, z) - \nabla f(w', z)) \| \\
&\quad + \| \sum_{i=1}^{d} \left( \frac{\mu}{2} e_i^{\mathrm{T}} \nabla^2 f(W_i^*, z) e_i \right) e_i \| + \| \sum_{i=1}^{d} \left( \frac{\mu}{2} e_i^{\mathrm{T}} \nabla^2 f\left( W_i^\dagger, z \right) e_i \right) e_i \|.
\end{aligned}
$$

For $\sum_{i=1}^{d} \langle \nabla f(w, z) - \nabla f(w', z), e_i \rangle e_i = \nabla f(w, z) - \nabla f(w', z)$, we have

$$\mathbb{E}\left\|\left(\nabla f(w,z)-\hat{\nabla}f(w,z)\right)-\left(\nabla f(w',z)-\hat{\nabla}f(w',z)\right)\right\|$$

$$\leq \|\sum_{i=1}^{d}\left(\frac{\mu}{2}e_i^{\mathrm{T}}\nabla^2 f\left(W_i^*,z\right)e_i\right)e_i\| + \|\sum_{i=1}^{d}\left(\frac{\mu}{2}e_i^{\mathrm{T}}\nabla^2 f\left(W_i^{\dagger},z\right)e_i\right)e_i\|$$

$$\leq d\beta\mu.$$

The formula for central difference is as follows:

$$\hat{\nabla}f(w) = \sum_{i=1}^{d}\frac{f\left(w+\mu\mathbf{e}_i,z\right)-f\left(w-\mu\mathbf{e}_i,z\right)}{\mu}\mathbf{e}_i$$

By applying the Taylor expansion there exist vectors $W_k^*$, $W_k^\star$, $W_k^\circledast$ and $W_k^{\circledast}$ with $j^{\mathrm{th}}$ coordinates in the intervals $(w, w+\mu e_i)$, $(w-\mu e_i, w)$, $(w', w'+\mu e_i)$, $(w'-\mu e_i, w')$ respectively, such that for any $w, w' \in \mathbb{R}^d$ we have

$$\mathbb{E}\left\|\left(\nabla f(w,z)-\hat{\nabla}f(w,z)\right)-\left(\nabla f(w',z)-\hat{\nabla}f(w',z)\right)\right\|$$

$$\leq \|\sum_{i=1}^{d}\langle\nabla f\left(w,z\right)-\nabla f\left(w',z\right),e_i\rangle e_i - \left(\nabla f\left(w,z\right)-\nabla f\left(w',z\right)\right)\|$$

$$+ \|\sum_{i=1}^{d}\left(\frac{\mu}{4}e_i^{\mathrm{T}}\nabla^2 f\left(W_i^*,z\right)e_i\right)e_i\| + \|\sum_{i=1}^{d}\left(\frac{\mu}{4}e_i^{\mathrm{T}}\nabla^2 f\left(W_i^\star,z\right)e_i\right)e_i\|$$

$$+ \|\sum_{i=1}^{d}\left(\frac{\mu}{4}e_i^{\mathrm{T}}\nabla^2 f\left(W_i^\circledast,z\right)e_i\right)e_i\| + \|\sum_{i=1}^{d}\left(\frac{\mu}{4}e_i^{\mathrm{T}}\nabla^2 f\left(W_i^{\circledast},z\right)e_i\right)e_i\|$$

.

For $\sum_{i=1}^{d}\langle\nabla f\left(w,z\right)-\nabla f\left(w',z\right),e_i\rangle e_i = \nabla f\left(w,z\right)-\nabla f\left(w',z\right)$, we have

$$\mathbb{E}\left\|\left(\nabla f(w,z)-\hat{\nabla}f(w,z)\right)-\left(\nabla f(w',z)-\hat{\nabla}f(w',z)\right)\right\|$$

$$\leq \|\sum_{i=1}^{d}\left(\frac{\mu}{4}e_i^{\mathrm{T}}\nabla^2 f\left(W_i^*,z\right)e_i\right)e_i\| + \|\sum_{i=1}^{d}\left(\frac{\mu}{4}e_i^{\mathrm{T}}\nabla^2 f\left(W_i^\star,z\right)e_i\right)e_i\|$$

$$+ \|\sum_{i=1}^{d}\left(\frac{\mu}{4}e_i^{\mathrm{T}}\nabla^2 f\left(W_i^\circledast,z\right)e_i\right)e_i\| + \|\sum_{i=1}^{d}\left(\frac{\mu}{4}e_i^{\mathrm{T}}\nabla^2 f\left(W_i^{\circledast},z\right)e_i\right)e_i\|$$

$$\leq d\beta\mu.$$

Comparing 1-point estimation, 2-point estimation with coordinate-wise estimation, the increment brought by the last one to the analysis process is $d\beta\mu$, which can be directly controlled by the smooth parameter $\mu$ and does not bring any associated increment of $\|w-w'\|$. It brings smaller approximation errors and does not bring new increments to the expansive factor, so that the generalization bound will be affected by the convexity of the function.

## E  PROOF OF GENERALIZATION ANALYSIS OF DIFFERENT DISTRIBUTION

The components about direction vector encompassed within the approximation error are defined as follows: $C_d \triangleq \mathbb{E}\left[\|u\|_2^3\right]$, $A_d \triangleq \mathbb{E}\left[\|u\|_2\right]$ and $\Gamma_K^d \triangleq \mathbb{E}\left[\left\|\frac{1}{K}\sum_{k=1}^{K}\langle\mathbf{V},u_k\rangle u_k - \mathbf{V}\right\| \mid \mathbf{V}\right]$, We can summarize the information in the following table.

Table 3: Summary of main distribution involved in this paper.

| Parameter | $\text{Unif}\left(\sqrt{d+2}\mathbb{B}^d\right)$ | $\text{Unif}\left(\sqrt{d}\mathbb{S}^{d-1}\right)$ | Standard normal |
|---|---|---|---|
| $\Gamma_d^K$ | $\sqrt{\frac{d+1}{K}}$ | $\sqrt{\frac{d-1}{K}}$ | $\sqrt{\frac{3d-1}{K}}$ |
| $A_d$ | $\frac{d}{d+1}\sqrt{d+2}$ | $\sqrt{d}$ | $\sqrt{d+3}$ |
| $C_d$ | $\frac{d}{d+3}\left(d+2\right)^{\frac{3}{2}}$ | $d^{\frac{3}{2}}$ | $(d+3)^{\frac{3}{2}}$ |

### E.1 PROOF OF UNIFORM DISTRIBUTION

**Lemma 7.** *Let $u_k \in \mathbb{R}^d, k \in \{1, 2 \ldots, K\}$ be i.i.d $\text{Unif}\left(\sqrt{d+2}\mathbb{B}^d\right)$ distribution. For every random vector $\mathbf{V} \in \mathbb{R}^d$ independent of all $u_k, k \in \{1, 2 \ldots, K\}$, it is true that*

$$\mathbb{E}\left[\left\|\frac{1}{K}\sum_{k=1}^{K}\langle\mathbf{V}, u_k\rangle u_k - \mathbf{V}\right\| \mid \mathbf{V}\right] \leq \sqrt{\frac{d^2 + 2d - 1}{K}}\|\mathbf{V}\|.$$

*Proof.* For fixed $\mathbf{V} \in \mathbb{R}^d$, we have due to independence

$$\mathbb{E}\left[\left\|\frac{1}{K}\sum_{k=1}^{K}\langle\mathbf{V}, u_k\rangle u_k - \mathbf{V}\right\|^2\right] = \frac{1}{K^2}\mathbb{E}\left[\left\|\sum_{k=1}^{K}\langle\mathbf{V}, u_k\rangle u_k - \mathbf{V}\right\|^2\right]$$

$$= \frac{1}{K^2}\sum_{k=1}^{K}\mathbb{E}\left[\left\|\langle\mathbf{V}, u_k\rangle u_k - \mathbf{V}\right\|^2\right]$$

$$= \frac{1}{K}\mathbb{E}\left[\left\|\langle\mathbf{V}, u_1\rangle u_1 - \mathbf{V}\right\|^2\right].$$

Now, again due to independence

$$\mathbb{E}\left[\left\|\langle\mathbf{V}, u_1\rangle u_1 - \mathbf{V}\right\|^2\right] = \mathbb{E}\left[\left\|\langle\mathbf{V}, u_1\rangle u_1\right\|^2 - 2\langle\langle\mathbf{V}, u_1\rangle u_1, \mathbf{V}\rangle + \|\mathbf{V}\|^2\right]$$

$$= \mathbb{E}\left[(\langle\mathbf{V}, u_1\rangle)^2\|u_1\|^2\right] - 2\mathbb{E}\left[\langle\mathbf{V}, u_1\rangle\langle u_1, \mathbf{V}\rangle\right] + \|\mathbf{V}\|^2$$

$$= \mathbf{V}^T\mathbb{E}\left[u_1 u_1^T\|u_1\|^2\right]\mathbf{V} - 2\mathbf{V}^T\mathbb{E}\left[u_1 u_1^T\right]\mathbf{V} + \|\mathbf{V}\|^2$$

$$\leq (d+2)\mathbf{V}^T\mathbf{V} - 2\|\mathbf{V}\|^2 + \|\mathbf{V}\|^2$$

$$= (d+1)\|\mathbf{V}\|^2.$$

Therefore,

$$\mathbb{E}\left[\left\|\frac{1}{K}\sum_{k=1}^{K}\langle\mathbf{V}, u_k\rangle u_k - \mathbf{V}\right\|^2\right] \leq \frac{(d+1)\|\mathbf{V}\|^2}{K}.$$

Thus, if $\mathbf{V}$ is random and independent of all $u_k$'s, it follows that

$$\mathbb{E}\left[\left\|\frac{1}{K}\sum_{k=1}^{K}\langle\mathbf{V}, u_k\rangle u_k - \mathbf{V}\right\| \mathbf{V}\right] \leq \sqrt{\mathbb{E}\left[\left\|\frac{1}{K}\sum_{k=1}^{K}\langle\mathbf{V}, u_k\rangle u_k - \mathbf{V}\right\|^2 \mid \mathbf{V}\right]}$$

$$\leq \sqrt{\frac{d+1}{K}\|\mathbf{V}\|^2}$$

$$= \sqrt{\frac{d+1}{K}} \|\mathbf{V}\|,$$

and our claim is proved. $\qquad \square$

**Lemma 8.** *Let* $u_k \in \mathbb{R}^d, k \in \{1, 2 \ldots, K\}$ *be i.i.d* $Unif\left(\sqrt{d}\mathbb{S}^{d-1}\right)$ *distribution. For every random vector* $\mathbf{V} \in \mathbb{R}^d$ *independent of all* $u_k, k \in \{1, 2 \ldots, K\}$, *it is true that*

$$\mathbb{E}\left[\left\|\frac{1}{K}\sum_{k=1}^{K}\langle\mathbf{V}, u_k\rangle u_k - \mathbf{V}\right\| \mid \mathbf{V}\right] \leq \sqrt{\frac{d^2-1}{K}}\|\mathbf{V}\|.$$

*Proof.* For fixed $\mathbf{V} \in \mathbb{R}^d$, we have due to independence

$$\mathbb{E}\left[\left\|\frac{1}{K}\sum_{k=1}^{K}\langle\mathbf{V}, u_k\rangle u_k - \mathbf{V}\right\|^2\right] = \frac{1}{K^2}\mathbb{E}\left[\left\|\sum_{k=1}^{K}\langle\mathbf{V}, u_k\rangle u_k - \mathbf{V}\right\|^2\right]$$

$$= \frac{1}{K^2}\sum_{k=1}^{K}\mathbb{E}\left[\left\|\langle\mathbf{V}, u_k\rangle u_k - \mathbf{V}\right\|^2\right]$$

$$= \frac{1}{K}\mathbb{E}\left[\left\|\langle\mathbf{V}, u_1\rangle u_1 - \mathbf{V}\right\|^2\right].$$

Now, again due to independence

$$\mathbb{E}\left[\left\|\langle\mathbf{V}, u_1\rangle u_1 - \mathbf{V}\right\|^2\right] = \mathbb{E}\left[\left\|\langle\mathbf{V}, u_1\rangle u_1\right\|^2 - 2\langle\langle\mathbf{V}, u_1\rangle u_1, \mathbf{V}\rangle + \|\mathbf{V}\|^2\right]$$

$$= \mathbb{E}\left[\left(\langle\mathbf{V}, u_1\rangle\right)^2\|u_1\|^2\right] - 2\mathbb{E}\left[\langle\mathbf{V}, u_1\rangle\langle u_1, \mathbf{V}\rangle\right] + \|\mathbf{V}\|^2$$

$$= \mathbf{V}^T\mathbb{E}\left[u_1 u_1^T\|u_1\|^2\right]\mathbf{V} - 2\mathbf{V}^T\mathbb{E}\left[u_1 u_1^T\right]\mathbf{V} + \|\mathbf{V}\|^2$$

$$= d\mathbf{V}^T\mathbf{V} - \|\mathbf{V}\|^2$$

$$= (d-1)\|\mathbf{V}\|^2$$

Therefore,

$$\mathbb{E}\left[\left\|\frac{1}{K}\sum_{k=1}^{K}\langle\mathbf{V}, u_k\rangle u_k - \mathbf{V}\right\|^2\right] = \frac{(d-1)\|\mathbf{V}\|^2}{K}.$$

Thus, if $\mathbf{V}$ is random and independent of all $u_k$'s, it follows that

$$\mathbb{E}\left[\left\|\frac{1}{K}\sum_{k=1}^{K}\langle\mathbf{V}, u_k\rangle u_k - \mathbf{V}\right\| \mid \mathbf{V}\right] \leq \sqrt{\mathbb{E}\left[\left\|\frac{1}{K}\sum_{k=1}^{K}\langle\mathbf{V}, u_k\rangle u_k - \mathbf{V}\right\|^2 \mid \mathbf{V}\right]}$$

$$\leq \sqrt{\frac{d-1}{K}\|\mathbf{V}\|^2}$$

$$= \sqrt{\frac{d-1}{K}}\|\mathbf{V}\|,$$

and our claim is proved. $\qquad \square$

**Lemma 9.** *For samples* $u$ *from the* $\ell_2$-*sphere, it is clear that* $\|u\|_2 = \sqrt{d}$ *and* $\|u\|_2^3 = d^{\frac{3}{2}}$. *When* $U \sim Uniform\left(\mathbb{B}^d\right)$, *the density* $p(t)$ *of* $\|Z\|_2$ *is given by* $d \cdot t^{d-1}$; *consequently, for any* $k > -d$ *we have*

$$\mathbb{E}\left[\|u\|_2^k\right] = \int_0^1 t^k p(t)dt = d\int_0^1 t^{d+k-1}dt = \frac{d}{d+k}.$$

*Thus for $Z \sim Uniform\left(\sqrt{d+2}\mathbb{B}^d\right)$ we have $\mathbb{E}\left[\|Z\|_2^k\right] = (d+2)^{k/2}d/(d+k)$.*

## F   PROOF OF ANALYSIS FRAMEWORK

***Proof of Lemma 3.*** Let $D$, $D'$ be two samples of size $n$ that differ in one sample, and let $\tilde{G}_t(\cdot), \tilde{G}'_t(\cdot)$ be two update rules of zeroth-order algorithms based on samples $D, D'$ respectively. Under the event $\mathcal{E}_t := \{\tilde{G}_t(\cdot) \equiv \tilde{G}'_t(\cdot)\}$, we have

$$\begin{aligned}
\delta_{t+1} =& \mathbb{E}\|\tilde{G}_t(w_t) - \tilde{G}'_t(w'_t)\| = \mathbb{E}\|\tilde{G}_t(w_t) - \tilde{G}_t(w'_t)\| \\
=& \mathbb{E}\|G_t(w_t) - G_t(w'_t) + \alpha_t\left[\mathcal{G}(w_t, z_{i_t}) - \mathcal{G}_f(w_t, z_{i_t}) - (\mathcal{G}(w'_t, z_{i_t}) - \mathcal{G}_f(w'_t, z_{i_t}))\right]\| \\
\leq& \|G_t(w_t) - G_t(w'_t)\| + \alpha_t\mathbb{E}\|\mathcal{G}(w_t, z_{i_t}) - \mathcal{G}_f(w_t, z_{i_t}) - (\mathcal{G}(w'_t, z_{i_t}) - \mathcal{G}_f(w'_t, z_{i_t}))\| \\
\leq& \eta\|w_t - w'_t\| + \alpha_t\mathbb{E}\|\mathcal{G}(w_t, z_{i_t}) - \mathcal{G}_f(w_t, z_{i_t}) - (\mathcal{G}(w'_t, z_{i_t}) - \mathcal{G}_f(w'_t, z_{i_t}))\| \\
\leq& \eta\|w_t - w'_t\| + \alpha_t\sigma_{t,1}.
\end{aligned}$$

The Under the event $\mathcal{E}_t^c := \{\tilde{G}_t(\cdot) \neq \tilde{G}'_t(\cdot)\}$, we have

$$\begin{aligned}
&\mathbb{E}\|\tilde{G}_t(w_t) - \tilde{G}'_t(w'_t)\| \\
=& \mathbb{E}\|G_t(w_t) - G'_t(w'_t) + \alpha_t\left[\mathcal{G}(w_t, z_{i_t}) - \mathcal{G}_f(w_t, z_{i_t}) - \left(\mathcal{G}(w'_t, z'_{i_t}) - \mathcal{G}_f(w'_t, z'_{i_t})\right)\right]\| \\
\leq& \|G_t(w_t) - G_t(w'_t)\| + \alpha_t\mathbb{E}\left\|\mathcal{G}(w_t, z_{i_t}) - \mathcal{G}_f(w_t, z_{i_t}) - \left(\mathcal{G}(w'_t, z'_{i_t}) - \mathcal{G}_f(w'_t, z'_{i_t})\right)\right\| \\
\leq& \min\{\eta, 1\}\|w_t - w'_t\| + 2\sigma_t + \alpha_t\mathbb{E}\left\|\mathcal{G}(w_t, z_{i_t}) - \mathcal{G}_f(w_t, z_{i_t}) - \left(\mathcal{G}(w'_t, z'_{i_t}) - \mathcal{G}_f(w'_t, z'_{i_t})\right)\right\| \\
\leq& \|w_t - w'_t\| + 2\sigma_t + \alpha_t\mathbb{E}\left\|\mathcal{G}(w_t, z_{i_t}) - \mathcal{G}_f(w_t, z_{i_t}) - \left(\mathcal{G}(w'_t, z'_{i_t}) - \mathcal{G}_f(w'_t, z'_{i_t})\right)\right\| \\
\leq& \|w_t - w'_t\| + 2\alpha_t\sigma_{t,2} + \alpha_t\sigma_{t,3}.
\end{aligned}$$

$\square$

***Proof of Lemma 4.*** Consider the events $\mathcal{E}_t \triangleq \left\{\tilde{G}_t(\cdot) \equiv \tilde{G}'_t(\cdot)\right\}$ and $\mathcal{E}_t^c \triangleq \left\{\tilde{G}_t(\cdot) \neq \tilde{G}'_t(\cdot)\right\}$ (see Eq. (4)). Recall that $\mathbb{P}(\mathcal{E}_t) = 1 - 1/n$ and $\mathbb{P}(\mathcal{E}_t^c) = 1/n$ for all $t \leq T$. For any $t_0 \geq 0$, we have

$$\begin{aligned}
\mathbb{E}[\delta_{t+1}|\mathcal{E}_{\delta_{t_0}}] =& \mathbb{P}(\mathcal{E}_t)\mathbb{E}[\delta_{t+1}|\mathcal{E}_{\delta_t}, \mathcal{E}_{\delta_{t_0}}] + \mathbb{P}(\mathcal{E}_t^c)\mathbb{E}[\delta_{t+1}|\mathcal{E}_{\delta_t}^c, \mathcal{E}_{\delta_{t_0}}] \\
=& (1 - \frac{1}{n})\mathbb{E}[\delta_{t+1}|\mathcal{E}_{\delta_t}, \mathcal{E}_{\delta_{t_0}}] + \frac{1}{n}\mathbb{E}[\delta_{t+1}|\mathcal{E}_{\delta_t}^c, \mathcal{E}_{\delta_{t_0}}] \\
\leq& (\eta + \frac{1}{n}(1 - \eta))\mathbb{E}[\delta_t|\mathcal{E}_{\delta_{t_0}}] + \alpha_t\left[(1 - \frac{1}{n})\sigma_{t,1} + \frac{1}{n}(2\sigma_{t,2} + \sigma_{t,3})\right].
\end{aligned}$$

Then we have

$$\mathbb{E}[\delta_T] \leq \sum_{t=t_0+1}^{T}\left(\eta + \frac{1}{n}(1 - \eta)\right)^{T-t}\alpha_t\left[(1 - \frac{1}{n})\sigma_{t,1} + \frac{1}{n}(2\sigma_{t,2} + \sigma_{t,3})\right]$$

$\square$

## G   PROOF OF GENERALIZATION ANALYSIS OF ZO-SGD

### G.1   ZO-SGD WITH 1-POINT GRADIENT ESTIMATION

**Lemma 10** (Growth Recursion). *Assume that for all $z$, $f(\cdot, z)$ is $L$-Lipschitz continuous, $\beta$-smooth, and $f(\cdot, z) \in [0, 1]$. Consider the sequences of updates $\left\{\tilde{G}_t\right\}_{t=1}^T$ and $\left\{\tilde{G}'_t\right\}_{t=1}^T$. Let $w_0 = w'_0$ be*

*the starting point, $w_{t+1} = \tilde{G}_t(w_t)$ and $w'_{t+1} = \tilde{G}'_t(w'_t)$ for any $t \in \{1, \dots, T\}$. Then for any $w_t, w'_t \in \mathbb{R}^d$ and $t \geq 0$ the following recursion holds:*

$$\mathbb{E}\left[\left\|\tilde{G}_t(w_t) - \tilde{G}'_t(w'_t)\right\|\right] \leq \begin{cases} \left(\eta_1 + \alpha_t \beta \left(\Gamma_d^k + 1\right) + \frac{\alpha_t L A_d}{\mu}\right)\|w_t - w'_t\| + \mu\beta\alpha_t C_d, & \text{if } \tilde{G}_t(\cdot) = \tilde{G}'_t(\cdot), \\ \|w_t - w'_t\| + \frac{\alpha_t A_d}{\mu} + 2\alpha_t L(\Gamma_d^k + 1) + \mu\beta\alpha_t C_d, & \text{if } \tilde{G}_t(\cdot) \neq \tilde{G}'_t(\cdot). \end{cases}$$

*Proof of Lemma 10.* The formula for 1-point gradient estimation is as follows:

$$\hat{\nabla}f(w) = \frac{1}{K}\sum_{k=1}^{K}\frac{f(w + \mu u_k)}{\mu}u_k.$$

Let $S$ and $S'$ be two samples of size $n$ differing in only a single example, and let $\tilde{G}_t(\cdot), \tilde{G}'_t(\cdot)$ be the update rules of the ZO-SGD for each of the sequences $S, S'$ respectively. By applying the Taylor expansion there exist vectors $W^*_{k,t}, W^\star_{k,t}$ with $j^{\text{th}}$ coordinates in the intervals $\left(w_t^{(j)}, w_t^{(j)} + \mu U_{k,t}^{(j)}\right) \cup \left(w_t^{(j)} + \mu U_{k,t}^{(j)}, w_t^{(j)}\right)$, $\left(w_t'^{(j)}, w_t'^{(j)} + \mu U_{k,t}^{(j)}\right) \cup \left(w_t'^{(j)} + \mu U_{k,t}^{(j)}, w_t'^{(j)}\right)$, such that for any $w_t, w'_t \in \mathbb{R}^d$ we have

$$\mathbb{E}\left[\left\|\tilde{G}_t(w_t) - \tilde{G}_t(w'_t)\right\|\right]$$

$$\leq \|G(w_t) - G'(w'_t)\| + \alpha_t\left\|(\nabla f(w_t) - \hat{\nabla}f(w_t)) - (\nabla f(w'_t) - \hat{\nabla}f(w'_t))\right\|$$

$$\leq \|G(w_t) - G'(w'_t)\| + \alpha_t\left\|\frac{1}{K}\sum_{k=1}^{K}\frac{<f(w_t) - f(w'_t), u_k^t>}{\mu}\right\|$$

$$+\alpha_t\left\|\left(\frac{1}{K}\sum_{k=1}^{K}\left\langle\nabla f(w_t) - \nabla f(w'_t), u_k^t\right\rangle u_k^t - (\nabla f(w_t) - \nabla f(w'_t))\right)\right\|$$

$$+\alpha_t\left\|\frac{1}{K}\sum_{k=1}^{K}\left(\frac{\mu}{2}u_k^{\mathrm{T}}\nabla_w^2 f\left(W^*_{k,t}\right)u_k^t\right)u_k^t\right\| + \alpha_t\left\|\frac{1}{K}\sum_{k=1}^{K}\left(\frac{\mu}{2}u_k^{\mathrm{T}}\nabla_w^2 f\left(W^\star_{k,t}\right)u_k^t\right)u_k^t\right\|$$

$$\leq \underbrace{\|G(w_t) - G'(w'_t)\|}_{A} + \underbrace{\alpha_t\Gamma_d^k\mathbb{E}\left[\|\nabla f(w_t) - \nabla f(w'_t)\|\right] + \alpha_t\frac{1}{\mu}\mathbb{E}\left\|u_k^t\right\|\mathbb{E}\|\nabla f(w_t) - \nabla f(w'_t)\|}_{B}$$

$$+\underbrace{\frac{\alpha_t}{K}\sum_{k=1}^{K}\mu\beta\mathbb{E}\left[\left\|u_k^t\right\|^3\right]}_{E}$$

Part A corresponds to the first-order stochastic gradient descent (SGD), which decomposes into $\eta$ and $\sigma_{t,2}$ terms. Part B corresponds to the scaling term introduced by the gradient approximation, which leads to inflation factor $\eta'$ and bound factor. In contrast to the 2-point estimate, one-point estimate has an additional term in Part B, which generates different expansive and bound factors. It should be noted that the term $\mu$ appears in the denominator, so its value needs to be carefully chosen and not too small. Part E corresponds to a fixed value that can be decomposed for both one-point and two-point estimates, and can only be constrained by $\beta$-smoothness, finally added to the bound factor. Under the conditions of $L$-Lipschitz continuous, $\beta$-smooth, $f(\cdot, z) \in [0, 1]$, $\mathbb{E}\left[\|U\|^3\right] \leq C_d$, we get $\eta' = \alpha_t\beta\Gamma_d^k + \frac{\alpha_t L A_d}{\mu}$, $\sigma_{t,1} = \mu\beta\alpha_t C_d$, $\sigma_{t,2} = L$, $\sigma_{t,3} = \frac{\alpha_t A_d}{\mu} + 2\alpha_t L\Gamma_d^k + \mu\beta\alpha C_d$.

$\square$

**Lemma 11.** *Assume that the loss function $f(\cdot, z)$ is L-Lipschitz and $\beta$-smooth for all $z \in \mathcal{Z}$. Consider the ZO-SGD algorithm with final-iterate estimates $W_T$ and $W'_T$, corresponding to the data-sets $S, S'$, respectively (that differ in exactly one entry). Then the discrepancy $\delta_T \triangleq \|W_T - W'_T\|$, under the event $\mathcal{E}_{\delta_{t_0}}$, satisfies the inequality*

$$\mathbb{E}\left[\delta_T \mid \mathcal{E}_{\delta_{t_0}}\right]$$

$$\leq \left(\frac{2L}{n}(\Gamma_d^k + 1) + \frac{\alpha_t L A_d}{\mu n} + \mu\beta C_d\right) \sum_{t=t_0+1}^{T} \alpha_t \prod_{j=t+1}^{T} \left(1 + \beta\alpha_j(\Gamma_d^k + 1) + \frac{\alpha_j L A_d}{\mu}\right)$$

*Proof of Lemma 11.* Consider the events $\mathcal{E}_t \triangleq \left\{\tilde{G}_t(\cdot) \equiv \tilde{G}'_t(\cdot)\right\}$ and $\mathcal{E}_t^c \triangleq \left\{\tilde{G}_t(\cdot) \neq \tilde{G}'_t(\cdot)\right\}$ (see Eq. (4)). Recall that $\mathbb{P}\left(\mathcal{E}_t\right) = 1 - 1/n$ and $\mathbb{P}\left(\mathcal{E}_t^c\right) = 1/n$ for all $t \leq T$. For any $t_0 \geq 0$, a direct application of Lemma 3 gives

$$\mathbb{E}\left[\delta_{t+1} \mid \mathcal{E}_{\delta_{t_0}}\right]$$
$$=\mathbb{P}\left(\mathcal{E}_t\right)\mathbb{E}\left[\delta_{t+1} \mid \mathcal{E}_t, \mathcal{E}_{\delta_{t_0}}\right] + \mathbb{P}\left(\mathcal{E}_t^c\right)\mathbb{E}\left[\delta_{t+1} \mid \mathcal{E}_t^c, \mathcal{E}_{\delta_{t_0}}\right]$$
$$=\left(1 - \frac{1}{n}\right)\mathbb{E}\left[\delta_{t+1} \mid \mathcal{E}_t, \mathcal{E}_{\delta_{t_0}}\right] + \frac{1}{n}\mathbb{E}\left[\delta_{t+1} \mid \mathcal{E}_t^c, \mathcal{E}_{\delta_{t_0}}\right]$$
$$\leq\left(1 + \beta\alpha_j(\Gamma_d^K + 1)\left(1 - \frac{1}{n}\right) + \frac{\alpha_t L A_d}{\mu}\right)\mathbb{E}\left[\delta_t \mid \mathcal{E}_{\delta_{t_0}}\right] + \frac{2\alpha_t L}{n}(\Gamma_d^K + 1) + \frac{\alpha L A_d}{\mu n} + \mu\beta\alpha_t C_d.$$

then by solving this recursion we have

$$\mathbb{E}\left[\delta_T \mid \mathcal{E}_{\delta_{t_0}}\right] \leq \left(\frac{2L}{n}(\Gamma_d^K + 1) + \mu\beta C_d\right)\sum_{t=t_0+1}^{T}\alpha_t\prod_{j=t+1}^{T}\left(1 + \beta\alpha_j(\Gamma_d^K + 1)\left(1 - \frac{1}{n}\right) + \frac{\alpha_j L A_d}{\mu}\right).$$

$\square$

*Proof of Theorem 1.* Recall that $\eta = 1 + \beta\alpha_t$ for general (nonconvex) losses . Assuming that $\alpha_t \leq C/t$ for all $t \leq T$, we have

$$\mathbb{E}\left[\delta_T \mid \mathcal{E}_{\delta_{t_0}}\right]$$
$$\leq\left(\frac{2L}{n}(\Gamma_d^K + 1) + \mu\beta C_d + \frac{A_d}{\mu n}\right)\sum_{t=t_0+1}^{T}\alpha_t\prod_{j=t+1}^{T}\left(1 + \alpha_j\beta(\Gamma_d^K + 1) + \frac{\alpha_j L A_d}{\mu}\right)$$
$$\leq C\left(\frac{2L}{n}(\Gamma_d^K + 1) + \mu\beta C_d + \frac{A_d}{\mu n}\right)\sum_{t=t_0+1}^{T}\frac{1}{t}\prod_{j=t+1}^{T}\left(1 + \frac{C\beta(\Gamma_d^K + 1) + \frac{CLA_d}{\mu}}{j}\right)$$
$$\leq C\left(\frac{2L}{n}(\Gamma_d^K + 1) + \mu\beta C_d + \frac{A_d}{\mu n}\right)\sum_{t=t_0+1}^{T}\frac{1}{t}\prod_{j=t+1}^{T}\exp\left(\frac{C\beta(\Gamma_d^K + 1) + \frac{CLA_d}{\mu}}{j}\right)$$
$$\leq C(eT)^{C\beta(\Gamma_d^K + 1) + \frac{CLA_d}{\mu}}\left(\frac{2L}{n}\Gamma_d^K + \mu\beta C_d + \frac{A_d}{\mu n}\right)\sum_{t=t_0+1}^{T}\frac{1}{t}\frac{1}{(t+1)^{C\beta(\Gamma_d^K + 1) + \frac{CLA_d}{\mu}}}$$
$$\leq\underbrace{\left(\beta(\Gamma_d^K + 1) + \frac{LA_d}{\mu}\right)^{-1}\left(\frac{2L}{n}\Gamma_d^K + \mu\beta C_d + \frac{A_d}{\mu n}\right)}_{D}\left(\left(\frac{eT}{t_0}\right)^{C\beta(\Gamma_d^K + 1) + \frac{CLA_d}{\mu}} - e^{C\beta(\Gamma_d^K + 1) + \frac{CLA_d}{\mu}}\right)$$

We define $q \triangleq C\beta(\Gamma_d^k + 1) + \frac{CA_d}{\mu}$ and find the value of $t_0$ that minimizes the right part of

$$\mathbb{E}\left[|f\left(W_T, z\right) - f\left(W'_T, z\right)|\right] \leq \frac{t_0}{n}\sup_{w,z}f(w, z) + L\mathbb{E}\left[\delta_T \mid \mathcal{E}_{\delta_{t_0}}\right] \leq \frac{t_0}{n} + LD\left(\left(\frac{eT}{t_0}\right)^q - e^q\right),$$

which is $t_0^* = \min\left\{(qnLD)^{1/(q+1)}(eT)^{q/(q+1)}, T\right\}$. Then we give

$$\mathbb{E}\left[|f\left(W_T, z\right) - f\left(W'_T, z\right)|\right]$$
$$\leq \max\left\{\frac{(qnLD)^{\frac{1}{q+1}}(eT)^{\frac{q}{q+1}}}{n}, \frac{1 + 1/q}{n}(qnLD)^{\frac{1}{q+1}}(eT)^{\frac{q}{q+1}} - LDe^q\right\}$$
$$\leq \frac{1 + \frac{1}{C\beta(\Gamma_d^k + 1) + \frac{CLA_d}{\mu}}}{n}\left(2L^2(\Gamma_d^k + 1) + \mu\beta C_d nL + \frac{A_d L}{\mu}\right)^{\frac{1}{C\beta(\Gamma_d^k + 1) + \frac{CLA_d}{\mu} + 1}}(eT)^{\frac{C\beta(\Gamma_d^k + 1) + \frac{CLA_d}{\mu}}{C\beta(\Gamma_d^k + 1) + \frac{CLA_d}{\mu} + 1}}$$

$$\leq \frac{1 + \frac{1}{C\beta(\Gamma_d^k+1)+\frac{CLA_d}{\mu}}}{n} n^{\frac{1}{C\beta(\Gamma_d^k+1)+\frac{CLA_d}{\mu}+1}} \left(2L^2(\Gamma_d^k+1) + C\beta L + \frac{A_d L}{\mu\sqrt{n}}\right)^{\frac{1}{C\beta(\Gamma_d^k+1)+\frac{CLA_d}{\mu}+1}} (eT)^{\frac{C\beta(\Gamma_d^k+1)+\frac{CLA_d}{\mu}}{C\beta(\Gamma_d^k+1)+\frac{CLA_d}{\mu}+1}}$$

$$\leq \left(1 + \left(\beta C + \frac{CL}{c}\right)^{-1}\right) \frac{1}{n^{1-\frac{c}{d^2 LC\sqrt{n}}}} \left(C\beta(\Gamma_d^k+1) + \frac{CLA_d}{\mu} + 1\right)^{\frac{1}{C\beta(\Gamma_d^k+1)+\frac{CLA_d}{\mu}+1}} (eT)^{\frac{C\beta(\Gamma_d^k+1)+\frac{CLA_d}{\mu}}{C\beta(\Gamma_d^k+1)+\frac{CLA_d}{\mu}+1}}$$

$$\leq \left(1 + \left(\beta C + \frac{CL}{c}\right)^{-1}\right) \frac{Te^{1+1/e}}{n^{1-\frac{c}{d^2 LC\sqrt{n}}}}$$

$\square$

**Theorem 10** (convex). *Assume that the loss $f(\cdot, z) \in [0,1]$ is L-Lipschitz, $\beta$-smooth and convex for all $z \in \mathcal{Z}$. Consider the update rule Eq.equation 10 with $T$ the total number of iterates, $\alpha_t \leq C/t$ for some (fixed) $C > 0$ and for all $t \leq T$. Then the generalization error of ZO-SGD with 2-point gradient estimation is bounded by*

$$\epsilon_{gen} \leq \left(1 + (\beta C)^{-1}\right)^2 \left(1 + (2+c)CL^2\right) 3Te/2n.$$

*Proof of Theorem 10.* Recall that $\eta = 1$ for convex losses . Assuming that $\alpha_t \leq C/t$ for all $t \leq T$, we have

$$\mathbb{E}\left[\delta_T \mid \mathcal{E}_{\delta_{t_0}}\right]$$

$$\leq \left(\frac{2L}{n}(\Gamma_d^K+1) + \mu\beta C_d + \frac{A_d}{\mu n}\right) \sum_{t=t_0+1}^{T} \alpha_t \prod_{j=t+1}^{T} \left(1 + \alpha_j\beta\Gamma_d^K + \frac{\alpha_j LA_d}{\mu}\right)$$

$$\leq C\left(\frac{2L}{n}(\Gamma_d^K+1) + \mu\beta C_d + \frac{A_d}{\mu n}\right) \sum_{t=t_0+1}^{T} \frac{1}{t} \prod_{j=t+1}^{T} \left(1 + \frac{C\beta\Gamma_d^K + \frac{CLA_d}{\mu}}{j}\right)$$

$$\leq C\left(\frac{2L}{n}(\Gamma_d^K+1) + \mu\beta C_d + \frac{A_d}{\mu n}\right) \sum_{t=t_0+1}^{T} \frac{1}{t} \prod_{j=t+1}^{T} \exp\left(\frac{C\beta\Gamma_d^K + \frac{CLA_d}{\mu}}{j}\right)$$

$$\leq C(eT)^{C\beta\Gamma_d^K + \frac{CLA_d}{\mu}} \left(\frac{2L}{n}(\Gamma_d^K+1) + \mu\beta C_d + \frac{A_d}{\mu n}\right) \sum_{t=t_0+1}^{T} \frac{1}{t} \frac{1}{(t+1)^{C\beta\Gamma_d^K \frac{CLA_d}{\mu}}}$$

$$\leq \underbrace{\left(\beta\Gamma_d^K + \frac{LA_d}{\mu}\right)^{-1} \left(\frac{2L}{n}(\Gamma_d^K+1) + \mu\beta C_d + \frac{A_d}{\mu n}\right)}_{D} \left(\left(\frac{eT}{t_0}\right)^{C\beta\Gamma_d^K + \frac{CLA_d}{\mu}} - e^{C\beta\Gamma_d^K + \frac{CLA_d}{\mu}}\right)$$

We define $q \triangleq C\beta\Gamma_d^K + \frac{CA_d}{\mu}$ and find the value of $t_0$ that minimizes the right part of

$$\mathbb{E}\left[|f(W_T, z) - f(W_T', z)|\right] \leq \frac{t_0}{n} \sup_{w,z} f(w,z) + L\mathbb{E}\left[\delta_T \mid \mathcal{E}_{\delta_{t_0}}\right] \leq \frac{t_0}{n} + LD\left(\left(\frac{eT}{t_0}\right)^q - e^q\right),$$

which is $t_0^* = \min\left\{(qnLD)^{1/(q+1)}(eT)^{q/(q+1)}, T\right\}$. Then we give

$$\mathbb{E}\left[|f(W_T, z) - f(W_T', z)|\right]$$

$$\leq \max\left\{\frac{(qnLD)^{\frac{1}{q+1}}(eT)^{\frac{q}{q+1}}}{n}, \frac{1+1/q}{n}(qnLD)^{\frac{1}{q+1}}(eT)^{\frac{q}{q+1}} - LDe^q\right\}$$

$$\leq \frac{1 + \frac{1}{C\beta\Gamma_d^K + \frac{CLA_d}{\mu}}}{n} \left(2L^2(\Gamma_d^K+1) + \mu\beta C_d nL + \frac{A_d L}{\mu}\right)^{\frac{1}{C\beta\Gamma_d^K + \frac{CLA_d}{\mu}+1}} (eT)^{\frac{C\beta\Gamma_d^K + \frac{CLA_d}{\mu}}{C\beta\Gamma_d^K + \frac{CLA_d}{\mu}+1}}$$

$$\leq \frac{1 + \frac{1}{C\beta\Gamma_d^K + \frac{CLA_d}{\mu}}}{n} n^{\frac{1}{C\beta\Gamma_d^K + \frac{CLA_d}{\mu}+1}} \left(2L^2(\Gamma_d^K+1) + C\beta L + \frac{A_d L}{\mu\sqrt{n}}\right)^{\frac{1}{C\beta\Gamma_d^K + \frac{CLA_d}{\mu}+1}} (eT)^{\frac{C\beta\Gamma_d^K + \frac{CLA_d}{\mu}}{C\beta\Gamma_d^K + \frac{CLA_d}{\mu}+1}}$$

$$\leq \left(1 + \left(\beta C + \frac{CL}{c}\right)^{-1}\right) \frac{1}{n^{1-\frac{c}{d^2 L C \sqrt{n}}}} \left(C\beta\Gamma_d^K + \frac{CLA_d}{\mu} + 1\right)^{\frac{1}{C\beta\Gamma_d^K + \frac{CLA_d}{\mu} + 1}} (eT)^{\frac{C\beta\Gamma_d^K + \frac{CLA_d}{\mu}}{C\beta\Gamma_d^K + \frac{CLA_d}{\mu} + 1}}$$

$$\leq \left(1 + \left(\beta C + \frac{CL}{c}\right)^{-1}\right) \frac{Te^{1+1/e}}{n^{1-\frac{c}{d^2 L C \sqrt{n}}}}$$

$\square$

**Theorem 11** (strongly convex). *Assume that the loss $f(\cdot, z) \in [0,1]$ is L-Lipschitz, $\beta$-smooth and $\gamma$-strongly convex for all $z \in \mathcal{Z}$. Consider the update rule Eq.equation 10 with $T$ the total number of iterates, $\alpha_t \leq C/t$ for some (fixed) $C > 0$ and for all $t \leq T$. Then the generalization error of ZO-SGD with 2-point gradient estimation is bounded by*

$$\epsilon_{gen} \leq \left(1 + (\beta C)^{-1}\right)^2 \left(1 + (2+c)CL^2\right) 3Te/2n.$$

*Proof of Theorem 11.* Recall that $\eta = 1 - \frac{\alpha_t \beta \gamma}{\beta + \gamma}$ for $\gamma$-strongly convex losses . Assuming that $\alpha_t \leq C/t$ for all $t \leq T$, we have

$$\mathbb{E}\left[\delta_T \mid \mathcal{E}_{\delta_{t_0}}\right]$$

$$\leq \left(\frac{2L}{n}(\Gamma_d^K + 1) + \mu\beta C_d + \frac{A_d}{\mu n}\right) \sum_{t=t_0+1}^{T} \alpha_t \prod_{j=t+1}^{T} \left(1 + \alpha_j \beta(\Gamma_d^K - \frac{\gamma}{\beta+\gamma}) + \frac{\alpha_j L A_d}{\mu}\right)$$

$$\leq C\left(\frac{2L}{n}(\Gamma_d^K + 1) + \mu\beta C_d + \frac{A_d}{\mu n}\right) \sum_{t=t_0+1}^{T} \frac{1}{t} \prod_{j=t+1}^{T} \left(1 + \frac{C\beta\Gamma_d^K + \frac{CLA_d}{\mu}}{j}\right)$$

$$\leq C\left(\frac{2L}{n}(\Gamma_d^K + 1) + \mu\beta C_d + \frac{A_d}{\mu n}\right) \sum_{t=t_0+1}^{T} \frac{1}{t} \prod_{j=t+1}^{T} \exp\left(\frac{C\beta\Gamma_d^K + \frac{CLA_d}{\mu}}{j}\right)$$

$$\leq C(eT)^{C\beta\Gamma_d^K + \frac{CLA_d}{\mu}} \left(\frac{2L}{n}(\Gamma_d^K + 1) + \mu\beta C_d + \frac{A_d}{\mu n}\right) \sum_{t=t_0+1}^{T} \frac{1}{t} \frac{1}{(t+1)^{C\beta\Gamma_d^K \frac{CLA_d}{\mu}}}$$

$$\leq \underbrace{\left(\beta\Gamma_d^K + \frac{LA_d}{\mu}\right)^{-1} \left(\frac{2L}{n}(\Gamma_d^K + 1) + \mu\beta C_d + \frac{A_d}{\mu n}\right)}_{D} \left(\left(\frac{eT}{t_0}\right)^{C\beta\Gamma_d^K + \frac{CLA_d}{\mu}} - e^{C\beta\Gamma_d^K + \frac{CLA_d}{\mu}}\right)$$

We define $q \triangleq C\beta\Gamma_d^K + \frac{CA_d}{\mu}$ and find the value of $t_0$ that minimizes the right part of

$$\mathbb{E}\left[|f(W_T, z) - f(W_T', z)|\right] \leq \frac{t_0}{n} \sup_{w,z} f(w, z) + L\mathbb{E}\left[\delta_T \mid \mathcal{E}_{\delta_{t_0}}\right] \leq \frac{t_0}{n} + LD\left(\left(\frac{eT}{t_0}\right)^q - e^q\right),$$

which is $t_0^* = \min\left\{(qnLD)^{1/(q+1)}(eT)^{q/(q+1)}, T\right\}$. Then we give

$$\mathbb{E}\left[|f(W_T, z) - f(W_T', z)|\right]$$

$$\leq \max\left\{\frac{(qnLD)^{\frac{1}{q+1}}(eT)^{\frac{q}{q+1}}}{n}, \frac{1+1/q}{n}(qnLD)^{\frac{1}{q+1}}(eT)^{\frac{q}{q+1}} - LDe^q\right\}$$

$$\leq \frac{1 + \frac{1}{C\beta\Gamma_d^K + \frac{CLA_d}{\mu}}}{n} \left(2L^2(\Gamma_d^K + 1) + \mu\beta C_d nL + \frac{A_d L}{\mu}\right)^{\frac{1}{C\beta\Gamma_d^K + \frac{CLA_d}{\mu} + 1}} (eT)^{\frac{C\beta\Gamma_d^K + \frac{CLA_d}{\mu}}{C\beta\Gamma_d^K + \frac{CLA_d}{\mu} + 1}}$$

$$\leq \frac{1 + \frac{1}{C\beta\Gamma_d^K + \frac{CLA_d}{\mu}}}{n} n^{\frac{1}{C\beta\Gamma_d^K + \frac{CLA_d}{\mu} + 1}} \left(2L^2(\Gamma_d^K + 1) + C\beta L + \frac{A_d L}{\mu\sqrt{n}}\right)^{\frac{1}{C\beta\Gamma_d^K + \frac{CLA_d}{\mu} + 1}} (eT)^{\frac{C\beta\Gamma_d^K + \frac{CLA_d}{\mu}}{C\beta\Gamma_d^K + \frac{CLA_d}{\mu} + 1}}$$

$$\leq \left(1 + \left(\beta C + \frac{CL}{c}\right)^{-1}\right) \frac{1}{n^{1-\frac{c}{d^2 L C \sqrt{n}}}} \left(C\beta\Gamma_d^K + \frac{CLA_d}{\mu} + 1\right)^{\frac{1}{C\beta\Gamma_d^K + \frac{CLA_d}{\mu} + 1}} (eT)^{\frac{C\beta\Gamma_d^K + \frac{CLA_d}{\mu}}{C\beta\Gamma_d^K + \frac{CLA_d}{\mu} + 1}}$$

$$\leq \left(1 + \left(\beta C + \frac{CL}{c}\right)^{-1}\right) \frac{T e^{1+1/e}}{n^{1-\frac{c}{d^2 LC\sqrt{n}}}}$$

$\square$

### G.2 ZO-SGD WITH 2-POINT GRADIENT ESTIMATION

**Lemma 12** (**2-point ZO-SGD Growth Recursion**). *Consider the sequences of updates* $\left\{\tilde{G}_t\right\}_{t=1}^{T}$ *and* $\left\{\tilde{G}'_t\right\}_{t=1}^{T}$. *Let* $w_0 = w'_0$ *be the starting point,* $w_{t+1} = \tilde{G}_t(w_t)$ *and* $w'_{t+1} = \tilde{G}'_t(w'_t)$ *for any* $t \in \{1, \ldots, T\}$. *Then for any* $w_t, w'_t \in \mathbb{R}^d$ *and* $t \geq 0$ *the following recursion holds*

$$\mathbb{E}\left[\left\|\tilde{G}_t(w_t) - \tilde{G}'_t(w'_t)\right\|\right] \leq \begin{cases} (\eta_1 + \alpha_t\beta\Gamma_d^k)\|w_t - w'_t\| + \mu\beta\alpha_t C_d, & \text{if } \tilde{G}_t(\cdot) = \tilde{G}'_t(\cdot), \\ \|w_t - w'_t\| + 2\alpha_t L(\Gamma_d^k + 1) + \mu\beta\alpha_t C_d, & \text{if } \tilde{G}_t(\cdot) \neq \tilde{G}'_t(\cdot). \end{cases}$$

*Proof of Lemma 12.* The formula for central difference is as follows:

$$\hat{\nabla} f(w) = \frac{1}{K}\sum_{k=1}^{K} \frac{f(w + \mu u_k) - f(w - \mu u_k)}{2\mu} u_k$$

Let $S$ and $S'$ be two samples of size $n$ differing in only a single example, and let $\tilde{G}_t(\cdot), \tilde{G}'_t(\cdot)$ be the update rules of the ZO-SGD for each of the sequences $S, S'$ respectively. By applying the Taylor expansion there exist vectors $W_{k,t}^*$, $W_{k,t}^\star$, $W_{k,t}^\circledast$ and $W_{k,t}^\divideontimes$ with $j^{\text{th}}$ coordinates in the intervals $\left(w_t^{(j)}, w_t^{(j)} + \mu U_{k,t}^{(j)}\right) \cup \left(w_t^{(j)} + \mu U_{k,t}^{(j)}, w_t^{(j)}\right), \left(w_t^{(j)} - \mu U_{k,t}^{(j)}, w_t^{(j)},\right) \cup \left(w_t^{(j)}, w_t^{(j)} - \mu U_{k,t}^{(j)}\right),$ $\left(w_t'^{(j)}, w_t'^{(j)} + \mu U_{k,t}^{(j)}\right) \cup \left(w_t'^{(j)} + \mu U_{k,t}^{(j)}, w_t'^{(j)}\right), \left(w_t'^{(j)} - \mu U_{k,t}^{(j)}, w_t'^{(j)}\right) \cup \left(w_t'^{(j)}, w_t'^{(j)} - \mu U_{k,t}^{(j)}\right),$ such that for any $w_t, w'_t \in \mathbb{R}^d$ we have

$$\mathbb{E}\left[\left\|\tilde{G}_t(w_t) - \tilde{G}_t(w'_t)\right\|\right]$$

$$\leq \|G(w_t) - G'(w'_t)\| + \alpha_t\left\|(\nabla f(w_t) - \hat{\nabla} f(w_t)) - (\nabla f(w'_t) - \hat{\nabla} f(w'_t))\right\|$$

$$\leq \|G(w_t) - G'(w'_t)\| + \alpha_t\left\|\left(\frac{1}{K}\sum_{k=1}^{K}\langle\nabla f(w_t) - \nabla f(w'_t,), u_k^t\rangle u_k^t - (\nabla f(w_t) - \nabla f(w'_t))\right)\right\|$$

$$+ \alpha_t\left\|\frac{1}{K}\sum_{k=1}^{K}\left(\frac{\mu}{4}u_k^{\mathrm{T}}\nabla_w^2 f(W_{k,t}^*) u_k^t\right)u_k^t\right\| + \alpha_t\left\|\frac{1}{K}\sum_{k=1}^{K}\left(\frac{\mu}{4}u_k^{\mathrm{T}}\nabla_w^2 f(W_{k,t}^\star) u_k^t\right)u_k^t\right\|$$

$$+ \alpha_t\left\|\frac{1}{K}\sum_{k=1}^{K}\left(\frac{\mu}{4}u_k^{\mathrm{T}}\nabla_w^2 f(W_{k,t}^\circledast) u_k^t\right)u_k^t\right\| + \alpha_t\left\|\frac{1}{K}\sum_{k=1}^{K}\left(\frac{\mu}{4}u_k^{\mathrm{T}}\nabla_w^2 f(W_{k,t}^\divideontimes) u_k^t\right)u_k^t\right\|$$

$$\leq \underbrace{\|G(w_t) - G'(w'_t)\|}_{A} + \underbrace{\alpha_t\Gamma_d^k\mathbb{E}\left[\|\nabla f(w_t, z_{i_t}) - \nabla f(w'_t, z_{i_t})\|\right]}_{B} + \underbrace{\frac{\alpha_t}{K}\sum_{k=1}^{K}\mu\beta\mathbb{E}\left[\|u_k^t\|^3\right]}_{E}$$

Part A corresponds to the first-order stochastic gradient descent (SGD), which decomposes into $\eta$ and $\sigma_{t,2}$ terms. Part B corresponds to the scaling term introduced by the gradient approximation, which leads to a new inflation factor $\eta'$ and a new bound. Part E corresponds to the irreducible term caused by the difference between the estimated gradient and the true gradient, denoted as $\sigma_{t,1}$. Under the conditions of $L$-Lipschitz and $\beta$-smooth and $\mathbb{E}\left[\|u_k^t\|^3\right] \leq C_d$, we get $\eta' = \alpha_t\beta\Gamma_d^k$, $\sigma_{t,1} = \mu\beta\alpha_t C_d$, $\sigma_{t,2} = L$, $\sigma_{t,3} = 2\alpha_t L\Gamma_d^k + \mu\beta\alpha C_d$. Thus the expansive and bound are the same as the forward difference results from the previous work, thus their generalization bounds are also identical.

$\square$

**Lemma 13.** *Assume that the loss function $f(\cdot, z)$ is $L$ Lipschitz and $\beta$-smooth for all $z \in \mathcal{Z}$. Consider the ZO-SGD algorithm with final-iterate estimates $W_T$ and $W'_T$, corresponding to the data-sets $S, S'$, respectively (that differ in exactly one entry). Then the discrepancy $\delta_T \triangleq \|W_T - W'_T\|$, under the event $\mathcal{E}_{\delta_{t_0}}$, satisfies the inequality*

$$\mathbb{E}\left[\delta_T \mid \mathcal{E}_{\delta_{t_0}}\right] \leq \left(\frac{2L}{n}(\Gamma_d^k + 1) + \mu\beta C_d\right) \sum_{t=t_0+1}^{T} \alpha_t \prod_{j=t+1}^{T} \left(1 + \beta\alpha_j(\Gamma_d^k + 1)\right)$$

*Proof of Lemma 13.* Consider the events $\mathcal{E}_t \triangleq \left\{\tilde{G}_t(\cdot) \equiv \tilde{G}'_t(\cdot)\right\}$ and $\mathcal{E}_t^c \triangleq \left\{\tilde{G}_t(\cdot) \neq \tilde{G}'_t(\cdot)\right\}$ (see Eq. (4)). Recall that $\mathbb{P}(\mathcal{E}_t) = 1 - 1/n$ and $\mathbb{P}(\mathcal{E}_t^c) = 1/n$ for all $t \leq T$. For any $t_0 \geq 0$, a direct application of Lemma 3 gives

$$\mathbb{E}\left[\delta_{t+1} \mid \mathcal{E}_{\delta_{t_0}}\right] = \mathbb{P}(\mathcal{E}_t)\,\mathbb{E}\left[\delta_{t+1} \mid \mathcal{E}_t, \mathcal{E}_{\delta_{t_0}}\right] + \mathbb{P}(\mathcal{E}_t^c)\,\mathbb{E}\left[\delta_{t+1} \mid \mathcal{E}_t^c, \mathcal{E}_{\delta_{t_0}}\right]$$

$$= \left(1 - \frac{1}{n}\right)\mathbb{E}\left[\delta_{t+1} \mid \mathcal{E}_t, \mathcal{E}_{\delta_{t_0}}\right] + \frac{1}{n}\mathbb{E}\left[\delta_{t+1} \mid \mathcal{E}_t^c, \mathcal{E}_{\delta_{t_0}}\right]$$

$$\leq \left(\eta + \alpha_t\beta\Gamma_d^k + \frac{1}{n}\left(1 - \eta - \alpha_t\beta\Gamma_d^k\right)\right)\mathbb{E}\left[\delta_t \mid \mathcal{E}_{\delta_{t_0}}\right] + \frac{2\alpha_t L}{\eta}(\Gamma_d^k + 1) + \mu\beta\alpha_t C_d.$$

then by solving this recursion we have

$$\mathbb{E}\left[\delta_T \mid \mathcal{E}_{\delta_{t_0}}\right] \leq \left(\frac{2L}{n}(\Gamma_d^k + 1) + \mu\beta C_d\right) \sum_{t=t_0+1}^{T} \alpha_t \prod_{j=t+1}^{T} \left(1 + \beta\alpha_j(\Gamma_d^k + 1)\left(1 - \frac{1}{n}\right)\right)$$

$\square$

*Proof of Theorem 2.* Recall that $\eta = 1 + \beta\alpha_t$ for general (nonconvex) losses . Assuming that $\alpha_t \leq C/t$ for all $t \leq T$, we have

$$\mathbb{E}\left[\delta_T \mid \mathcal{E}_{\delta_{t_0}}\right] \leq \left(\frac{2L}{n}(\Gamma_d^k + 1) + \mu\beta C_d\right) \sum_{t=t_0+1}^{T} \alpha_t \prod_{j=t+1}^{T} \left(1 + \alpha_j\beta(\Gamma_d^k + 1)\right)$$

$$\leq C\left(\frac{2L}{n}(\Gamma_d^k + 1) + \mu\beta C_d\right) \sum_{t=t_0+1}^{T} \frac{1}{t} \prod_{j=t+1}^{T} \left(1 + \frac{C\beta(\Gamma_d^k + 1)}{j}\right)$$

$$\leq C\left(\frac{2L}{n}(\Gamma_d^k + 1) + \mu\beta C_d\right) \sum_{t=t_0+1}^{T} \frac{1}{t} \prod_{j=t+1}^{T} \exp\left(\frac{C\beta(\Gamma_d^k + 1)}{j}\right)$$

$$\leq C(eT)^{C\beta(\Gamma_d^k + 1)}\left(\frac{2L}{n}(\Gamma_d^k + 1) + \mu\beta C_d\right) \sum_{t=t_0+1}^{T} \frac{1}{t}\frac{1}{(t+1)^{C\beta(\Gamma_d^k + 1)}}$$

$$\leq \underbrace{(\beta(\Gamma_d^k + 1))^{-1}\left(\frac{2L}{n}(\Gamma_d^k + 1) + \mu\beta C_d\right)}_{D}\left(\left(\frac{eT}{t_0}\right)^{C\beta(\Gamma_d^k + 1)} - e^{C\beta(\Gamma_d^k + 1)}\right)$$

We define $q \triangleq C\beta(\Gamma_d^k + 1)$ and find the value of $t_0$ that minimizes the right part of

$$\mathbb{E}\left[|f(W_T, z) - f(W'_T, z)|\right] \leq \frac{t_0}{n}\sup_{w,z} f(w, z) + L\mathbb{E}\left[\delta_T \mid \mathcal{E}_{\delta_{t_0}}\right] \leq \frac{t_0}{n} + LD\left(\left(\frac{eT}{t_0}\right)^q - e^q\right),$$

which is $t_0^* = \min\left\{(qnLD)^{1/(q+1)}(eT)^{q/(q+1)}, T\right\}$ and $\mu \leq \frac{\Gamma_d^k + 1}{n\beta C_d}$ Then we gives

$$\mathbb{E}\left[|f(W_T, z) - f(W'_T, z)|\right]$$

$$\leq \max\left\{\frac{(qnLD)^{\frac{1}{q+1}}(eT)^{\frac{q}{q+1}}}{n}, \frac{1 + 1/q}{n}(qnLD)^{\frac{1}{q+1}}(eT)^{\frac{q}{q+1}} - LDe^q\right\}$$

$$\leq \frac{1 + \frac{1}{C\beta(\Gamma_d^k+1)}}{n} \left((2+c)CL^2\left(\Gamma_d^k+1\right)\right)^{\frac{1}{C\beta(\Gamma_d^k+1)+1}} (eT)^{\frac{C\beta(\Gamma_d^k+1)}{C\beta(\Gamma_d^k+1)+1}}$$

$$\leq \left(1+(\beta C)^{-1}\right)^2\left(1+(2+c)CL^2\right)\frac{3Te}{2n}.$$

$\square$

**Theorem 12** (convex). *Assume that the loss $f(\cdot,z) \in [0,1]$ is L-Lipschitz, $\beta$-smooth and convex for all $z \in \mathcal{Z}$. Consider the update rule Eq.equation 11 with $T$ the total number of iterates, $\alpha_t \leq C/t$ for some (fixed) $C > 0$ and for all $t \leq T$. Then the generalization error of ZO-SGD with 2-point gradient estimation is bounded by*

$$\epsilon_{gen} \leq \left(1+(\beta C)^{-1}\right)^2\left(1+(2+c)CL^2\right)3Te/2n.$$

*Proof of Theorem 12.* Recall that $\eta = 1$ for convex losses. Assuming that $\alpha_t \leq C/t$ for all $t \leq T$, we have

$$\mathbb{E}\left[\delta_T \mid \mathcal{E}_{\delta_{t_0}}\right] \leq \left(\frac{2L}{n}(\Gamma_d^k+1)+\mu\beta C_d\right)\sum_{t=t_0+1}^{T}\alpha_t\prod_{j=t+1}^{T}\left(1+\alpha_j\beta\Gamma_d^k\right)$$

$$\leq C\left(\frac{2L}{n}(\Gamma_d^k+1)+\mu\beta C_d\right)\sum_{t=t_0+1}^{T}\frac{1}{t}\prod_{j=t+1}^{T}\left(1+\frac{C\beta(\Gamma_d^k+1)}{j}\right)$$

$$\leq C\left(\frac{2L}{n}(\Gamma_d^k+1)+\mu\beta C_d\right)\sum_{t=t_0+1}^{T}\frac{1}{t}\prod_{j=t+1}^{T}\exp\left(\frac{C\beta(\Gamma_d^k+1)}{j}\right)$$

$$\leq C(eT)^{C\beta(\Gamma_d^k+1)}\left(\frac{2L}{n}\left(\Gamma_d^k+1\right)+\mu\beta C_d\right)\sum_{t=t_0+1}^{T}\frac{1}{t}\frac{1}{(t+1)^{C\beta(\Gamma_d^k+1)}}$$

$$\leq \underbrace{\left(\beta\Gamma_d^k\right)^{-1}\left(\frac{2L}{n}\left(\Gamma_d^k+1\right)+\mu\beta C_d\right)}_{D}\left(\left(\frac{eT}{t_0}\right)^{C\beta(\Gamma_d^k+1)}-e^{C\beta(\Gamma_d^k+1)}\right)$$

We define $q \triangleq C\beta\Gamma_d^k$ and find the value of $t_0$ that minimizes the right part of

$$\mathbb{E}\left[|f\left(W_T,z\right)-f\left(W_T',z\right)|\right] \leq \frac{t_0}{n}\sup_{w,z}f(w,z)+L\mathbb{E}\left[\delta_T\mid\mathcal{E}_{\delta_{t_0}}\right] \leq \frac{t_0}{n}+LD\left(\left(\frac{eT}{t_0}\right)^q-e^q\right),$$

which is $t_0^* = \min\left\{(qnLD)^{1/(q+1)}(eT)^{q/(q+1)},T\right\}$ and $\mu \leq \frac{\Gamma_d^k+1}{n\beta C_d}$ Then we gives

$$\mathbb{E}\left[|f\left(W_T,z\right)-f\left(W_T',z\right)|\right]$$

$$\leq \max\left\{\frac{(qnLD)^{\frac{1}{q+1}}(eT)^{\frac{q}{q+1}}}{n},\frac{1+1/q}{n}(qnLD)^{\frac{1}{q+1}}(eT)^{\frac{q}{q+1}}-LDe^q\right\}$$

$$\leq \frac{1+\frac{1}{C\beta(\Gamma_d^k+1)}}{n}\left((2+c)CL^2\left(\Gamma_d^k+1\right)\right)^{\frac{1}{C\beta(\Gamma_d^k+1)+1}}(eT)^{\frac{C\beta(\Gamma_d^k+1)}{C\beta(\Gamma_d^k+1)+1}}$$

$$\leq \left(1+(\beta C)^{-1}\right)^2\left(1+(2+c)CL^2\right)\frac{3Te}{2n}.$$

$\square$

**Theorem 13** (strongly convex). *Assume that the loss $f(\cdot,z) \in [0,1]$ is L-Lipschitz, $\beta$-smooth and $\gamma$-strongly convex for all $z \in \mathcal{Z}$. Consider the update rule Eq.equation 11 with $T$ the total number of iterates, $\alpha_t \leq C/t$ for some (fixed) $C > 0$ and for all $t \leq T$. Then the generalization error of ZO-SGD with 2-point gradient estimation is bounded by*

$$\epsilon_{gen} \leq \left(1+(\beta C)^{-1}\right)^2\left(1+(2+c)CL^2\right)3Te/2n.$$

*Proof of Theorem 13.* Recall that $\eta = 1 - \frac{\alpha_t \beta \gamma}{\beta + \gamma}$ for $\gamma$-strongly convex losses . Assuming that $\alpha_t \leq C/t$ for all $t \leq T$, we have

$$\mathbb{E}\left[\delta_T \mid \mathcal{E}_{\delta_{t_0}}\right] \leq \left(\frac{2L}{n}(\Gamma_d^k + 1) + \mu\beta C_d\right) \sum_{t=t_0+1}^{T} \alpha_t \prod_{j=t+1}^{T} \left(1 + \alpha_j\beta(\Gamma_d^K - \frac{\gamma}{\beta+\gamma})\right)$$

$$\leq C\left(\frac{2L}{n}(\Gamma_d^k + 1) + \mu\beta C_d\right) \sum_{t=t_0+1}^{T} \frac{1}{t} \prod_{j=t+1}^{T} \left(1 + \frac{C\beta(\Gamma_d^k + 1)}{j}\right)$$

$$\leq C\left(\frac{2L}{n}(\Gamma_d^k + 1) + \mu\beta C_d\right) \sum_{t=t_0+1}^{T} \frac{1}{t} \prod_{j=t+1}^{T} \exp\left(\frac{C\beta(\Gamma_d^k + 1)}{j}\right)$$

$$\leq C(eT)^{C\beta(\Gamma_d^k+1)} \left(\frac{2L}{n}(\Gamma_d^k + 1) + \mu\beta C_d\right) \sum_{t=t_0+1}^{T} \frac{1}{t} \frac{1}{(t+1)^{C\beta(\Gamma_d^k+1)}}$$

$$\leq \underbrace{(\beta\Gamma_d^k)^{-1} \left(\frac{2L}{n}(\Gamma_d^k + 1) + \mu\beta C_d\right)}_{D} \left(\left(\frac{eT}{t_0}\right)^{C\beta(\Gamma_d^k+1)} - e^{C\beta(\Gamma_d^k+1)}\right)$$

We define $q \triangleq C\beta\Gamma_d^k$ and find the value of $t_0$ that minimizes the right part of

$$\mathbb{E}\left[|f(W_T, z) - f(W_T', z)|\right] \leq \frac{t_0}{n} \sup_{w,z} f(w,z) + L\mathbb{E}\left[\delta_T \mid \mathcal{E}_{\delta_{t_0}}\right] \leq \frac{t_0}{n} + LD\left(\left(\frac{eT}{t_0}\right)^q - e^q\right),$$

which is $t_0^* = \min\left\{(qnLD)^{1/(q+1)}(eT)^{q/(q+1)}, T\right\}$ and $\mu \leq \frac{\Gamma_d^k+1}{n\beta C_d}$ Then we gives

$$\mathbb{E}\left[|f(W_T, z) - f(W_T', z)|\right]$$

$$\leq \max\left\{\frac{(qnLD)^{\frac{1}{q+1}}(eT)^{\frac{q}{q+1}}}{n}, \frac{1+1/q}{n}(qnLD)^{\frac{1}{q+1}}(eT)^{\frac{q}{q+1}} - LDe^q\right\}$$

$$\leq \frac{1 + \frac{1}{C\beta(\Gamma_d^k+1)}}{n}\left((2+c)CL^2\left(\Gamma_d^k + 1\right)\right)^{\frac{1}{C\beta(\Gamma_d^k+1)+1}}(eT)^{\frac{C\beta(\Gamma_d^k+1)}{C\beta(\Gamma_d^k+1)+1}}$$

$$\leq \left(1 + (\beta C)^{-1}\right)^2\left(1 + (2+c)CL^2\right)\frac{3Te}{2n}.$$

$\square$

### G.3 ZO-SGD WITH COORDINATE-WISE GRADIENT ESTIMATION

**Lemma 14.** *Consider the sequences of updates $\left\{\tilde{G}_t\right\}_{t=1}^{T}$ and $\left\{\tilde{G}_t'\right\}_{t=1}^{T}$. Let $w_0 = w_0'$ be the starting point, $w_{t+1} = \tilde{G}_t(w_t)$ and $w_{t+1}' = \tilde{G}_t'(w_t')$ for any $t \in \{1, \ldots, T\}$. Then for any $w_t, w_t' \in \mathbb{R}^d$ and $t \geq 0$ the following recursion holds*

$$\mathbb{E}\left[\left\|\tilde{G}_t(w_t) - \tilde{G}_t'(w_t')\right\|\right] \leq \begin{cases} \eta \|w_t - w_t'\| + \alpha_t\mu\beta d, & \text{if } \tilde{G}_t(\cdot) = \tilde{G}_t'(\cdot), \\ \|w_t - w_t'\| + 2\alpha_t L + \alpha_t\mu\beta d, & \text{if } \tilde{G}_t(\cdot) \neq \tilde{G}_t'(\cdot). \end{cases}$$

*Proof of Theorem 3 (Convex).* The formula for coordinate-wise estimator is as follows:

$$w_{t+1} = w_t - \alpha_t \begin{cases} \sum_{i=1}^{d} \frac{f(w_t + \mu\mathbf{e}_i, z_{i_t}) - f(w_t, z_{i_t})}{\mu}\mathbf{e}_i, & \text{forward difference,} \\ \sum_{i=1}^{d} \frac{f(w_t + \mu\mathbf{e}_i, z_{i_t}) - f(w_t - \mu\mathbf{e}_i, z_{i_t})}{2\mu}\mathbf{e}_i, & \text{central difference.} \end{cases}$$

Let $S$ and $S'$ be two samples of size $n$ differing in only a single example, and let $\tilde{G}_t(\cdot), \tilde{G}_t'(\cdot)$ be the update rules of the ZO-SGD for each of the sequences $S, S'$ respectively. By applying the Taylor expansion there exist vectors $W_{i,t}^*$ and $W_{i,t}^\dagger$ with $i^{\text{th}}$ coordinates in the intervals $(w_t, w_t + \mu e_i)$ and $(w_t', w_t' + \mu e_i)$, respectively, such that for any $w_t, w_t' \in \mathbb{R}^d$ we have

$$\|\tilde{G}_t\left(w_t\right) - \tilde{G}'_t\left(w'_t\right)\| = \|\tilde{G}_t\left(w_t\right) - \tilde{G}_t\left(w'_t\right)\|$$

$$= \|w_t - w'_t - \alpha_t \sum_{i=1}^{d} \langle \nabla f\left(w_t, z_{i_t}\right) - \nabla f\left(w'_t, z_{i_t}\right), e_i \rangle e_i$$

$$- \alpha_t \sum_{i=1}^{d} \left(\frac{\mu}{2} e_i^{\mathrm{T}} \nabla^2 f\left(W_{i,t}^*, z_{i_t}\right) e_i\right) e_i + \alpha_t \sum_{i=1}^{d} \left(\frac{\mu}{2} e_i^{\mathrm{T}} \nabla^2 f\left(W_{i,t}^\dagger, z_{i_t}\right) e_i\right) e_i\|$$

$$= \|w_t - \alpha_t \nabla f\left(w_t, z_{i_t}\right) - \left(w'_t - \alpha_t \nabla f\left(w'_t, z_{i_t}\right)\right)$$

$$- \alpha_t \sum_{i=1}^{d} \left(\frac{\mu}{2} e_i^{\mathrm{T}} \nabla_w^2 f\left(w, z_{i_t}\right)\Big|_{w=W_{i,t}^*} e_i\right) e_i + \alpha_t \sum_{i=1}^{d} \left(\frac{\mu}{2} e_i^{\mathrm{T}} \nabla_w^2 f\left(w, z_{i_t}\right)\Big|_{w=W_{i,t}^\dagger} e_i\right) e_i$$

$$- \alpha_t \left(\sum_{i=1}^{d} \langle \nabla f\left(w_t, z_{i_t}\right) - \nabla f\left(w'_t, z_{i_t}\right), e_i \rangle e_i - \left(\nabla f\left(w_t, z_{i_t}\right) - \nabla f\left(w'_t, z_{i_t}\right)\right)\right)\|$$

$$= \|G\left(w_t\right) - G\left(w'_t\right) - \alpha_t \sum_{i=1}^{d} \left(\frac{\mu}{2} e_i^{\mathrm{T}} \nabla_w^2 f\left(w, z_{i_t}\right)\Big|_{w=W_{i,t}^*} e_i\right) e_i$$

$$+ \alpha_t \sum_{i=1}^{d} \left(\frac{\mu}{2} e_i^{\mathrm{T}} \nabla_w^2 f\left(w, z_{i_t}\right)\Big|_{w=W_{i,t}^\dagger} e_i\right) e_i\|.$$

For $\sum_{i=1}^{d} \langle \nabla f\left(w_t, z_{i_t}\right) - \nabla f\left(w'_t, z_{i_t}\right), e_i \rangle e_i = \nabla f\left(w_t, z_{i_t}\right) - \nabla f\left(w'_t, z_{i_t}\right)$. This is the key, why ZO-SGD using coordinate-wise estimator is non-expansive. The last display and the triangle inequality give

$$\mathbb{E}\left[\left\|\tilde{G}_t\left(w_t\right) - \tilde{G}_t\left(w'_t\right)\right\|\right]$$

$$\leq \|G\left(w_t\right) - G\left(w'_t\right)\| + 2\alpha_t \sum_{i=1}^{d} \frac{\mu\beta}{2} \mathbb{E}\left[\|e_i\|^3\right]$$

$$\leq \|G\left(w_t\right) - G\left(w'_t\right)\| + \alpha_t \mu\beta d.$$

Above all, coordinate-wise estimator

Let $S$ and $S'$ be two samples of size $n$ differing in only a single example. Consider the gradient updates $G_1, \ldots, G_T$ and $G'_1, \ldots, G'_T$ induced by running ZO-SGD on sample $S$ and $S'$, respectively. Let $w_T$ and $w'_T$ denote the corresponding outputs of ZO-SGD. We now fix an example $z \in Z$ and apply the Lipschitz condition on $f(\cdot; z)$ to get

$$\mathbb{E}\left|f\left(w_T; z\right) - f\left(w'_T; z\right)\right| \leq L\mathbb{E}\left[\delta_T\right],$$

where $\delta_T = \|w_T - w'_T\|$. Observe that at step $t$, with probability $1 - 1/n$, the example selected by SGD is the same in both $S$ and $S'$. In this case we have that $G_t = G'_t$ and we can use the 1-expansivity of the update rule $G_t$ which uses the fact that the objective function is convex and that $\alpha_t \leq 2/\beta$. With probability $1/n$ the selected example is different in which case we use that both $G_t$ and $G'_t$ are $\alpha_t L$-bounded as a consequence of Lemma 12. Hence, we have

$$\mathbb{E}\left[\delta_{t+1}\right] \leq \left(1 - \frac{1}{n}\right) \mathbb{E}\left[\delta_t\right] + \frac{1}{n} \mathbb{E}\left[\delta_t\right] + \frac{2\alpha_t L}{n} + \frac{\beta\alpha_t}{n} = \mathbb{E}\left[\delta_t\right] + \frac{2\alpha_t L + \alpha_t \beta}{n}$$

Unraveling the recursion gives

$$\mathbb{E}\left[\delta_T\right] \leq \frac{2L + \beta}{n} \sum_{t=1}^{T} \alpha_t$$

Plugging this back into equation, we obtain

$$\mathbb{E}\left|f\left(w_T; z\right) - f\left(w'_T; z\right)\right| \leq \frac{2L^2 + L\beta}{n} \sum_{t=1}^{T} \alpha_t$$

□

*Proof of Theorem 3 (Strongly Convex).* Let $S$ and $S'$ be two samples of size $n$ differing in only a single example. Consider the gradient updates $G_1, \ldots, G_T$ and $G'_1, \ldots, G'_T$ induced by running ZO-SGD on sample $S$ and $S'$, respectively. Let $w_T$ and $w'_T$ denote the corresponding outputs of ZO-SGD. We now fix an example $z \in Z$ and apply the Lipschitz condition on $f(\cdot; z)$ to get

$$\mathbb{E} \left| f(w_T; z) - f(w'_T; z) \right| \leq L \mathbb{E} [\delta_T]$$

If $\alpha \leq 1/\beta$ : since $\frac{2\alpha\beta\gamma}{\beta+\gamma} \geq \alpha\gamma$ and $\alpha\gamma \leq 1$, $G_{f,\alpha}$ is $(1 - \alpha\gamma)$-expansive. With probability $1/n$ the selected example is different in which case we use that both $G_t$ and $G'_t$ are $\alpha L$-bounded

$$\mathbb{E} [\delta_{t+1}] \leq \left(1 - \frac{1}{n}\right) \mathbb{E} [\delta_t] + \frac{1}{n} \mathbb{E} [\delta_t] + \frac{2\alpha L}{n} + \frac{\mu\alpha}{n} = (1 - \alpha\gamma) \mathbb{E} [\delta_t] + \frac{2\alpha L + \alpha\beta}{n}.$$

Unraveling the recursion gives

$$\mathbb{E} [\delta_T] \leq \frac{2\alpha L + \alpha\beta}{n} \sum_{t=0}^{T} (1 - \alpha\gamma)^t \leq \frac{2L + \beta}{n\gamma}.$$

Plugging this back into equation, we obtain

$$\mathbb{E} \left| f(w_T; z) - f(w'_T; z) \right| \leq \frac{2L^2 + \beta L}{n\gamma}.$$

$\square$

**Lemma 15.** *Assume that the loss function $f(\cdot, z)$ is $L$ Lipschitz and $\beta$-smooth for all $z \in \mathcal{Z}$. Consider the ZO-SGD algorithm with final-iterate estimates $W_T$ and $W'_T$, corresponding to the data-sets $S, S'$, respectively (that differ in exactly one entry). Then the discrepancy $\delta_T \triangleq \|W_T - W'_T\|$, under the event $\mathcal{E}_{\delta_{t_0}}$, satisfies the inequality*

$$\mathbb{E} \left[ \delta_T \mid \mathcal{E}_{\delta_{t_0}} \right] \leq \left( \frac{2L}{n} + \mu\beta \right) \sum_{t=t_0+1}^{T} \alpha_t \prod_{j=t+1}^{T} (1 + \beta\alpha_j)$$

*Proof of Lemma 15.* Consider the events $\mathcal{E}_t \triangleq \left\{ \tilde{G}_t(\cdot) \equiv \tilde{G}'_t(\cdot) \right\}$ and $\mathcal{E}_t^c \triangleq \left\{ \tilde{G}_t(\cdot) \neq \tilde{G}'_t(\cdot) \right\}$ (see Eq. (4)). Recall that $\mathbb{P}(\mathcal{E}_t) = 1 - 1/n$ and $\mathbb{P}(\mathcal{E}_t^c) = 1/n$ for all $t \leq T$. For any $t_0 \geq 0$, a direct application of Lemma 3 gives

$$\mathbb{E} \left[ \delta_{t+1} \mid \mathcal{E}_{\delta_{t_0}} \right] = \mathbb{P}(\mathcal{E}_t) \mathbb{E} \left[ \delta_{t+1} \mid \mathcal{E}_t, \mathcal{E}_{\delta_{t_0}} \right] + \mathbb{P}(\mathcal{E}_t^c) \mathbb{E} \left[ \delta_{t+1} \mid \mathcal{E}_t^c, \mathcal{E}_{\delta_{t_0}} \right]$$

$$= \left(1 - \frac{1}{n}\right) \mathbb{E} \left[ \delta_{t+1} \mid \mathcal{E}_t, \mathcal{E}_{\delta_{t_0}} \right] + \frac{1}{n} \mathbb{E} \left[ \delta_{t+1} \mid \mathcal{E}_t^c, \mathcal{E}_{\delta_{t_0}} \right]$$

$$\leq (1 + \alpha_t\beta) \mathbb{E} \left[ \delta_t \mid \mathcal{E}_{\delta_{t_0}} \right] + \frac{2\alpha_t L}{n} + \mu\beta\alpha_t.$$

then by solving this recursion we have

$$\mathbb{E} \left[ \delta_T \mid \mathcal{E}_{\delta_{t_0}} \right] \leq \left( \frac{2L}{n} + \mu\beta \right) \sum_{t=t_0+1}^{T} \alpha_t \prod_{j=t+1}^{T} (1 + \beta\alpha_j).$$

$\square$

*Proof of Theorem 3 (Nonconvex).* Recall that $\eta = 1 + \beta\alpha_t$ for general (nonconvex) losses . Assuming that $\alpha_t \leq C/t$ for all $t \leq T$ and $\mu \leq \frac{c}{n}$ we have

$$\mathbb{E} \left[ \delta_T \mid \mathcal{E}_{\delta_{t_0}} \right] \leq \left( \frac{2L}{n} + \mu\beta \right) \sum_{t=t_0+1}^{T} \alpha_t \prod_{j=t+1}^{T} (1 + \alpha_j\beta)$$

$$\leq C \left( \frac{2L}{n} + \mu\beta C_d \right) \sum_{t=t_0+1}^{T} \frac{1}{t} \prod_{j=t+1}^{T} \left( 1 + \frac{C\beta}{j} \right)$$

$$\leq C \left( \frac{2L}{n} + \mu\beta C_d \right) \sum_{t=t_0+1}^{T} \frac{1}{t} \prod_{j=t+1}^{T} \exp\left( \frac{C\beta}{j} \right)$$

$$\leq C(eT)^{C\beta} \left( \frac{2L}{n} + \mu\beta \right) \sum_{t=t_0+1}^{T} \frac{1}{t} \frac{1}{(t+1)^{C\beta}}$$

$$\leq \underbrace{\frac{2L + c\beta}{n}}_{D} \left( \left( \frac{eT}{t_0} \right)^{C\beta} - e^{C\beta} \right)$$

We define $q \triangleq C\beta$ and find the value of $t_0$ that minimizes the right part of

$$\mathbb{E}\left[ |f\left(W_T, z\right) - f\left(W'_T, z\right)| \right] \leq \frac{t_0}{n} \sup_{w,z} f(w,z) + L\mathbb{E}\left[ \delta_T \mid \mathcal{E}_{\delta_{t_0}} \right] \leq \frac{t_0}{n} + LD\left( \left( \frac{eT}{t_0} \right)^q - e^q \right),$$

which is $t_0^* = \min\left\{ (qnLD)^{1/(q+1)}(eT)^{q/(q+1)}, T \right\}$. Then we gives

$$\mathbb{E}\left[ |f\left(W_T, z\right) - f\left(W'_T, z\right)| \right]$$

$$\leq \max\left\{ \frac{(qnLD)^{\frac{1}{q+1}}(eT)^{\frac{q}{q+1}}}{n}, \frac{1+1/q}{n}(qnLD)^{\frac{1}{q+1}}(eT)^{\frac{q}{q+1}} - LDe^q \right\}$$

$$\leq \frac{1 + \frac{1}{\beta C}}{n} \left( 2CL^2 + \beta CcL \right)^{\frac{1}{\beta C+1}} (eT)^{\frac{\beta C}{\beta C+1}}.$$

$$\square$$

## H  Proof of Generalization Analysis of ZO-GD

In the case of full-batch GD and ZO-GD the algorithm is deterministic and we assume that $z_1, z_2, \ldots, z_i, \ldots, z_n, z'_j$ are i.i.d. and define $S \triangleq (z_1, z_2, \ldots, z_i, \ldots, z_n)$ and $S' \triangleq (z_1, z_2, \ldots, z'_j, \ldots, z_n)$, $W_0 = W'_0$, the updates for any $t \geq 1$ are

$$w_{t+1} = w_t - \frac{\alpha_t}{n} \sum_{j=1}^{n} \nabla f\left(w_t, z_j\right) = \frac{1}{n} \sum_{i=1}^{n} \left(w_t - \alpha_t \nabla f(w_t, z_i)\right),$$

$$w'_{t+1} = w'_t - \frac{\alpha_t}{n} \sum_{j=1, j\neq i}^{n} \nabla f\left(w'_t, z_j\right) - \frac{\alpha_t}{n} \nabla f\left(w'_t, z'_i\right)$$

$$= \frac{1}{n} \left[ \sum_{i=1, i\neq j}^{n} \left(w'_t - \alpha_t \nabla f(w'_t, z_i)\right) + \left(w'_t - \alpha_t \nabla f(w'_t, z'_j)\right) \right].$$

Then, for any $t \geq 1$

$$\delta_{t+1} \leq \left\| \frac{1}{n} \sum_{i=1, i\neq j}^{n} \left[ (w_t - w'_t) - \alpha_t \left( \nabla f(w_t, z_i) - \nabla f(w'_t, z_i) \right) \right] \right\|$$

$$+ \frac{1}{n} \left\| (w_t - w'_t) - \alpha_t \left( \nabla f(w_t, z_j) - \nabla f(w'_t, z'_j) \right) \right\|$$

$$\leq \frac{n-1}{n} \eta \delta_t + \frac{1}{n} \min\{\eta, 1\}\delta_t + \frac{2\alpha_t L}{n}$$

$$\leq \eta \delta_t + \frac{2\alpha_t L}{n}.$$

Therefore, for full-batch ZO-GD, we have

$$w_{t+1} = w_t - \frac{\alpha_t}{n} \sum_{j=1}^{n} \hat{\nabla} f\left(w_t, z_j\right)$$

$$= \frac{1}{n} \sum_{i=1}^{n} (w_t - \alpha_t \nabla f(w_t, z_i)) + \frac{\alpha_t}{n} \sum_{i=1}^{n} \nabla f(w_t, z_i) - \frac{\alpha_t}{n} \sum_{i=1}^{n} \hat{\nabla} f(w_t, z_i),$$

$$w'_{t+1} = w'_t - \frac{\alpha_t}{n} \sum_{i=1, i \neq j}^{n} \hat{\nabla} f(w'_t, z_i) - \frac{\alpha_t}{n} \hat{\nabla} f(w'_t, z'_j)$$

$$= \frac{1}{n} \left[ \sum_{i=1, i \neq j}^{n} (w'_t - \alpha_t \nabla f(w'_t, z_i)) + \left( w'_t - \alpha_t \nabla f(w'_t, z'_j) \right) \right]$$

$$+ \frac{\alpha_t}{n} \sum_{i=1, i \neq j}^{n} \left( \nabla f(w'_t, z_i) - \hat{\nabla} f(w'_t, z_i) \right) + \frac{\alpha_t}{n} \left( \nabla f(w'_t, z'_j) - \hat{\nabla} f(w'_t, z'_j) \right).$$

Then, for any $t \geq 1$

$$\delta_{t+1} \leq \eta \delta_t + \frac{2\alpha_t L}{n}$$

$$+ \frac{\alpha_t}{n} \left\| \sum_{i=1}^{n} (\nabla f(w_t, z_i) - \hat{\nabla} f(w_t, z_i)) - \sum_{i=1, i \neq j}^{n} \left( \nabla f(w'_t, z_i) - \hat{\nabla} f(w'_t, z_i) \right) - \left( \nabla f(w'_t, z'_j) - \hat{\nabla} f(w'_t, z'_j) \right) \right\|$$

*Proof of Theorem 4.* Recall that $u_k^t$ are independent for all k$\leq$ K, t$\leq$T and the loss function is Lipschitz , smooth and $f(\cdot) \leq \frac{1}{2}$. The last display and the triangle inequality give

$$\delta_{t+1} \leq \eta \delta_t + \frac{2\alpha_t L}{n} + \frac{\alpha_t}{n} \Gamma_d^K \sum_{i=1, i \neq j}^{n} \|\nabla f(w_t, z_i) - \nabla f(w'_t, z_i)\|$$

$$+ \frac{\alpha_t}{n} \Gamma_d^K \|\nabla f(w_t, z_j) - \nabla f(w'_t, z'_j)\| + \frac{\alpha_t A_d}{n\mu} \sum_{i=1, i \neq j}^{n} \|f(w_t, z_i) - f(w'_t, z_i)\|$$

$$+ \frac{\alpha_t A_d}{n\mu} \|f(w_t, z_j) - f(w'_t, z'_j)\| + \alpha_t \mu \beta C_d$$

$$\leq \left[ 1 + \alpha_t \left( \left( \Gamma_d^K + 1 \right) \beta + \frac{A_d L}{\mu} \right) \right] \delta_t + \frac{2\alpha_t L \left( \Gamma_d^K + 1 \right)}{n} + \frac{A_d \alpha_t}{n\mu} + \mu \beta C_d \alpha_t$$

Then by solving the recursion we find

$$\delta_T \leq \sum_{t=1}^{T} \left( \frac{2\alpha_t L \left( \Gamma_d^K + 1 \right)}{n} + \frac{A_d \alpha_t}{n\mu} + \mu \beta C_d \alpha_t \right) \prod_{j=t+1}^{T} \left( 1 + \alpha_t \left( \left( \Gamma_d^K + 1 \right) \beta + \frac{A_d L}{\mu} \right) \right)$$

Under the choice $\alpha_t \leq \frac{C}{t \left( (\Gamma_d^K + 1)\beta + \frac{A_d L}{\mu} \right)}$ the last display gives

$$\delta_T \leq \left( \frac{2L \left( \Gamma_d^K + 1 \right)}{n} + \frac{A_d}{n\mu} + \mu \beta C_d \right) \sum_{t=1}^{T} \frac{C}{t \left( \left( \Gamma_d^K + 1 \right) \beta + \frac{A_d L}{\mu} \right)} \prod_{j=t+1}^{T} \left( 1 + \frac{C}{j} \right)$$

$$\leq \frac{\frac{2L(\Gamma_d^K + 1)}{n} + \frac{A_d}{n\mu} + \mu \beta C_d}{\left( \Gamma_d^K + 1 \right) \beta + \frac{A_d L}{\mu}} \sum_{t=1}^{T} \frac{C}{t} \prod_{j=t+1}^{T} \left( 1 + \frac{C}{j} \right)$$

$$\leq \frac{2L^2 + \beta + \beta L}{n \beta L} \sum_{t=1}^{T} \frac{C}{t} \exp \sum_{j=t+1}^{T} \frac{C}{j}$$

$$\leq \frac{2L^2 + \beta + \beta L}{n \beta L} \sum_{t=1}^{T} \frac{C}{t} \exp \left( C \log \left( \frac{eT}{t+1} \right) \right)$$

$$\leq \frac{C\left(2L^2 + \beta + \beta L\right)}{n\beta L} \sum_{t=1}^{T} \frac{1}{t} \left(\frac{eT}{t+1}\right)^{C}$$

$$\leq \frac{C\left(eT\right)^{C}\left(2L^2 + \beta + \beta L\right)}{n\beta L} \sum_{t=1}^{T} \frac{1}{t^{C+1}}$$

$$\leq \frac{\left(eT\right)^{C}\left(2L^2 + \beta + \beta L\right)}{n\beta L} \min\{C+1, C \log\left(eT\right)\}.$$

Then the generalization error of ZO-GD with 1-point gradient estimation is bounded by

$$\epsilon_{\text{gen}} \leq \frac{2L^2 + \beta + \beta L}{n\beta} \left(eT\right)^{C} \min\{C+1, C \log\left(eT\right)\}.$$

$\square$

**Theorem 14** (convex). *Assume that the loss $f(\cdot, z) \in [0,1]$ is L-Lipschitz, convex, $\beta$-smooth for all $z \in \mathcal{Z}$. Consider the update rule Eq.equation 13 with $T$ the total number of iterates, $\alpha_t \leq C/t\left(\beta(\Gamma_d^k + 1) + LA_d/\mu\right)$ for some (fixed) $C > 0$ and for all $t \leq T$. Then the generalization error of ZO-GD with 1-point gradient estimation is bounded by*

$$\epsilon_{gen} \leq \min\{C+1, C\log(eT)\}(2L^2 + \beta + \beta L)(eT)^{C}/(n\beta).$$

*Proof of Theorem 14.* Recall that $\eta = 1$, $u_k^t$ are independent for all k$\leq$ K, t$\leq$T and the loss function is Lipschitz , smooth and $f(\cdot) \leq \frac{1}{2}$. The last display and the triangle inequality give

$$\delta_{t+1} \leq \eta\delta_t + \frac{2\alpha_t L}{n} + \frac{\alpha_t}{n}\Gamma_d^K \sum_{i=1, i\neq j}^{n} \|\nabla f(w_t, z_i) - \nabla f(w_t', z_i)\| + \frac{\alpha_t}{n}\Gamma_d^K \|\nabla f(w_t, z_j) - \nabla f(w_t', z_j')\|$$

$$+ \frac{\alpha_t A_d}{n\mu} \sum_{i=1, i\neq j}^{n} \|f(w_t, z_i) - f(w_t', z_i)\| + \frac{\alpha_t A_d}{n\mu} \|f(w_t, z_j) - f(w_t', z_j')\| + \alpha_t \mu\beta C_d$$

$$\leq \left[1 + \alpha_t\left(\Gamma_d^K \beta + \frac{A_d L}{\mu}\right)\right]\delta_t + \frac{2\alpha_t L\left(\Gamma_d^K + 1\right)}{n} + \frac{A_d \alpha_t}{n\mu} + \mu\beta C_d \alpha_t$$

Then by solving the recursion we find

$$\delta_T \leq \sum_{t=1}^{T} \left(\frac{2\alpha_t L\left(\Gamma_d^K + 1\right)}{n} + \frac{A_d \alpha_t}{n\mu} + \mu\beta C_d \alpha_t\right) \prod_{j=t+1}^{T} \left(1 + \alpha_t\left(\Gamma_d^K \beta + \frac{A_d L}{\mu}\right)\right)$$

Under the choice $\alpha_t \leq \frac{C}{t\left((\Gamma_d^K + 1)\beta + \frac{A_d L}{\mu}\right)}$ the last display gives

$$\delta_T \leq \left(\frac{2L\left(\Gamma_d^K + 1\right)}{n} + \frac{A_d}{n\mu} + \mu\beta C_d\right) \sum_{t=1}^{T} \frac{C}{t\left(\left(\Gamma_d^K + 1\right)\beta + \frac{A_d L}{\mu}\right)} \prod_{j=t+1}^{T} \left(1 + \frac{C}{j}\right)$$

$$\leq \frac{\frac{2L\left(\Gamma_d^K + 1\right)}{n} + \frac{A_d}{n\mu} + \mu\beta C_d}{\left(\Gamma_d^K + 1\right)\beta + \frac{A_d L}{\mu}} \sum_{t=1}^{T} \frac{C}{t} \prod_{j=t+1}^{T} \left(1 + \frac{C}{j}\right)$$

$$\leq \frac{2L^2 + \beta + \beta L}{n\beta L} \sum_{t=1}^{T} \frac{C}{t} \exp \sum_{j=t+1}^{T} \frac{C}{j}$$

$$\leq \frac{2L^2 + \beta + \beta L}{n\beta L} \sum_{t=1}^{T} \frac{C}{t} \exp\left(C\log\left(\frac{eT}{t+1}\right)\right)$$

$$\leq \frac{C\left(2L^2 + \beta + \beta L\right)}{n\beta L} \sum_{t=1}^{T} \frac{1}{t} \left(\frac{eT}{t+1}\right)^{C}$$

$$\leq \frac{C\,(eT)^C\,\left(2L^2+\beta+\beta L\right)}{n\beta L}\sum_{t=1}^{T}\frac{1}{t^{C+1}}$$

$$\leq \frac{(eT)^C\,\left(2L^2+\beta+\beta L\right)}{n\beta L}\min\{C+1, C\log(eT)\}.$$

Then the generalization error of ZO-GD with 1-point gradient estimation is bounded by

$$\epsilon_{\text{gen}} \leq \frac{2L^2+\beta+\beta L}{n\beta}\,(eT)^C\min\{C+1, C\log(eT)\}.$$

$\square$

**Theorem 15** (strongly convex)**.** *Assume that the loss $f(\cdot, z) \in [0,1]$ is $L$-Lipschitz, $\gamma$-strongly convex, $\beta$-smooth for all $z \in \mathcal{Z}$. Consider the update rule Eq.equation 13 with $T$ the total number of iterates, $\alpha_t \leq C/t\big(\beta(\Gamma_d^k+1)+LA_d/\mu\big)$ for some (fixed) $C>0$ and for all $t \leq T$. Then the generalization error of ZO-GD with 1-point gradient estimation is bounded by*

$$\epsilon_{gen} \leq \min\{C+1, C\log(eT)\}(2L^2+\beta+\beta L)(eT)^C/(n\beta).$$

*Proof of Theorem 15.* Recall that $\eta = 1 - \frac{\alpha_t\beta\gamma}{\beta+\gamma}$, $u_k^t$ are independent for all $k \leq K$, $t \leq T$ and the loss function is Lipschitz , smooth and $f(\cdot) \leq \frac{1}{2}$. The last display and the triangle inequality give

$$\delta_{t+1} \leq \eta\delta_t + \frac{2\alpha_t L}{n} + \frac{\alpha_t}{n}\Gamma_d^K\sum_{i=1, i\neq j}^{n}\|\nabla f(w_t, z_i) - \nabla f(w_t', z_i)\|$$

$$+ \frac{\alpha_t}{n}\Gamma_d^K\left\|\nabla f(w_t, z_j) - \nabla f(w_t', z_j')\right\| + \frac{\alpha_t A_d}{n\mu}\sum_{i=1, i\neq j}^{n}\|f(w_t, z_i) - f(w_t', z_i)\|$$

$$+ \frac{\alpha_t A_d}{n\mu}\left\|f(w_t, z_j) - f(w_t', z_j')\right\| + \alpha_t\mu\beta C_d$$

$$\leq \left[1 + \alpha_t\left(\Gamma_d^K\beta - \frac{\beta\gamma}{\beta+\gamma} + \frac{A_d L}{\mu}\right)\right]\delta_t + \frac{2\alpha_t L\left(\Gamma_d^K+1\right)}{n} + \frac{A_d\alpha_t}{n\mu} + \mu\beta C_d\alpha_t$$

Then by solving the recursion we find

$$\delta_T \leq \sum_{t=1}^{T}\left(\frac{2\alpha_t L\left(\Gamma_d^K+1\right)}{n} + \frac{A_d\alpha_t}{n\mu} + \mu\beta C_d\alpha_t\right)\prod_{j=t+1}^{T}\left(1 + \alpha_t\left(\Gamma_d^K\beta - \frac{\beta\gamma}{\beta+\gamma} + \frac{A_d L}{\mu}\right)\right)$$

Under the choice $\alpha_t \leq \frac{C}{t\left(\left(\Gamma_d^K+1\right)\beta + \frac{A_d L}{\mu}\right)}$ the last display gives

$$\delta_T \leq \left(\frac{2L\left(\Gamma_d^K+1\right)}{n} + \frac{A_d}{n\mu} + \mu\beta C_d\right)\sum_{t=1}^{T}\frac{C}{t\left(\left(\Gamma_d^K+1\right)\beta + \frac{A_d L}{\mu}\right)}\prod_{j=t+1}^{T}\left(1 + \frac{C}{j}\right)$$

$$\leq \frac{\frac{2L\left(\Gamma_d^K+1\right)}{n} + \frac{A_d}{n\mu} + \mu\beta C_d}{\left(\Gamma_d^K+1\right)\beta + \frac{A_d L}{\mu}}\sum_{t=1}^{T}\frac{C}{t}\prod_{j=t+1}^{T}\left(1 + \frac{C}{j}\right)$$

$$\leq \frac{2L^2+\beta+\beta L}{n\beta L}\sum_{t=1}^{T}\frac{C}{t}\exp\sum_{j=t+1}^{T}\frac{C}{j}$$

$$\leq \frac{2L^2+\beta+\beta L}{n\beta L}\sum_{t=1}^{T}\frac{C}{t}\exp\left(C\log\left(\frac{eT}{t+1}\right)\right)$$

$$\leq \frac{C\left(2L^2+\beta+\beta L\right)}{n\beta L}\sum_{t=1}^{T}\frac{1}{t}\left(\frac{eT}{t+1}\right)^C$$

$$\leq \frac{C\,(eT)^C\,\left(2L^2 + \beta + \beta L\right)}{n\beta L} \sum_{t=1}^{T} \frac{1}{t^{C+1}}$$

$$\leq \frac{(eT)^C\,\left(2L^2 + \beta + \beta L\right)}{n\beta L} \min\{C+1, C\log(eT)\}.$$

Then the generalization error of ZO-GD with 1-point gradient estimation is bounded by

$$\epsilon_{\text{gen}} \leq \frac{2L^2 + \beta + \beta L}{n\beta}\,(eT)^C \min\{C+1, C\log(eT)\}.$$

$\square$

*Proof of Theorem 5.* Recall that $u_k^t$ are independent for all $k \leq K$, $t \leq T$ and the loss function is Lipschitz and smooth . The last display and the triangle inequality give

$$\delta_{t+1} \leq \eta\delta_t + \frac{2\alpha_t L}{n} + \frac{\alpha_t}{n}\Gamma_d^K \sum_{i=1, i\neq j}^{n} \|\nabla f(w_t, z_i) - \nabla f(w_t', z_i)\|$$

$$+ \frac{\alpha_t}{n}\Gamma_d^K \left\|\nabla f(w_t, z_j) - \nabla f(w_t', z_j')\right\| + \alpha_t \mu\beta C_d$$

$$\leq \left[1 + \alpha_t\left(\Gamma_d^K + 1\right)\beta\right]\delta_t + \frac{2\alpha_t L\left(\Gamma_d^K + 1\right)}{n} + \mu\beta C_d \alpha_t$$

Then by solving the recursion we find

$$\delta_T \leq \sum_{t=1}^{T} \left(\frac{2\alpha_t L\left(\Gamma_d^K + 1\right)}{n} + \mu\beta C_d \alpha_t\right) \prod_{j=t+1}^{T}\left(1 + \alpha_j\left(\Gamma_d^K + 1\right)\beta\right)$$

Under the choice $\alpha_t \leq \frac{C}{t\left(\Gamma_d^K+1\right)\beta}$ the last display gives

$$\delta_T \leq \left(\frac{2L}{n}\Gamma_K^d + \mu\beta C_d\right) \sum_{t=1}^{T} \alpha_t \prod_{j=t+1}^{T}\left(1 + \beta\alpha_j\left(1 + \Gamma_K^d\right)\right)$$

$$\leq \frac{\Gamma_K^d + 1}{n}(2+c)L \sum_{t=1}^{T} \frac{C}{t\left(\Gamma_K^d + 1\right)} \prod_{j=t+1}^{T}\left(1 + \frac{C\beta}{j}\right)$$

$$\leq \frac{(2+c)L}{n} \sum_{t=1}^{T} \frac{C}{t} \exp\left(\sum_{j=t+1}^{T} \frac{C\beta}{j}\right) \leq \frac{(2+c)L}{n} \sum_{t=1}^{T} \frac{C}{t} \exp\left(C\beta \log\left(\frac{eT}{t+1}\right)\right)$$

$$= \frac{C(2+c)L}{n} \sum_{t=1}^{T} \frac{1}{t}\left(\frac{eT}{t+1}\right)^{C\beta} \leq \frac{C(eT)^{C\beta}(2+c)L}{n} \sum_{t=1}^{T} \frac{1}{t^{C\beta+1}}$$

$$\leq \frac{(eT)^{C\beta}(2+c)L}{n} \min\left\{\frac{C\beta+1}{\beta}, C\log(eT)\right\}.$$

Then the generalization error of ZO-GD with 2-point gradient estimation is bounded by

$$\epsilon_{\text{gen}} \leq \frac{(eT)^{C\beta}(2+c)L^2}{n} \min\left\{\frac{C\beta+1}{\beta}, C\log(eT)\right\}.$$

$\square$

**Theorem 16** (convex). *Assume that the loss $f(\cdot, z) \in [0, 1]$ is L-Lipschitz, convex and $\beta$-smooth for all $z \in \mathcal{Z}$. Consider the update rule Eq.equation 14 with $T$ the total number of iterates, $\alpha_t \leq C/\left(\Gamma_d^k + 1\right)t$ for some (fixed) $C > 0$ and for all $t \leq T$. Then the generalization error of ZO-GD with 2-point gradient estimation is bounded by*

$$\epsilon_{gen} \leq \frac{(2+c)L^2}{n}\,(eT)^{C\beta} \min\{\frac{C\beta+1}{\beta}, C\log(eT)\}.$$

*Proof of Theorem 16.* Recall that $\eta = 1$, $u_k^t$ are independent for all $k \leq K$, $t \leq T$ and the loss function is Lipschitz and smooth. The last display and the triangle inequality give

$$
\delta_{t+1} \leq \eta \delta_t + \frac{2\alpha_t L}{n} + \frac{\alpha_t}{n} \Gamma_d^K \sum_{i=1, i \neq j}^{n} \|\nabla f(w_t, z_i) - \nabla f(w_t', z_i)\|
$$
$$
+ \frac{\alpha_t}{n} \Gamma_d^K \left\| \nabla f(w_t, z_j) - \nabla f(w_t', z_j') \right\| + \alpha_t \mu \beta C_d
$$
$$
\leq \left(1 + \alpha_t \Gamma_d^K \beta \right) \delta_t + \frac{2\alpha_t L \left( \Gamma_d^K + 1 \right)}{n} + \mu \beta C_d \alpha_t
$$

Then by solving the recursion we find

$$
\delta_T \leq \sum_{t=1}^{T} \left( \frac{2\alpha_t L \left( \Gamma_d^K + 1 \right)}{n} + \mu \beta C_d \alpha_t \right) \prod_{j=t+1}^{T} \left( 1 + \alpha_j \Gamma_d^K \beta \right)
$$

Under the choice $\alpha_t \leq \frac{C}{t(\Gamma_d^K + 1)\beta}$ the last display gives

$$
\delta_T \leq \left( \frac{2L}{n} \Gamma_K^d + \mu \beta C_d \right) \sum_{t=1}^{T} \alpha_t \prod_{j=t+1}^{T} \left( 1 + \beta \alpha_j \Gamma_d^K \right)
$$
$$
\leq \frac{\Gamma_K^d + 1}{n} (2 + c) L \sum_{t=1}^{T} \frac{C}{t \left( \Gamma_K^d + 1 \right)} \prod_{j=t+1}^{T} \left( 1 + \frac{C\beta}{j} \right)
$$
$$
\leq \frac{(2+c)L}{n} \sum_{t=1}^{T} \frac{C}{t} \exp \left( \sum_{j=t+1}^{T} \frac{C\beta}{j} \right) \leq \frac{(2+c)L}{n} \sum_{t=1}^{T} \frac{C}{t} \exp \left( C\beta \log \left( \frac{eT}{t+1} \right) \right)
$$
$$
= \frac{C(2+c)L}{n} \sum_{t=1}^{T} \frac{1}{t} \left( \frac{eT}{t+1} \right)^{C\beta} \leq \frac{C(eT)^{C\beta}(2+c)L}{n} \sum_{t=1}^{T} \frac{1}{t^{C\beta+1}}
$$
$$
\leq \frac{(eT)^{C\beta}(2+c)L}{n} \min \left\{ \frac{C\beta+1}{\beta}, C \log(eT) \right\}.
$$

Then the generalization error of ZO-GD with 2-point gradient estimation is bounded by

$$
\epsilon_{\text{gen}} \leq \frac{(eT)^{C\beta}(2+c)L^2}{n} \min \left\{ \frac{C\beta+1}{\beta}, C \log(eT) \right\}.
$$

$\square$

**Theorem 17** (strongly convex). *Assume that the loss $f(\cdot, z) \in [0,1]$ is L-Lipschitz, $\gamma$-strongly convex and $\beta$-smooth for all $z \in \mathcal{Z}$. Consider the update rule Eq.equation 14 with $T$ the total number of iterates, $\alpha_t \leq C / \left( \Gamma_d^k + 1 \right) t$ for some (fixed) $C > 0$ and for all $t \leq T$. Then the generalization error of ZO-GD with 2-point gradient estimation is bounded by*

$$
\epsilon_{gen} \leq \frac{(2+c)L^2}{n} (eT)^{C\beta} \min \{ \frac{C\beta+1}{\beta}, C \log(eT) \}.
$$

*Proof of Theorem 16.* Recall that $\eta = 1 - \frac{\alpha_t \beta \gamma}{\beta + \gamma}$, $u_k^t$ are independent for all $k \leq K$, $t \leq T$ and the loss function is Lipschitz and smooth. The last display and the triangle inequality give

$$
\delta_{t+1} \leq \eta \delta_t + \frac{2\alpha_t L}{n} + \frac{\alpha_t}{n} \Gamma_d^K \sum_{i=1, i \neq j}^{n} \|\nabla f(w_t, z_i) - \nabla f(w_t', z_i)\|
$$
$$
+ \frac{\alpha_t}{n} \Gamma_d^K \left\| \nabla f(w_t, z_j) - \nabla f(w_t', z_j') \right\| + \alpha_t \mu \beta C_d
$$
$$
\leq \left(1 + \alpha_t \Gamma_d^K \beta - \frac{\alpha_t \beta \gamma}{\beta + \gamma} \right) \delta_t + \frac{2\alpha_t L \left( \Gamma_d^K + 1 \right)}{n} + \mu \beta C_d \alpha_t
$$

Then by solving the recursion we find

$$\delta_T \leq \sum_{t=1}^{T} \left( \frac{2\alpha_t L \left( \Gamma_d^K + 1 \right)}{n} + \mu\beta C_d \alpha_t \right) \prod_{j=t+1}^{T} \left( 1 + \alpha_j \Gamma_d^K - \frac{\alpha_t \beta \gamma}{\beta + \gamma} \beta \right)$$

Under the choice $\alpha_t \leq \frac{C}{t(\Gamma_d^K + 1)\beta}$ the last display gives

$$\delta_T \leq \left( \frac{2L}{n} \Gamma_K^d + \mu\beta C_d \right) \sum_{t=1}^{T} \alpha_t \prod_{j=t+1}^{T} \left( 1 + \beta\alpha_j \Gamma_d^K - \frac{\alpha_t \beta \gamma}{\beta + \gamma} \right)$$

$$\leq \left( \frac{2L}{n} \Gamma_K^d + \mu\beta C_d \right) \sum_{t=1}^{T} \alpha_t \prod_{j=t+1}^{T} \left( 1 + \beta\alpha_j (\Gamma_d^K + 1) \right)$$

$$\leq \frac{\Gamma_K^d + 1}{n} (2 + c) L \sum_{t=1}^{T} \frac{C}{t \left( \Gamma_K^d + 1 \right)} \prod_{j=t+1}^{T} \left( 1 + \frac{C\beta}{j} \right)$$

$$\leq \frac{(2 + c) L}{n} \sum_{t=1}^{T} \frac{C}{t} \exp\left( \sum_{j=t+1}^{T} \frac{C\beta}{j} \right) \leq \frac{(2 + c) L}{n} \sum_{t=1}^{T} \frac{C}{t} \exp\left( C\beta \log\left( \frac{eT}{t+1} \right) \right)$$

$$= \frac{C(2 + c) L}{n} \sum_{t=1}^{T} \frac{1}{t} \left( \frac{eT}{t+1} \right)^{C\beta} \leq \frac{C(eT)^{C\beta}(2 + c) L}{n} \sum_{t=1}^{T} \frac{1}{t^{C\beta + 1}}$$

$$\leq \frac{(eT)^{C\beta}(2 + c) L}{n} \min\left\{ \frac{C\beta + 1}{\beta}, C \log(eT) \right\}.$$

Then the generalization error of ZO-GD with 2-point gradient estimation is bounded by

$$\epsilon_{\text{gen}} \leq \frac{(eT)^{C\beta}(2 + c) L^2}{n} \min\left\{ \frac{C\beta + 1}{\beta}, C \log(eT) \right\}.$$

$\square$

*Proof of Theorem 6.* Recall that $t \leq T$ and the loss function is Lipschitz and smooth . The last display and the triangle inequality give

$$\delta_{t+1} \leq \eta\delta_t + \frac{2\alpha_t L}{n} + \mu\beta\alpha_t.$$

In this case we can use the 1-expansivity of the update rule $\tilde{G}_t$ which uses the fact that the objective function is convex, $\alpha_t \leq 2/\beta$ and $\mu \leq \frac{c}{n}$. Hence, we have

$$\mathbb{E}\left[\delta_{t+1}\right] \leq \mathbb{E}\left[\delta_t\right] + \frac{2\alpha_t L + \alpha_t \beta c}{n}.$$

Unraveling the recursion gives

$$\mathbb{E}\left[\delta_T\right] \leq \frac{2L + \beta c}{n} \sum_{t=1}^{T} \alpha_t.$$

Then the generalization error of ZO-GD with coordinate gradient estimation is bounded by

$$\epsilon_{gen} \leq \frac{2L^2 + \beta cL}{n} \sum_{t=1}^{T} \alpha_t.$$

In this case we can use the $(1 - \alpha\gamma)$-expansivity of the update rule $\tilde{G}_t$ which uses the fact that the objective function is $\gamma$-strongly convex , $\alpha_t \leq 1/\beta$ and $\mu \leq \frac{c}{n}$. Hence, we have

$$\mathbb{E}\left[\delta_{t+1}\right] \leq (1 - \alpha\gamma) \mathbb{E}\left[\delta_t\right] + \frac{2\alpha_t L + \alpha_t \beta c}{n}.$$

Unraveling the recursion gives

$$\mathbb{E}\left[\delta_T\right] \leq \frac{2\alpha L + \alpha\beta c}{n} \sum_{t=0}^{T} (1-\alpha\gamma)^t \leq \frac{2L + \beta c}{n\gamma}.$$

Then the generalization error of ZO-GD with coordinate gradient estimation is bounded by

$$\epsilon_{gen} \leq \frac{2L^2 + \beta c L}{n\gamma}.$$

In this case we can use the $(1 + \alpha_t\beta)$-expansivity of the update rule $\tilde{G}_t$ which uses the fact that the objective function is Lipschitz, smooth and $\mu \leq \frac{cL}{n\beta}$. Hence, we have

$$\mathbb{E}\left[\delta_{t+1}\right] \leq (1 + \alpha_t\beta)\,\mathbb{E}\left[\delta_t\right] + \frac{2\alpha_t L + \alpha_t\beta c}{n}$$

Then by solving the recursion we find

$$\delta_T \leq \sum_{t=1}^{T} \left(\frac{2\alpha_t L}{n} + \frac{\alpha_t\beta c}{n}\right) \prod_{j=t+1}^{T} (1 + \alpha_j\beta).$$

Under the choice $\alpha_t \leq \frac{C}{t}$ the last display gives

$$\delta_T \leq \frac{(2+c)L}{n} \sum_{t=1}^{T} \alpha_t \prod_{j=t+1}^{T} (1 + \beta\alpha_j) \leq \frac{(2+c)L}{n} \sum_{t=1}^{T} \frac{C}{t} \prod_{j=t+1}^{T} \left(1 + \frac{C\beta}{j}\right)$$

$$\leq \frac{(2+c)L}{n} \sum_{t=1}^{T} \frac{C}{t} \exp\left(\sum_{j=t+1}^{T} \frac{C\beta}{j}\right) \leq \frac{(2+c)L}{n} \sum_{t=1}^{T} \frac{C}{t} \exp\left(C\beta \log\left(\frac{eT}{t+1}\right)\right)$$

$$= \frac{C(2+c)L}{n} \sum_{t=1}^{T} \frac{1}{t} \left(\frac{eT}{t+1}\right)^{C\beta} \leq \frac{C(eT)^{C\beta}(2+c)L}{n} \sum_{t=1}^{T} \frac{1}{t^{C\beta+1}}$$

$$\leq \frac{(eT)^{C\beta}(2+c)L}{n} \min\left\{\frac{C\beta+1}{\beta}, C\log(eT)\right\}.$$

Then the generalization error of ZO-GD with coordinate gradient estimation is bounded by

$$\epsilon_{\text{gen}} \leq \frac{(eT)^{C\beta}(2+c)L^2}{n} \min\left\{\frac{C\beta+1}{\beta}, C\log(eT)\right\}.$$

$\square$

## I    PROOF OF SVRG GENERALIZATION ANALYSIS

### I.1    SVRG

**Theorem 18 (Convex Case).** *Assume that the loss function $f(\cdot; z)$ is $\beta$-smooth, convex and L-Lipschitz for every $z$. Suppose that we run SVRG with step sizes $\alpha_t \leq c/ms$ for $m$ steps. Then, at stage S we have*

$$\mathbb{E}[\delta_S] \leq \frac{6cL}{ns} + \left(\frac{2\beta c}{s} + 1\right)\mathbb{E}[\delta_{S-1}].$$

*After S stages, SVRG satisfies uniform stability with*

$$\epsilon_{gen} \leq \frac{6L\,(eS)^{2\beta c}}{n} \min\left\{c + \beta^{-1}, c\log(eS)\right\}.$$

**Theorem 19 (Strongly Convex Case).** *Assume that the loss function $f(\cdot; z)$ is $\beta$-smooth,$\gamma$-strongly convex and L-Lipschitz for every $z$. Suppose that we run SVRG with step sizes $\alpha_t \leq 1/\beta$ for m steps. Then, at one stage we have*

$$\mathbb{E}[\delta_t + 1] \leq (1 - \alpha\gamma)\mathbb{E}[\delta_t] + 2\alpha\beta[\delta_1] + \frac{6L\alpha_t}{n}.$$

*After S stages, SVRG satisfies uniform stability with*

$$\epsilon_{gen} \leq \frac{6SL^2}{n\gamma}.$$

**Theorem 20 (Nonconvex Case).** *Assume that the loss function $f(\cdot; z)$ is $\beta$-smooth, and L-Lipschitz for every $z$. Suppose that we run SVRG with step sizes $\alpha_t \leq C/ms$ for m steps. Then, at stage S we have*

$$\mathbb{E}[\delta_S] \leq \exp\frac{C\beta}{s}\left(1 + \frac{2C\beta}{s}\right)\mathbb{E}[\delta_{S-1}] + \frac{6CL}{ns}\exp\frac{C\beta}{s}.$$

*After K stages, SVRG satisfies uniform stability with*

$$\epsilon_{gen} \leq \frac{2L^2 e^{C\beta} S^{3C\beta}}{n\beta}.$$

At stage $S$, we have

$$\mathbb{E}[\delta_{t+1}] \leq \|w_t - w_t' - \alpha_t\left(\nabla f\left(w_t, z_t\right) - \nabla f\left(w_t', z_t\right)\right)\| + \alpha_t\|\nabla f\left(w_1, z_t\right) - \nabla f\left(w_1', z_t\right)\|$$

$$+ \alpha_t\left\|\frac{1}{n}\sum_{i=1,i\neq j}^n \nabla f\left(w_1, z_i\right) - \frac{1}{n}\sum_{i=1,i\neq j}^n \nabla f\left(w_1', z_i\right)\right\|$$

$$+ \frac{\alpha_t}{n}\|\nabla f\left(w_1, z_j\right) - \nabla f\left(w_1', z_j'\right)\|$$

$$\leq \eta\mathbb{E}[\delta_t] + \frac{2L\alpha_t}{n} + \alpha_t\beta\mathbb{E}[\delta_1] + \frac{2L\alpha_t}{n} + \alpha_t\beta\mathbb{E}[\delta_1] + \frac{2L\alpha_t}{n}$$

$$= \eta\mathbb{E}[\delta_t] + 2\alpha_t\beta\mathbb{E}[\delta_1] + \frac{6L\alpha_t}{n}.$$

*Proof of Theorem 18.*

$$\mathbb{E}[\delta_{t+1}] \leq \mathbb{E}[\delta_t] + 2\alpha_t\beta\mathbb{E}[\delta_1] + \frac{6L\alpha_t}{n}$$

$$\leq \frac{6L}{n}\sum_{t=1}^m \alpha_t + \left(2\beta\sum_{n=1}^m \alpha_t + 1\right)\mathbb{E}[\delta_1] \leq \frac{6L}{n}\sum_{t=1}^m \frac{C}{ms} + \left(2\beta\sum_{n=1}^m \frac{C}{ms} + 1\right)\mathbb{E}[\delta_1]$$

$$\leq \frac{6CL}{ns} + \left(\frac{2\beta C}{s} + 1\right)\mathbb{E}[\delta_1].$$

For the $S$-stage, we have $w_s = w_m$ and $w_{S-1} = w_1$. Then

$$\mathbb{E}[\delta_S] \leq \frac{6CL}{ns} + \left(\frac{2\beta C}{s} + 1\right)\mathbb{E}[\delta_{S-1}].$$

Summing the above inequality for $S$ stages, and unraveling the recursion gives

$$\mathbb{E}[\delta_S] \leq \frac{6CL}{n}\sum_{s=1}^S \frac{1}{s}\prod_{j=s+1}^S \left(1 + \frac{2\beta C}{j}\right) \leq \frac{6L\left(eS\right)^{2\beta C}}{n}\min\left\{C + \beta^{-1}, C\log(eS)\right\}.$$

Note that $\mathbb{E}\left|f\left(w_T; z\right) - f\left(w_T'; z\right)\right| \leq L\mathbb{E}[\delta_T]$, then we have

$$\epsilon_{gen} \leq \frac{6L^2\left(eS\right)^{2\beta C}}{n}\min\left\{C + \beta^{-1}, C\log(eS)\right\}.$$

$\square$

*Proof of Theorem 19.*

$$\mathbb{E}[\delta_t + 1] \leq (1 - \alpha\gamma)\mathbb{E}[\delta_t] + 2\alpha\beta[\delta_1] + \frac{6L\alpha}{n}$$

$$= (1 - \alpha\gamma)^m \mathbb{E}[\delta_1] + 2\alpha\beta \sum_{t=1}^{m-1}(1 - \alpha\gamma)^t \mathbb{E}[\delta_1] + \frac{6L\alpha}{n}\sum_{t=0}^{m-1}(1 - \alpha\gamma)^t$$

$$\leq \frac{6L\alpha}{n}\left(\frac{1 - (1-\alpha\gamma)^m}{\alpha\gamma}\right) + \left((1 - (1-\alpha\gamma)^m)\frac{2\beta}{\gamma} + (1-\alpha\gamma)^m\right)\mathbb{E}[\delta_1]$$

$$\leq \mathbb{E}[\delta_1] + \frac{6L}{n\gamma}.$$

For the $S$-stage, we have $w_s = w_m$ and $w_{S-1} = w_1$. Then

$$\mathbb{E}[\delta_S] \leq \mathbb{E}[\delta_{S-1}] + \frac{6L}{n\gamma}.$$

Summing the above inequality for $S$ stages, and unraveling the recursion gives

$$\mathbb{E}[\delta_S] \leq \frac{6SL}{n\gamma}.$$

Note that $\mathbb{E}\left|f(w_T; z) - f(w_T'; z)\right| \leq L\mathbb{E}[\delta_T]$, then we have

$$\epsilon_{\text{gen}} \leq \frac{6SL^2}{n\gamma}.$$

$\square$

*Proof of Theorem 20.*

$$\mathbb{E}[\delta_{t+1}] \leq (1 + \alpha_t\beta)\mathbb{E}[\delta_t] + 2\alpha_t\beta\mathbb{E}[\delta_1] + \frac{6L\alpha_t}{n}$$

$$\leq \left(1 + \frac{C\beta}{sm}\right)\mathbb{E}[\delta_t] + \frac{2C\beta}{sm}\mathbb{E}[\delta_1] + \frac{6CL}{nsm}$$

$$= \left(1 + \frac{C\beta}{sm}\right)^m \mathbb{E}[\delta_1] + \frac{2C\beta}{sm}\sum_{t=0}^{m}\left(1 + \frac{C\beta}{sm}\right)^t \delta_1 + \frac{6LC}{nsm}\sum_{t=0}^{m}\left(1 + \frac{C\beta}{sm}\right)^t$$

$$\leq \exp\frac{C\beta}{s}\mathbb{E}[\delta_1] + \frac{2C\beta}{s}\exp\frac{C\beta}{s}\mathbb{E}[\delta_1] + \frac{6CL}{ns}\exp\frac{C\beta}{s}$$

$$= \exp\frac{C\beta}{s}\left(1 + \frac{2C\beta}{s}\right)\mathbb{E}[\delta_1] + \frac{6CL}{ns}\exp\frac{C\beta}{s}.$$

For the $S$-stage, we have $w_s = w_m$ and $w_{S-1} = w_1$. Then,

$$\mathbb{E}[\delta_S] \leq \exp\frac{C\beta}{s}\left(1 + \frac{2C\beta}{s}\right)\mathbb{E}[\delta_{S-1}] + \frac{6CL}{ns}\exp\frac{C\beta}{s}.$$

Summing the above inequality for $S$ stages, and unraveling the recursion gives

$$\mathbb{E}[\delta_S] \leq \frac{6LCe^{C\beta}}{n}\sum_{s=1}^{S}\frac{1}{s}\prod_{j=s+1}^{S}\exp\frac{3C\beta}{j}$$

$$\leq \frac{6LCe^{3C\beta}S^{C\beta}}{n}\sum_{s=1}^{S}\frac{1}{s}\frac{1}{(s+1)^{3C\beta}}$$

$$\leq \frac{6LCe^{3C\beta}S^{C\beta}}{n}\sum_{s=1}^{S}\frac{1}{s^{1+3C\beta}}$$

$$\leq \frac{2Le^{3C\beta}S^{3C\beta}}{n\beta}.$$

Note that $\mathbb{E}\left|f\left(w_T; z\right) - f\left(w_T'; z\right)\right| \le L\mathbb{E}\left[\delta_T\right]$, then we have

$$\epsilon_{\text{gen}} \le \frac{2L^2 e^{C\beta} S^{3C\beta}}{n\beta}.$$

$\square$

### I.2   ZO-SVRG

*Proof of Theorem 7.* Recall that $\eta = 1 + \alpha_t\beta$, and then we have

$$
\begin{aligned}
\mathbb{E}[\delta_{t+1}] &\le \left(1 + \alpha_t\beta\left(\Gamma_d^k + 1\right) + \frac{\alpha_t LA_d}{\mu}\right)\mathbb{E}[\delta_t] + 2\left(\alpha_t\beta\left(\Gamma_d^k + 1\right) + \frac{\alpha_t LA_d}{\mu}\right)\mathbb{E}[\delta_1] \\
&\quad + \frac{6L\left(\Gamma_d^k + 1\right)\alpha_t}{n} + 3\mu\beta C_d\alpha_t + \frac{A_d}{\mu n} \\
&\le \left(1 + \frac{C}{sm}\right)\mathbb{E}[\delta_t] + \frac{2C}{sm}\mathbb{E}[\delta_1] + \frac{6L^2 C + 3\beta c + 3\beta LC}{n\beta Lsm} \\
&\le \left(1 + \frac{C}{sm}\right)^m\mathbb{E}[\delta_1] + \frac{2C}{sm}\sum_{t=0}^{m}\left(1 + \frac{C}{sm}\right)^t\delta_1 \\
&\quad + \frac{6L^2 C + 3\beta C + 3\beta LC}{n\beta Lsm}\sum_{t=0}^{m}\left(1 + \frac{C}{sm}\right)^t \\
&\le \exp\frac{C}{s}\left(1 + \frac{2C}{s}\right)\mathbb{E}[\delta_1] + \frac{6L^2 + 3\beta + 3\beta L}{n\beta Ls}\exp\frac{C}{s}.
\end{aligned}
$$

For the $S$-stage, we have $w_s = w_m$ and $w_{S-1} = w_1$. Then,

$$\mathbb{E}[\delta_S] \le \exp\frac{C}{s}\left(1 + \frac{2C}{s}\right)\mathbb{E}[\delta_{S-1}] + \frac{6L^2 + 3\beta + 3\beta L}{n\beta Ls}\exp\frac{C}{s}.$$

Summing the above inequality for $S$ stages, and unraveling the recursion gives

$$
\begin{aligned}
\mathbb{E}[\delta_S] &\le \frac{6L^2 + 3\beta + 3\beta L}{n\beta L}\sum_{s=1}^{S}\frac{1}{s}\prod_{j=s+1}^{S}\exp\frac{3C}{j} \\
&\le \frac{\left(6L^2 + 3\beta + 3\beta L\right)e^C S^{3C}}{n\beta L}\sum_{s=1}^{S}\frac{1}{s}\frac{1}{(s+1)^{3C}} \\
&\le \frac{\left(6L^2 + 3\beta + 3\beta L\right)e^C S^{3C}}{n\beta L}\sum_{s=1}^{S}\frac{1}{s^{1+3C}} \\
&\le \frac{\left(2L^2 + \beta + \beta L\right)e^C S^{3C}}{n\beta LC}.
\end{aligned}
$$

Note that $\mathbb{E}\left|f\left(w_T; z\right) - f\left(w_T'; z\right)\right| \le L\mathbb{E}\left[\delta_T\right]$, then we have

$$\epsilon_{\text{gen}} \le \frac{\left(2L^2 + \beta + \beta L\right)e^C S^{3C}}{n\beta C}.$$

$\square$

**Theorem 21** (convex). *Assume that the loss function $f(\cdot; z)$ is $\beta$-smooth, convex, and L-Lipschitz for every $z$. Suppose that we run ZO-SVRG with step sizes $\alpha_t \le C/\left(sm(\beta(\Gamma_d^k + 1) + \frac{LA_d}{\mu})\right)$ for $T$ steps. After $S$ stages, ZO-SVRG satisfies with Then, at stage $S$ we have $\mathbb{E}[\delta_S] \le \exp\frac{C\beta}{s}\left(1 + \frac{2\beta c}{s}\right)\mathbb{E}[\delta_{S-1}] + \frac{6L^2+3\beta+3\beta L}{\beta Lns}\exp\frac{C}{s}$. After $S$ stages, ZO-SVRG satisfies with $\epsilon_{gen} \le (2L^2 + \beta + \beta L)S^{3C}\exp C/(C\beta n)$.*

*Proof of Theorem 21.* Recall that $\eta = 1$, and then we have

$$
\begin{aligned}
\mathbb{E}[\delta_{t+1}] &\leq \left(1 + \alpha_t \beta \Gamma_d^k + \frac{\alpha_t L A_d}{\mu}\right) \mathbb{E}[\delta_t] + 2\left(\alpha_t \beta \Gamma_d^k + \frac{\alpha_t L A_d}{\mu}\right) \mathbb{E}[\delta_1] \\
&\quad + \frac{6L\left(\Gamma_d^k + 1\right)\alpha_t}{n} + 3\mu\beta C_d \alpha_t + \frac{A_d}{\mu n} \\
&\leq \left(1 + \frac{C}{sm}\right) \mathbb{E}[\delta_t] + \frac{2C}{sm} \mathbb{E}[\delta_1] + \frac{6L^2 C + 3\beta c + 3\beta L C}{n\beta L sm} \\
&\leq \left(1 + \frac{C}{sm}\right)^m \mathbb{E}[\delta_1] + \frac{2C}{sm} \sum_{t=0}^{m} \left(1 + \frac{C}{sm}\right)^t \delta_1 + \frac{6L^2 C + 3\beta C + 3\beta L C}{n\beta L sm} \sum_{t=0}^{m} \left(1 + \frac{C}{sm}\right)^t \\
&\leq \exp\frac{C}{s}\left(1 + \frac{2C}{s}\right) \mathbb{E}[\delta_1] + \frac{6L^2 + 3\beta + 3\beta L}{n\beta L s} \exp\frac{C}{s}.
\end{aligned}
$$

For the $S$-stage, we have $w_s = w_m$ and $w_{S-1} = w_1$. Then,

$$
\mathbb{E}[\delta_S] \leq \exp\frac{C}{s}\left(1 + \frac{2C}{s}\right) \mathbb{E}[\delta_{S-1}] + \frac{6L^2 + 3\beta + 3\beta L}{n\beta L s} \exp\frac{C}{s}.
$$

Summing the above inequality for $S$ stages, and unraveling the recursion gives

$$
\begin{aligned}
\mathbb{E}[\delta_S] &\leq \frac{6L^2 + 3\beta + 3\beta L}{n\beta L} \sum_{s=1}^{S} \frac{1}{s} \prod_{j=s+1}^{S} \exp\frac{3C}{j} \\
&\leq \frac{\left(6L^2 + 3\beta + 3\beta L\right) e^C S^{3C}}{n\beta L} \sum_{s=1}^{S} \frac{1}{s} \frac{1}{(s+1)^{3C}} \\
&\leq \frac{\left(6L^2 + 3\beta + 3\beta L\right) e^C S^{3C}}{n\beta L} \sum_{s=1}^{S} \frac{1}{s^{1+3C}} \\
&\leq \frac{\left(2L^2 + \beta + \beta L\right) e^C S^{3C}}{n\beta L C}.
\end{aligned}
$$

Note that $\mathbb{E}\left|f\left(w_T; z\right) - f\left(w_T'; z\right)\right| \leq L\mathbb{E}\left[\delta_T\right]$, then we have

$$
\epsilon_{\text{gen}} \leq \frac{\left(2L^2 + \beta + \beta L\right) e^C S^{3C}}{n\beta C}.
$$

$\square$

**Theorem 22** (strongly convex). *Assume that the loss function $f(\cdot; z)$ is $\beta$-smooth, $\gamma$-strongly convex, and L-Lipschitz for every $z$. Suppose that we run ZO-SVRG with step sizes $\alpha_t \leq C/\left(sm(\beta(\Gamma_d^k + 1) + \frac{LA_d}{\mu})\right)$ for $T$ steps. After $S$ stages, ZO-SVRG satisfies with Then, at stage $S$ we have $\mathbb{E}[\delta_S] \leq \exp\frac{C\beta}{s}\left(1 + \frac{2\beta c}{s}\right)\mathbb{E}[\delta_{S-1}] + \frac{6L^2 + 3\beta + 3\beta L}{\beta L n s}\exp\frac{C}{s}$. After $S$ stages, ZO-SVRG satisfies with*

$$
\epsilon_{gen} \leq (2L^2 + \beta + \beta L)S^{3C}\exp C/(C\beta n).
$$

*Proof of Theorem 22.* Recall that $\eta = 1 - \frac{\alpha_t \beta \gamma}{\beta + \gamma}$, and then we have

$$
\begin{aligned}
\mathbb{E}[\delta_{t+1}] &\leq \left(1 + \alpha_t \beta \Gamma_d^k - \frac{\alpha_t \beta \gamma}{\beta + \gamma} + \frac{\alpha_t L A_d}{\mu}\right) \mathbb{E}[\delta_t] + 2\left(\alpha_t \beta \Gamma_d^k + \frac{\alpha_t L A_d}{\mu}\right) \mathbb{E}[\delta_1] \\
&\quad + \frac{6L\left(\Gamma_d^k + 1\right)\alpha_t}{n} + 3\mu\beta C_d \alpha_t + \frac{A_d}{\mu n} \\
&\leq \left(1 + \alpha_t \beta(\Gamma_d^k + 1) + \frac{\alpha_t L A_d}{\mu}\right) \mathbb{E}[\delta_t] + 2\left(\alpha_t \beta \Gamma_d^k + \frac{\alpha_t L A_d}{\mu}\right) \mathbb{E}[\delta_1]
\end{aligned}
$$

$$+ \frac{6L\left(\Gamma_d^k + 1\right)\alpha_t}{n} + 3\mu\beta C_d\alpha_t + \frac{A_d}{\mu n}$$

$$\leq \left(1 + \frac{C}{sm}\right)\mathbb{E}[\delta_t] + \frac{2C}{sm}\mathbb{E}[\delta_1] + \frac{6L^2C + 3\beta c + 3\beta LC}{n\beta Lsm}$$

$$\leq \left(1 + \frac{C}{sm}\right)^m \mathbb{E}[\delta_1] + \frac{2C}{sm}\sum_{t=0}^{m}\left(1 + \frac{C}{sm}\right)^t \delta_1 + \frac{6L^2C + 3\beta C + 3\beta LC}{n\beta Lsm}\sum_{t=0}^{m}\left(1 + \frac{C}{sm}\right)^t$$

$$\leq \exp\frac{C}{s}\left(1 + \frac{2C}{s}\right)\mathbb{E}[\delta_1] + \frac{6L^2 + 3\beta + 3\beta L}{n\beta Ls}\exp\frac{C}{s}.$$

For the $S$-stage, we have $w_s = w_m$ and $w_{S-1} = w_1$. Then,

$$\mathbb{E}[\delta_S] \leq \exp\frac{C}{s}\left(1 + \frac{2C}{s}\right)\mathbb{E}[\delta_{S-1}] + \frac{6L^2 + 3\beta + 3\beta L}{n\beta Ls}\exp\frac{C}{s}.$$

Summing the above inequality for $S$ stages, and unraveling the recursion gives

$$\mathbb{E}[\delta_S] \leq \frac{6L^2 + 3\beta + 3\beta L}{n\beta L}\sum_{s=1}^{S}\frac{1}{s}\prod_{j=s+1}^{S}\exp\frac{3C}{j}$$

$$\leq \frac{\left(6L^2 + 3\beta + 3\beta L\right)e^C S^{3C}}{n\beta L}\sum_{s=1}^{S}\frac{1}{s}\frac{1}{(s+1)^{3C}}$$

$$\leq \frac{\left(6L^2 + 3\beta + 3\beta L\right)e^C S^{3C}}{n\beta L}\sum_{s=1}^{S}\frac{1}{s^{1+3C}}$$

$$\leq \frac{\left(2L^2 + \beta + \beta L\right)e^C S^{3C}}{n\beta LC}.$$

Note that $\mathbb{E}\left|f\left(w_T; z\right) - f\left(w_T'; z\right)\right| \leq L\mathbb{E}\left[\delta_T\right]$, then we have

$$\epsilon_{\text{gen}} \leq \frac{\left(2L^2 + \beta + \beta L\right)e^C S^{3C}}{n\beta C}.$$

$\square$

*Proof of Theorem 8.*

$$\mathbb{E}[\delta_{t+1}] \leq \left(1 + \alpha_t\beta\left(\Gamma_d^k + 1\right)\right)\mathbb{E}[\delta_t] + 2\alpha_t\beta\left(\Gamma_d^k + 1\right)\mathbb{E}[\delta_1] + \frac{6L\left(\Gamma_d^k + 1\right)\alpha_t}{n} + 3\mu\beta C_d\alpha_t$$

$$\leq \left(1 + \frac{C\beta}{sm}\right)\mathbb{E}[\delta_t] + \frac{2C\beta}{sm}\mathbb{E}[\delta_1] + \frac{6CL + 3cC}{nsm}$$

$$\leq \left(1 + \frac{C\beta}{sm}\right)^m \mathbb{E}[\delta_1] + \frac{2C\beta}{sm}\sum_{t=0}^{m}\left(1 + \frac{C\beta}{sm}\right)^t \delta_1 + \frac{6LC + 3cC}{nsm}\sum_{t=0}^{m}\left(1 + \frac{C\beta}{sm}\right)^t$$

$$\leq \exp\frac{C\beta}{s}\left(1 + \frac{2C\beta}{s}\right)\mathbb{E}[\delta_1] + \frac{6L + 3c}{n\beta}\exp\frac{C\beta}{s}.$$

For the $S$-stage, we have $w_s = w_m$ and $w_{S-1} = w_1$. Then,

$$\mathbb{E}[\delta_S] \leq \exp\frac{C\beta}{s}\left(1 + \frac{2C\beta}{s}\right)\mathbb{E}[\delta_{S-1}] + \frac{6L + 3c}{n\beta}\exp\frac{C\beta}{s}.$$

Summing the above inequality for $S$ stages, and unraveling the recursion gives

$$\mathbb{E}[\delta_S] \leq \frac{(6L + 3c)e^{C\beta}}{n\beta}\sum_{s=1}^{S}\frac{1}{s}\prod_{j=s+1}^{S}\exp\frac{3C\beta}{j}$$

$$\leq \frac{(6L+3c)e^{C\beta}S^{3C\beta}}{n} \sum_{s=1}^{S} \frac{1}{s} \frac{1}{(s+1)^{3C\beta}}$$

$$\leq \frac{(6L+3c)e^{C\beta}S^{3C\beta}}{n} \sum_{s=1}^{S} \frac{1}{s^{1+3C\beta}}$$

$$\leq \frac{(2L+c)e^{C\beta}S^{3C\beta}}{n\beta}.$$

Note that $\mathbb{E}\left|f\left(w_T; z\right) - f\left(w_T'; z\right)\right| \leq L\mathbb{E}\left[\delta_T\right]$, then we have

$$\epsilon_{\text{gen}} \leq \frac{(2L^2+cL)e^{C\beta}S^{3C\beta}}{n\beta}.$$

$\square$

**Theorem 23** (convex). *Assume that the loss function $f\left(\cdot; z\right)$ is $\beta$-smooth, convex, and L-Lipschitz for every z. Suppose that we run ZO-SVRG with step sizes $\alpha_t \leq C/sm\left(\Gamma_d^k + 1\right)$ for T steps. After S stages, ZO-SVRG satisfies*

$$\epsilon_{\text{gen}} \leq (2L^2 + cL)S^{3\beta C} \exp C\beta/(\beta n).$$

*Proof of Theorem 23.* Recall $\eta = 1$, then we have

$$\mathbb{E}[\delta_{t+1}] \leq \left(1 + \alpha_t\beta\Gamma_d^k\right)\mathbb{E}[\delta_t] + 2\alpha_t\beta\Gamma_d^k\mathbb{E}[\delta_1] + \frac{6L\left(\Gamma_d^k + 1\right)\alpha_t}{n} + 3\mu\beta C_d\alpha_t$$

$$\leq \left(1 + \frac{C\beta}{sm}\right)\mathbb{E}[\delta_t] + \frac{2C\beta}{sm}\mathbb{E}[\delta_1] + \frac{6CL + 3cC}{nsm}$$

$$\leq \left(1 + \frac{C\beta}{sm}\right)^m\mathbb{E}[\delta_1] + \frac{2C\beta}{sm}\sum_{t=0}^{m}\left(1 + \frac{C\beta}{sm}\right)^t\delta_1 + \frac{6LC + 3cC}{nsm}\sum_{t=0}^{m}\left(1 + \frac{C\beta}{sm}\right)^t$$

$$\leq \exp\frac{C\beta}{s}\left(1 + \frac{2C\beta}{s}\right)\mathbb{E}[\delta_1] + \frac{6L + 3c}{n\beta}\exp\frac{C\beta}{s}.$$

For the $S$-stage, we have $w_s = w_m$ and $w_{S-1} = w_1$. Then,

$$\mathbb{E}[\delta_S] \leq \exp\frac{C\beta}{s}\left(1 + \frac{2C\beta}{s}\right)\mathbb{E}[\delta_{S-1}] + \frac{6L + 3c}{n\beta}\exp\frac{C\beta}{s}.$$

Summing the above inequality for $S$ stages, and unraveling the recursion gives

$$\mathbb{E}[\delta_S] \leq \frac{(6L+3c)e^{C\beta}}{n\beta}\sum_{s=1}^{S}\frac{1}{s}\prod_{j=s+1}^{S}\exp\frac{3C\beta}{j}$$

$$\leq \frac{(6L+3c)e^{C\beta}S^{3C\beta}}{n}\sum_{s=1}^{S}\frac{1}{s}\frac{1}{(s+1)^{3C\beta}}$$

$$\leq \frac{(6L+3c)e^{C\beta}S^{3C\beta}}{n}\sum_{s=1}^{S}\frac{1}{s^{1+3C\beta}}$$

$$\leq \frac{(2L+c)e^{C\beta}S^{3C\beta}}{n\beta}.$$

Note that $\mathbb{E}\left|f\left(w_T; z\right) - f\left(w_T'; z\right)\right| \leq L\mathbb{E}\left[\delta_T\right]$, then we have

$$\epsilon_{\text{gen}} \leq \frac{(2L^2+cL)e^{C\beta}S^{3C\beta}}{n\beta}.$$

$\square$

**Theorem 24** (strongly convex). *Assume that the loss function $f(\cdot; z)$ is $\beta$-smooth, convex,and L-Lipschitz for every $z$. Suppose that we run ZO-SVRG with step sizes $\alpha_t \leq C/sm \left(\Gamma_d^k + 1\right)$ for $T$ steps. After $S$ stages, ZO-SVRG satisfies*

$$\epsilon_{gen} \leq (2L^2 + cL)S^{3\beta C} \exp C\beta/(\beta n).$$

*Proof of Theorem 24.* Recall $\eta = 1 - \frac{\alpha_t \beta \gamma}{\beta + \gamma}$, then we have

$$
\begin{aligned}
\mathbb{E}[\delta_{t+1}] &\leq \left(1 + \alpha_t \beta \Gamma_d^k - \frac{\alpha_t \beta \gamma}{\beta + \gamma}\right) \mathbb{E}[\delta_t] + 2\alpha_t \beta \Gamma_d^k \mathbb{E}[\delta_1] + \frac{6L\left(\Gamma_d^k + 1\right)\alpha_t}{n} + 3\mu\beta C_d \alpha_t \\
&\leq \left(1 + \alpha_t \beta(\Gamma_d^k + 1)\right) \mathbb{E}[\delta_t] + 2\alpha_t \beta \Gamma_d^k \mathbb{E}[\delta_1] + \frac{6L\left(\Gamma_d^k + 1\right)\alpha_t}{n} + 3\mu\beta C_d \alpha_t \\
&\leq \left(1 + \frac{C\beta}{sm}\right) \mathbb{E}[\delta_t] + \frac{2C\beta}{sm} \mathbb{E}[\delta_1] + \frac{6CL + 3cC}{nsm} \\
&\leq \left(1 + \frac{C\beta}{sm}\right)^m \mathbb{E}[\delta_1] + \frac{2C\beta}{sm} \sum_{t=0}^{m} \left(1 + \frac{C\beta}{sm}\right)^t \delta_1 + \frac{6LC + 3cC}{nsm} \sum_{t=0}^{m} \left(1 + \frac{C\beta}{sm}\right)^t \\
&\leq \exp \frac{C\beta}{s} \left(1 + \frac{2C\beta}{s}\right) \mathbb{E}[\delta_1] + \frac{6L + 3c}{n\beta} \exp \frac{C\beta}{s}.
\end{aligned}
$$

For the $S$-stage, we have $w_s = w_m$ and $w_{S-1} = w_1$. Then,

$$\mathbb{E}[\delta_S] \leq \exp \frac{C\beta}{s} \left(1 + \frac{2C\beta}{s}\right) \mathbb{E}[\delta_{S-1}] + \frac{6L + 3c}{n\beta} \exp \frac{C\beta}{s}.$$

Summing the above inequality for $S$ stages, and unraveling the recursion gives

$$
\begin{aligned}
\mathbb{E}[\delta_S] &\leq \frac{(6L + 3c)e^{C\beta}}{n\beta} \sum_{s=1}^{S} \frac{1}{s} \prod_{j=s+1}^{S} \exp \frac{3C\beta}{j} \\
&\leq \frac{(6L + 3c)e^{C\beta}S^{3C\beta}}{n} \sum_{s=1}^{S} \frac{1}{s} \frac{1}{(s+1)^{3C\beta}} \\
&\leq \frac{(6L + 3c)e^{C\beta}S^{3C\beta}}{n} \sum_{s=1}^{S} \frac{1}{s^{1+3C\beta}} \\
&\leq \frac{(2L + c)e^{C\beta}S^{3C\beta}}{n\beta}.
\end{aligned}
$$

Note that $\mathbb{E}\left|f\left(w_T; z\right) - f\left(w_T'; z\right)\right| \leq L\mathbb{E}\left[\delta_T\right]$, then we have

$$\epsilon_{\text{gen}} \leq \frac{(2L^2 + cL)e^{C\beta}S^{3C\beta}}{n\beta}.$$

$\square$

*Proof of Theorem 9 (Nonconvex).*

$$
\begin{aligned}
\mathbb{E}[\delta_{t+1}] &\leq (1 + \alpha_t \beta) \mathbb{E}[\delta_t] + 2\alpha_t \beta \mathbb{E}[\delta_1] + \frac{6L\alpha_t}{n} + 3\mu\beta\alpha_t \\
&\leq \left(1 + \frac{C\beta}{sm}\right) \mathbb{E}[\delta_t] + \frac{2C\beta}{sm} \mathbb{E}[\delta_1] + \frac{6CL + 3cC}{nsm} \\
&\leq \left(1 + \frac{C\beta}{sm}\right)^m \mathbb{E}[\delta_1] + \frac{2C\beta}{sm} \sum_{t=0}^{m} \left(1 + \frac{C\beta}{sm}\right)^t \delta_1 + \frac{6LC + 3cC}{nsm} \sum_{t=0}^{m} \left(1 + \frac{C\beta}{sm}\right)^t \\
&\leq \exp \frac{C\beta}{s} \left(1 + \frac{2C\beta}{s}\right) \mathbb{E}[\delta_1] + \frac{6L + 3c}{n\beta} \exp \frac{C\beta}{s}.
\end{aligned}
$$

For the $S$-stage, we have $w_s = w_m$ and $w_{S-1} = w_1$. Then,

$$\mathbb{E}[\delta_S] \leq \exp \frac{C\beta}{s} \left(1 + \frac{2C\beta}{s}\right) \mathbb{E}[\delta_{S-1}] + \frac{6L + 3c}{n\beta} \exp \frac{C\beta}{s}..$$

Summing the above inequality for $S$ stages, and unraveling the recursion gives

$$\mathbb{E}[\delta_S] \leq \frac{(6L + 3c)e^{C\beta}}{n\beta} \sum_{s=1}^{S} \frac{1}{s} \prod_{j=s+1}^{S} \exp \frac{3C\beta}{j}$$

$$\leq \frac{(6L + 3c)e^{C\beta}S^{3C\beta}}{n} \sum_{s=1}^{S} \frac{1}{s} \frac{1}{(s+1)^{3C\beta}}$$

$$\leq \frac{(6L + 3c)e^{C\beta}S^{3C\beta}}{n} \sum_{s=1}^{S} \frac{1}{s^{1+3C\beta}}$$

$$\leq \frac{(2L + c)e^{C\beta}S^{3C\beta}}{n\beta}.$$

Note that $\mathbb{E}|f(w_T; z) - f(w'_T; z)| \leq L\mathbb{E}[\delta_T]$, then we have

$$\epsilon_{\text{gen}} \leq \frac{(2L^2 + cL)e^{C\beta}S^{3C\beta}}{n\beta}.$$

$\square$

*Proof of Theorem 9 (Convex).*

$$\mathbb{E}[\delta_{t+1}] \leq \mathbb{E}[\delta_t] + 2\alpha_t\beta\mathbb{E}[\delta_1] + \frac{(6L + 3c)\alpha_t}{n}$$

$$\leq \frac{(6L + 3c)}{n} \sum_{t=1}^{m} \alpha_t + \left(2\beta \sum_{n=1}^{m} \alpha_t + 1\right) \mathbb{E}[\delta_1]$$

$$\leq \frac{6L + 3c}{n} \sum_{t=1}^{m} \frac{C}{ms} + \left(2\beta \sum_{n=1}^{m} \frac{C}{ms} + 1\right) \mathbb{E}[\delta_1]$$

$$\leq \frac{6CL + 3cC}{ns} + \left(\frac{2\beta C}{s} + 1\right) \mathbb{E}[\delta_1].$$

For the S-stage, we have $w_s = w_m$ and $w_{S-1} = w_1$. Then,

$$\mathbb{E}[\delta_S] \leq \frac{6cL + 3cC}{ns} + \left(\frac{2\beta c}{s} + 1\right) \mathbb{E}[\delta_{S-1}].$$

Summing the above inequality for $S$ stages, and unraveling the recursion gives

$$\mathbb{E}[\delta_S] \leq \frac{6CL + 3cC}{n} \sum_{s=1}^{S} \frac{1}{s} \prod_{j=s+1}^{S} \left(1 + \frac{2\beta c}{j}\right)$$

$$\leq \frac{(6L + 3c)(eS)^{2\beta c}}{n} \min\left\{c + \beta^{-1}, c\log(eS)\right\}.$$

Note that $\mathbb{E}|f(w_T; z) - f(w'_T; z)| \leq L\mathbb{E}[\delta_T]$, then we have

$$\epsilon_{\text{gen}} \leq \frac{(6L^2 + 3cL)(eS)^{2\beta c}}{n} \min\left\{c + \beta^{-1}, c\log(eS)\right\}.$$

$\square$

*Proof of Theorem 9 (Strongly Convex).*

$$\mathbb{E}[\delta_t + 1] \leq (1 - \alpha\gamma) \,\mathbb{E}[\delta_t] + 2\alpha\beta[\delta_1] + \frac{(6L + 3c)\alpha}{n}$$

$$= (1 - \alpha\gamma)^m \,\mathbb{E}[\delta_1] + 2\alpha\beta \sum_{t=1}^{m-1} (1 - \alpha\gamma)^t \,\mathbb{E}[\delta_1] + \frac{(6L + 3c)\alpha}{n} \sum_{t=0}^{m-1} (1 - \alpha\gamma)^t$$

$$\leq \frac{(6L + 3c)\alpha}{n} \left( \frac{1 - (1 - \alpha\gamma)^m}{\alpha\gamma} \right) + \left( (1 - (1 - \alpha\gamma)^m) \frac{2\beta}{\gamma} + (1 - \alpha\gamma)^m \right) \mathbb{E}[\delta_1]$$

$$\leq \mathbb{E}[\delta_1] + \frac{6L + 3c}{n\gamma}.$$

For the S-stage, we have $w_s = w_m$ and $w_{S-1} = w_1$. Then,

$$\mathbb{E}[\delta_S] \leq \mathbb{E}[\delta_{S-1}] + \frac{6L + 3c}{n\gamma}.$$

Summing the above inequality for S stages, and unraveling the recursion gives

$$\mathbb{E}[\delta_S] \leq \frac{6LS + 3cS}{n\gamma}.$$

Note that $\mathbb{E}\left| f\left(w_T; z\right) - f\left(w_T'; z\right)\right| \leq L\mathbb{E}\left[\delta_T\right]$, then we have

$$\epsilon_{\text{gen}} \leq \frac{6SL^2 + 3cLS}{n\gamma}.$$

$\square$