# OpenReview forum: "General Stability Analysis for Zeroth-Order Optimization Algorithms"
_ICLR.cc/2024/Conference — ICLR 2024 poster_

### Official Review · Reviewer_N2Y8 · 2023-10-19

**Soundness:** 3 good
**Presentation:** 3 good
**Contribution:** 3 good
**Rating:** 8
**Confidence:** 3

**Summary:**

This paper considers the generalization bound of zero-order methods and presents a general analysis framework. Results on 2-point, 1-point, and coordinate-wise gradient estimators are established, which improves the existing work in this direction. Based on these results, the authors show that coordinate estimation leads to tighter generalization bounds for many zeroth-order methods. Experiments are also provided to verify the theoretical conclusions.

**Strengths:**

1. The paper is well-presented and well-organized. The motivation, technique, and results are clearly stated.

2. The framework established is a nice theoretical contribution. Consequently, the generalization bounds of many zero-order methods are developed. Moreover, the differences between these generalization bounds are captured and analyzed, making the whole theory complete and convincing.

3. The technique used in the paper is solid and interesting.

**Weaknesses:**

It would be better if the author emphasized the new technique and idea used in this paper compared with the existing work by Nikolakakis et al. (2022).

**Questions:**

In Figure 1 (b), why did the generalization bound of ZO-SVRG decrease at the beginning?

**Details Of Ethics Concerns:**

Non.

---

> ### Author Response · Authors · 2023-11-20
> **To N2Y8**
>
> **Q1:**
> It would be better if the author emphasized the new technique and idea used in this paper compared with the existing work by Nikolakakis et al. (2022).
>
> **A1:**
> Thanks for your valuable comments. There exist  key differences between (Nikolakakis et al. 2022) and our work, which are summarized as below:
>
> 1) Objective function: **ZO-SGD with 2-point gradient estimator Vs. general ZO optimization.** Nikolakakis et al. (2022) merely considered the generalization bound of ZO-SGD with 2-point gradient estimator. In this paper, we established a unified generalization analysis framework  for  ZO-GD, ZO-SGD as well as ZO-SVRG algorithms with 1-point / 2-point / coordinate-wise gradient estimators, which cover the mainstream estimators in existing zeroth-order algorithms.
>
> 2) Conditions: **Gaussian distribution and non-convex condition Vs. general distribution and convex conditions**.  Nikolakakis et al. (2022) just considered the $u_k\sim\mathcal{N}\left(0, I_d\right)$ and non-convex loss function. In this paper, we derive the generalization bounds for more distributions (e.g.,uniform $u_k\sim \mathcal{U}\left(\sqrt{d+2} \mathbb{B}^d\right)$, $u_k\sim \mathcal{U}\big(\sqrt{d} \mathbb{S}^{d-1}\big)$ see **Appendix E**) and wider convexity conditions (strongly convex see **Theorem 3,6,9 and Appendix G, H, I**, convex see **Theorem 3,6,9 and Appendix G,H, I** and non-convex **Theorem 1-9**).
>
> 3)  **Filling the gap on the  generalization bounds of ZO-SVRG.** ZO-SVRG is a  stage-wise algorithm with multiplication and summation from previous stages, which is totally different from ZO-SGD. To the best of our knowledge, this paper is the first endeavor to investigate the ZO-SVRG's generalization behavior. Meanwhile, our derived estimations of ZO-SGD $\Big({O}\big(T^{\frac{\beta C}{\beta C + 1}}/n\big)\Big)$ are tighter than the related results $\Big({O}\big(T/n\big)\Big)$ in Nikolakakis et al. 2022.
>
> We have polished the comparisons with the related works and highlighted our contributions (see **Appendix J**).
>
>
>
> **Q2:**
> In Figure 1(b), why did the generalization bound of ZO-SVRG decrease at the beginning?
>
> **A2:**
> As shown in Figure 1(b), ZO-SVRG still lies in the fitting stage at the beginning. The training loss, testing loss as well as their difference are all declining due to random initialization.
>
> To enrich the empirical evaluations, we have incorporated new results in the revision in **Section 5 Numerical Experiments and Appendix B**. Particularly,
> we have compared the generalization gaps of several estimators for zeroth-order algorithms (see **Figure 2**).
> Extended experiments verify our theoretical findings on generalization bounds.

---

### Official Review · Reviewer_x656 · 2023-10-28

**Soundness:** 4 excellent
**Presentation:** 4 excellent
**Contribution:** 3 good
**Rating:** 8
**Confidence:** 3

**Summary:**

In this paper, a new simple framework for analyzing the generalization error of zeroth order optimization for Lipschitz and smooth objective functions is proposed and several novel generalization bounds are provided. The framework proposed consists of a method for analyzing the stability of a given optimization algorithm, based on the notions of boundedness and expansivity of the corresponding update rule. The update rules considered all correspond to zeroth order analogues of first order optimization algorithms, where a first order oracle is approximately simulated using queries to the values of the objective function (i.e., a zeroth order oracle). The simulated oracles considered are based on either 1-point, 2-point or coordinate-wise approximation, and the latter achieves the tightest generalization bounds for each of the algorithms considered, matching, in each case, the best known bounds achievable by the corresponding first order algorithms in each of the cases where the objective is strongly convex, convex or non-convex. The algorithms considered are Gradient Descent, Stochastic Gradient Descent as well as Stochastic Variance Reduced Gradient method. The theoretical guarantees of the paper are accompanied by experimental results on real world data.

**Strengths:**

The results provided in this paper, which are obtained by leveraging a simple yet powerful framework and the idea to approximate the gradient of a smooth and Lipschitz function coordinate-wise, are strong and, to the best of my knowledge, novel. The presentation of the results is clear and detailed.

**Weaknesses:**

One potential weakness of the paper is that the results assume that the objective function is smooth, which, in many important optimization problems (e.g., learning ReLU networks) is not true.

**Questions:**

Could your results be extended to the case where the objective function is not smooth? Are there any generalization bounds for this setting, even for first order algorithms?

---

> ### Author Response · Authors · 2023-11-20
> **To x656**
>
> **Q1:**
> Could your results be extended to the case where the objective function is not smooth? Are there any generalization bounds for this setting, even for first order algorithms?
>
> **A1:**
> Thanks for your constructive comments. Indeed, there exist several works on non-smooth settings for first-order optimization, see e.g., [Ref1][Ref2]. However, these researches are limited to the convex or weakly convex settings.
> Moreover, the absence of smooth condition posed challenges on bounding the approximation error induced by zeroth-order optimization. Our analysis framework established here can not be extended to the non-smooth setting directly.
> We are striving to address this issue in the future work.
>
>
> We also added the related discussions in the revised paper, see **new Remark 3 at Appendix J**.
>
>
>
> [Ref1] Bassily R, Feldman V, Guzmán C, et al. Stability of stochastic gradient descent on nonsmooth convex losses, NeurIPS, 2020.
>
> [Ref2]Lei Y. Stability and generalization of stochastic optimization with nonconvex and nonsmooth problems, ICML, 2023.

---

> > ### Comment · Reviewer_x656 · 2023-11-21
> >
> > Thank you for your response. My rating remains the same.

---

> > > ### Author Response · Authors · 2023-11-23
> > > **To x656**
> > >
> > > Thank you very much for your valuable suggestions, which have helped improve our paper.  Thank you again for your review.

---

### Official Review · Reviewer_qUEd · 2023-10-30

**Soundness:** 3 good
**Presentation:** 3 good
**Contribution:** 2 fair
**Rating:** 6
**Confidence:** 4

**Summary:**

This paper focuses on zeroth-order optimization algorithms. While previous stability analysis results were limited to the basic 2-point zeroth-order estimate with a Gaussian distribution in stochastic gradient descent (SGD) algorithms, this paper introduces a general proof framework for stability analysis. This framework is applicable to convex, strongly convex, and non-convex conditions and provides results for various zeroth-order optimization algorithms, including SGD, gradient descent (GD), and stochastic variance-reduced gradient (SVRG) methods. It also covers different zeroth-order estimates, such as 1-point and 2-point estimates with various distributions and coordinate estimates. The general analysis reveals that coordinate estimation can lead to improved generalization bounds for SGD, GD, and SVRG versions of zeroth-order optimization algorithms by reducing the expansion in stability analysis.

**Strengths:**

- The paper exhibits a well-organized structure, including a clear motivation, an extensive literature review, and a rigorous theoretical analysis. However, I have not verified the validity of all statements in the Appendix.
- One intriguing and original contribution of the paper is its theoretical assertion that coordinate estimation can enhance the generalization bounds for zeroth-order optimization algorithms like SGD, GD, and SVRG.

**Weaknesses:**

- Theoretical contributions, especially those stemming from the primary theoretical lemmas (Lemma 3 and 4), are incremental compared to prior works such as (Hardt et al., 2016) and (Nikolakakis et al., 2022).
- The paper could benefit from more extensive numerical experiments and a more detailed implementation section. Specifically, it should include a comparison of the performance of various zeroth-order estimators, providing empirical support for the favorable theoretical results associated with coordinate estimation.

(Hardt et al., 2016) Moritz Hardt, Ben Recht, and Yoram Singer. Train faster, generalize better: Stability of stochastic gradient descent. In International conference on machine learning, pp. 1225–1234. PMLR, 2016.

(Nikolakakis et al., 2022) Konstantinos Nikolakakis, Farzin Haddadpour, Dionysis Kalogerias, and Amin Karbasi. Black-box generalization: Stability of zeroth-order learning.

**Questions:**

Majors:

1. What is the zeroth-order estimator used in the experiments?
2. How do different zeroth-order estimators affect the optimization performance?

Minors:
- Please indicate beta and C in Table 1.
- Figure 2(b):  Generalization error: GD - >  Generalization error: SGD?
- More implementation details are needed.
- Utilize parenthetical citations to enhance readability.

---

> ### Author Response · Authors · 2023-11-20
> **To Reviewer qUEd**
>
> **Q1:**
> Theoretical contributions, especially those stemming from the primary theoretical lemmas (Lemma 3 and 4), are incremental compared to prior works such as (Hardt et al., 2016) [2] and (Nikolakakis et al., 2022) [1].
>
> **A1:**
> Thanks for your constructive comments. The zeroth-order optimization often is considered as an approximation of the first-order optimization. Naturally, it enjoys some similar strategy for error analysis.  However, compared to prior works (e.g., Hardt et al., 2016 [Ref1], Nikolakakis et al., 2022 [Ref2]), there are key differences stated as below:
>
> 1) **Unified generalization analysis framework for zeroth-order optimization.**
> The existing analysis techniques (Hardt et al., 2016 [Ref1], Nikolakakis et al., 2022[Ref2]) mainly focus on ZO-SGD, and  can not be  applied to the other zeroth-order optimization approach (e.g., ZO-SVRG) directly.  And our proof framework can encompass the generalization analysis of ZO-GD, ZO-SGD, and ZO-SVRG algorithms under various convexity conditions.
>
>
>
> 2) **First stability analysis for  zeroth-order SVRG algorithm.**
> In contrast to SGD and GD algorithms, SVRG is a stagewise algorithm with new expansion factors among each stage and its stability-based generalization is unexplored for both first-order and zeroth-order settings. The update strategy of SVRG makes   the existing theoretical techniques  in [Ref1] and [Ref2] are not applicable directly. Here, we fill this theoretical gap by establishing  stability-based error bounds for zeroth-order SVRG,   where analysis technique are developed by the fine-grained  error decomposition and estimations.
>
>
>
> 3) **Diverse estimations for zeroth-order approximation.** Nikolakakis et al. 2022 [Ref2] only considered the two-point estimator under Gaussian distribution. In contrast, this paper investigates one-point, two-point, and coordinate-wise estimators under different distributions and two difference forms. Theoretical results demonstrate that the smaller approximation error of these estimators are, the better generalization performance they get. The coordinate-wise estimators usually enjoy smaller approximation error. In addition, compared to first-order algorithms, coordinate-wise estimators can achieve similar generalization results under different convexity conditions.
> In particular,  we have empirically verified that the zeroth-order algorithms with coordinate-wise estimator usually own the smallest generalization bounds compared with those using other estimators.
>
> In summary, this paper provides a unified framework for generalization analysis of zeroth-order optimization and fills the theoretical gap on the stability-based generalization for SVRG algorithm. Based on your important comments, we have added the above illustrations in (**Section 1: Introduction** and **Appendix J** ) to highlight our work clearly.
>
>
>  [Ref1] M. Hardt et al., Train faster, generalize better: Stability of stochastic gradient descent, International conference on machine learning (ICML), 2016.
>
> [Ref2] K. Nikolakakis et al., Black-box generalization: Stability of zeroth-order learning. Advances in Neural Information Processing Systems (NeurIPS), 2022.
>
>
>
> **Q2:**
> The paper could benefit from more extensive numerical experiments and a more detailed implementation section.
>
> **A2:**
> Thanks for your constructive comments. In the revised version, we have conducted more numerical experiments to verify the theoretical results, where the implementation details are also provided (see **Section 5.1 Experimental Setups**).
>
> Firstly, in **Section 5.2**, we compare the generalization errors of different ZO algorithms associated with the same gradient estimators. Experimental results in **Figure 1** show that ZO-SGD can achieve the competitive generalization performance compared with ZO-GD and ZO-SVRG.
>
> Secondly, we compare the generalization errors of same ZO algorithms with  different gradient estimators in  **Section 5.3 and Appendix B**. Experimental results in **Figure 2** demonstrate that ZO algorithms with coordinate-wise gradient estimator usually have the lowest generalization error.
>
> **Q3:**
> What is the zeroth-order estimator used in the experiments?
>
> **A3:**
> In the experiments, we employ the coordinate-wise estimator  as the zeroth-order estimator.  Implementing code can be found in  the **Supplementary Material**, lines 45-59 of the algs.py file.

---

> > ### Author Response · Authors · 2023-11-20
> > **To Reviewer qUEd**
> >
> > **Q4:**
> > How do different zeroth-order estimators affect the optimization performance?
> >
> > **A4:**
> > Indeed, different zeroth-order estimators affect the optimization performance by their approximation errors and variances with respect to the true gradient.
> >
> > Specifically, 1-point gradient estimator may suffer from high variance (the variance explodes as $\mu$ goes to 0). Despite of this drawback in theory, it is still a useful black-box optimization algorithm for certain applications, e.g., reinforcement learning [Ref3].
> >
> > For 2-point gradient estimator, the variance is bounded under forward difference or central difference structure. Thus, the relevant 2-point gradient estimator is more stable than 1-point one [Ref4]. Normally, zeroth-order optimization algorithms with 2-point gradient estimators own comparable convergence rate with respect to their first-order counterparts but may suffer from the dimension penalty of order $d$ [Ref5][Ref6].
> >
> > For the coordinate-wise gradient estimator, its variance often is zero from the  deterministic approximation procedure. The optimization performance of ZO algorithms with coordinate-wise gradient estimators is directly influenced by the approximation error. Moreover,  the approximation error can be  controlled by  $O(d\mu^2)$, where the smoothing parameter $\mu$  can be arbitrarily small [Ref7]. However, coordinate-wise gradient estimator needs to compute $O(d)$ function queries to achieve a more accurate approximation, which may not be applicable in high dimensional optimization problems.
> >
> >
> >
> > [Ref3] T. Salimans et al. Evolution strategies as a scalable alternative to reinforcement learning.
> >
> > [Ref4]A. D. Flaxman, A. T. Kalai, and H. B. McMahan, Online convex
> > optimization in the bandit setting: Gradient descent without a gradient,in Proc.16th Annual ACM-SIAM Symposium on Discrete algorithms(SODA), 2005.
> >
> > [Ref5] Nesterov Y, Spokoiny V. Random gradient-free minimization of convex functions.Foundations of Computational Mathematics(Found Comut Math), 2017.
> >
> > [Ref6] Ghadimi S, Lan G. Stochastic first-and zeroth-order methods for nonconvex stochastic programming. SIAM Journal on Optimization(SIOPT), 2013.
> >
> > [Ref7] Ji K, Wang Z, Zhou Y, et al. Improved zeroth-order variance reduced algorithms and analysis for nonconvex optimization. In International conference on machine learning(ICML), 2019.
> >
> >
> > **Q5:**
> > Please indicate beta and C in Table 1.
> >
> > **A5:**
> > We apologize for not explicitly indicating the meanings of these two symbols in the main text. In the latest submitted PDF, we have included explanations for them.
> > Beta is a constant of smoothness and $C$ is a positive constant.
> >
> > **Q6:**
> > Figure 2(b): Generalization error: GD - > Generalization error: SGD?
> >
> > **A6:**
> > In Figure 2(b), the Generalization error of GD is larger than that of SGD, aligning with our results and consistent with established theoretical findings [Ref8],[Ref9],[Ref10].
> >
> > [Ref8]Amir I, Koren T, Livni R. SGD generalizes better than GD (and regularization doesn’t help), Conference on Learning Theory(COLT), 2021.
> >
> > [Ref9]KeKeskar N S,et al. On Large-Batch Training for Deep Learning: Generalization Gap and Sharp Minima, International Conference on Learning Representations(ICLR), 2016.
> >
> > [Ref10]Sekhari A, Sridharan K, Kale S. Sgd: The role of implicit regularization, batch-size and multiple-epochs, Advances In Neural Information Processing Systems (NeurIPS), 2021.
> >
> > **Q7:**
> > More implementation details are needed.
> >
> > **A7:**
> > Thanks for your constructive comments. In our latest submission, we have incorporated additional implementation details and new comparative experiments (see **Section 5 Numerical Experiments**). Detailed experimental discussions have been included to analyze the generalization performance of ZO-SGD, ZO-GD, and ZO-SVRG, incorporating scenarios (see **Subsection 5.2**). Furthermore, we have introduced comparisons of the generalization bounds and convergence properties for multiple estimators (see **Subsection 5.3 and Appendix B**). More details also can be found in A2 to Q2.
> >
> > **Q8:**
> > Utilize parenthetical citations to enhance readability.
> >
> > **A8:**
> > Thanks. We have used the parenthetical citations to enhance the readability and further proofread the whole paper.

---

> > > ### Comment · Reviewer_qUEd · 2023-11-21
> > >
> > > Thank the authors for the response! The response and the revision has addressed all my concerns. As an result, I increase my rating to 6.

---

> > > > ### Author Response · Authors · 2023-11-23
> > > > **To Reviewer qUEd**
> > > >
> > > > We sincerely appreciate the time and effort you invested in reviewing our manuscript. Your insightful comments and suggestions have been invaluable in enhancing the quality of our work.

---

### Official Review · Reviewer_PNwi · 2023-10-31

**Soundness:** 3 good
**Presentation:** 3 good
**Contribution:** 3 good
**Rating:** 6
**Confidence:** 4

**Summary:**

In this submission, the authors proposed a general frame-work to prove the generalization error for the zero-order optimization methods. They proved the generalization error bounds for different zero-order optimization algorithms such as SG, GD and SVRG under different convexity conditions. They also conduct numerical experiments.

**Strengths:**

This submission is very clear with simple structures and languages. The main idea is natural and understandable. The mathematical and theoretical analysis is strict and the empirical results are consistent with the theoretical analysis.

**Weaknesses:**

There is no significant weakness for this submission.

Only one question is that for the ZO-SGD, ZO-GD and ZO-SVRG with one-point and two-point gradient estimation, the authors only presented the results for the none-convex setting. Is there any theoretical results for the general convex and strongly convex settings? Like the results for the coordinate-wise gradient estimation. If the authors could add these theoretical results, it could make the theoretical contributions much more complete.

**Questions:**

Please check the weakness section.

---

> ### Author Response · Authors · 2023-11-20
> **To Reviewer PNwi**
>
> **Q1:**
> Only one question is that for the ZO-SGD, ZO-GD and ZO-SVRG with one-point and two-point gradient estimation, the authors only presented the results for the non-convex setting. Is there any theoretical results for the general convex and strongly convex settings? Like the results for the coordinate-wise gradient estimation. If the authors could add these theoretical results, it could make the theoretical contributions much more complete.
>
> **A1:**
> Thanks for your constructive comments.convex and strongly convex conditions.
> Following your  valuable suggestions, we have added the corresponding generalization bounds  under convex and strongly convex conditions. For details, please refer to **Appendix (F.1, F.2, G.1, G.2, H.2)**.  We expect that the additional results can make our analysis more complete and improve its readability.
> Explicitly, the generalization bounds for ZO-SGD are  $O(T/n^{1-\frac{c}{d^2 L C \sqrt{n}}})$ and  $O(T/n^{1-\frac{c}{d^2 L C \sqrt{n}}})$  under one-point strongly convex condition and one-point convex conditions respectively. Meanwhile, generalization bounds for ZO-SGD are $O(T/n)$ under two-point strongly convex condition  and $O(T/n)$ under two-point convex condition. For ZO-GD, we present the generalization bounds $O(T^{C}/n)$ under one-point strongly convex condition, $O(T^{C}/n)$ under one-point convex condition, $O(T^{\beta C}/n)$ under two-point strongly convex condition, and $O(T^{\beta C}/n)$ under two-point convex condition. Similarly, ZO-SVRG exhibits generalization bounds $O(S^{3 C}/n)$ under one-point strongly convex condition, $O(S^{3 C}/n)$ under one-point convex condition, $O(S^{3 \beta C}/n)$ under two-point strongly convex condition, and $O(S^{3\beta C}/n)$ under two-point convex condition. Indeed, there are some preliminary discussions in Remark 1 in the original version.
>
> In essential, the convexity property has impact  on the expansion coefficient $\eta$ involving in the generalization bounds of first-order optimization methods (see **Appendix (Lemma 6)**) . It is easy to deduce that $\eta_t=1-\frac{\alpha_t\beta\gamma}{\beta+\gamma}$ (strong convex), $\eta = 1$ (convex), and $\eta_t = 1 + \alpha_t\beta$ (non-convex). However, for the zeroth-order case, there often is additional increment induced by the procedures of one-point or two-point estimation (Ref1). We can verify  $\eta^{\prime}_t=1+\alpha_t\left(\frac{L}{\mu}\mathbb{E}\|u\|+\beta\mathbb{E}\left[\|u\|^2\right]\right)$ (one-point convex condition), $\eta^{\prime}_t=1+\alpha_t\mu\beta\mathbb{E}\left[\|u\|^2\right]$ (two-point convex condition),  $\eta^{\prime}_t=1-\frac{\alpha_t\beta\gamma}{\beta+\gamma}+\alpha_t\left(\frac{L}{\mu}\mathbb{E}\|u\|+\beta\mathbb{E}\left[\|u\|^2\right]\right)$ (one-point strongly convex condition), and $\eta^{\prime}=1-\frac{\alpha_t\beta\gamma}{\beta+\gamma}+\alpha_t\mu\beta\mathbb{E}\left[\|u\|^2\right]$ (two-point strongly convex condition).
>
> Due to the additional increments (e.g., $\eta^{\prime}$ may exceed $1$), multiplication process.
> the additional results for the general convex and strongly convex settings are analogous to the existing error bounds under non-convex conditions (e.g., **Theorems 1,2,4,5,7,8**).
>
>
> [Ref1] K. Nikolakakis et al., Black-box generalization: Stability of zeroth-order learning. Advances in Neural Information Processing Systems (NeurIPS), 2022.

---

### Meta-Review · Area_Chair_b56a · 2023-12-07

**Metareview:**

This paper studies stability analysis for zeroth-order (ZO) optimization algorithms. Previous results along this line were limited to 2-point ZO estimate with Gaussian distribution in SGD. This paper provides a more general analysis which covers many important classes, including convex, strongly convex, and non-convex problems, and various ZO variants of SGD, GD, and SVRG. 1-point and 2-point coordinate estimates were also discussed. Overall, this is a solid theoretical contribution to ZO.

**Justification For Why Not Higher Score:**

I feel that the results are not significant enough to make a spotlight.

**Justification For Why Not Lower Score:**

Results are important, should be accepted.

---

### Decision · Program_Chairs · 2024-01-16

Accept (poster)